# Adjoint Matching: Fine-tuning Flow and Diffusion Generative Models with Memoryless Stochastic Optimal Control

**Carles Domingo-Enrich, Michal Drozdzal, Brian Karrer & Ricky T. Q. Chen**
Meta AI
cd2754@nyu.edu,{mdrozdzal,briankarrer,rtqichen}@meta.com

## Abstract

Dynamical generative models that produce samples through an iterative process, such as Flow Matching and denoising diffusion models, have seen widespread use, but there have not been many theoretically-sound methods for improving these models with reward fine-tuning. In this work, we cast reward fine-tuning as stochastic optimal control (SOC). Critically, we prove that a very specific *memoryless* noise schedule must be enforced during fine-tuning, in order to account for the dependency between the noise variable and the generated samples. We also propose a new algorithm named *Adjoint Matching* which outperforms existing SOC algorithms, by casting SOC problems as a regression problem. We find that our approach significantly improves over existing methods for reward fine-tuning, achieving better consistency, realism, and generalization to unseen human preference reward models, while retaining sample diversity.

Base model (Flow Matching) w/ Guidance        **Adjoint Matching (Ours)**

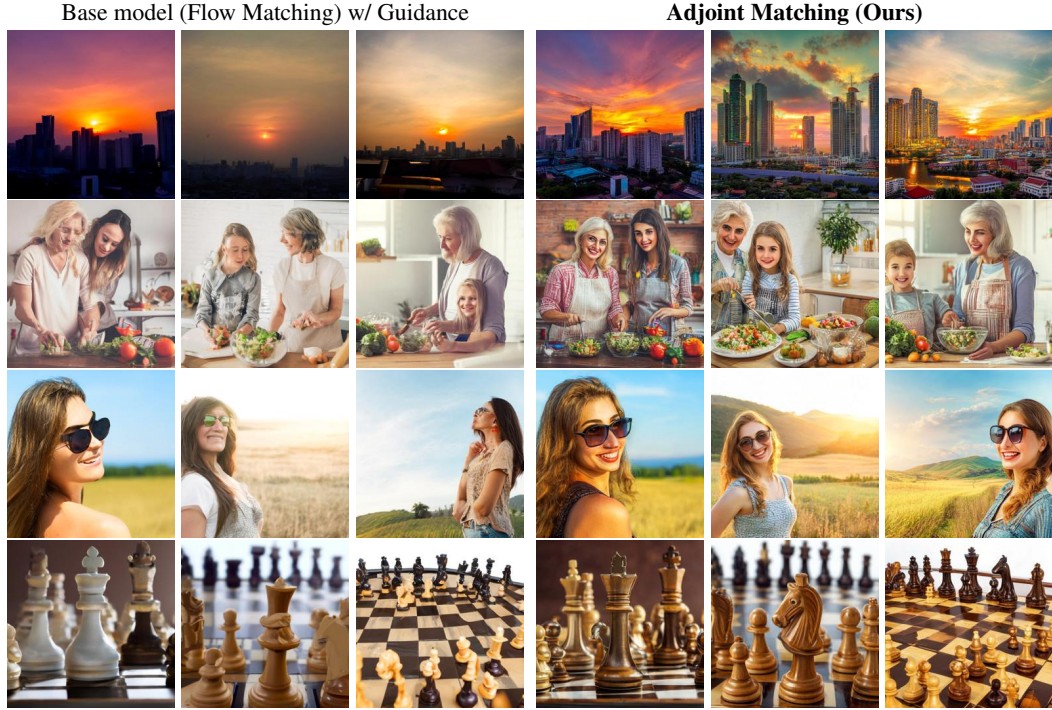

Figure 1: We introduce Adjoint Matching, a theoretically-driven yet simple algorithm for reward fine-tuning that works for a large family of dynamical generative models, including for the first time, Flow Matching models. Text prompts: "*Beautiful colorful sunset midst of building in Bangkok Thailand*", "*Beautiful grandma and granddaughter are mixing salad and smiling while cooking in kitchen*", "*The beautiful young woman in sunglasses is standing at the background of field and hill. She is smiling and looking over shoulder*", "*Chess, intellectual games, figure horse, chess board*".

# 1 INTRODUCTION

Flow Matching (Lipman et al., 2023; Albergo & Vanden-Eijnden, 2023; Liu et al., 2023) and denoising diffusion (Song & Ermon, 2019; Ho et al., 2020; Song et al., 2021b; Kingma et al., 2021) models are being used for many generative modeling applications, including text-to-image (Rombach et al., 2022; Esser et al., 2024), text-to-video (Singer et al., 2022), and text-to-audio (Le et al., 2024; Vyas et al., 2023). In most cases, the base generative model does not achieve the desired sample quality. To improve the generated samples, it is common to resort to techniques such as classifier-free guidance (Ho & Salimans, 2022; Zheng et al., 2023) to get better text-to-sample alignment, or to fine-tune using human preference reward models to improve sample quality and realism (Wallace et al., 2023a; Clark et al., 2024).

In the adjacent field of large language models, the behavior of the model is aligned to human preferences through fine-tuning with reinforcement learning from human feedback (RLHF). Either explicitly or implicitly, RLHF methods (Ziegler et al., 2020; Stiennon et al., 2020; Ouyang et al., 2022; Bai et al., 2022) assume a reward model $r(x)$ that captures human preferences, with the goal of modifying the base generative model such that it generates the following *tilted distribution*:

$$p^*(x) \propto p^{\text{base}}(x) \exp(r(x)), \tag{1}$$

where $p_{\text{base}}$ is the base generative model's sample distribution.

Inspired by this, fine-tuning methods have been developed to improve denoising diffusion models based on human preference data; either using a reward-based approach (Fan & Lee, 2023; Black et al., 2024; Fan et al., 2023; Xu et al., 2023; Clark et al., 2024; Uehara et al., 2024a;b), or direct preference optimization (Wallace et al., 2023a). However, unlike the fine-tuning methods designed for large language models, most of the existing methods to a large degree ignore $p^{\text{base}}$ and focus solely on the reward model. Reward models can range from standard evaluation metrics such as ClipScore (Hessel et al., 2021; Kirstain et al., 2023) to specialized models that have been trained on human preferences (Schuhmann & Beaumont, 2022; Xu et al., 2023; Wu et al., 2023c). As these are parameterized by neural networks, they fall pray to adversarial examples which lead to the generation of undesirable artifacts (Goodfellow et al., 2014; Mordvintsev et al., 2015). This has led some works to consider adding regularization during fine-tuning (Fan et al., 2024; Uehara et al., 2024b) to incentivize staying close to the base model distribution; however, there does not yet exist a *simple*, generic approach which actually provably generates from the tilted distribution (1).

The main contributions of our paper are as follows:

 (i) We present a stochastic optimal control (SOC) formulation for reward fine-tuning of dynamical generative models. Importantly, we prove that the naïve approach considered by prior works lead to a *value function bias* problem that biases the fine-tuned model away from the tilted distribution (1). This problem has also been observed by Uehara et al. (2024b) but they propose a more complicated solution which involves training a separate generative model.

 (ii) Instead, we propose a very simple solution: the *memoryless noise schedule*. This is a unique noise schedule that completely removes the dependency between noise variables and the generated samples, resulting in provable convergence to the tilted distribution. This allows us to fine-tune dynamical generative models in full generality, including being the first to fine-tune noiseless Flow Matching models.

 (iii) We also propose a new method for solving SOC problems, called *Adjoint Matching*, which combines the scalability of gradient-based methods and the simplicity of a least-squares regression objective. This can be applied to general SOC problems, beyond reward fine-tuning.

 (iv) We perform extensive comparisons to baseline approaches, and analyze them from multiple perspectives such as realism, consistency, and diversity. We find that our proposed method provides generalization to unseen human preference reward models, better text-to-sample consistency, and retains good diversity.

# 2 PRELIMINARIES ON DYNAMICAL GENERATIVE MODELS

We are interested in fine-tuning base generative models $p^{\text{base}}(X_1)$ where samples are generated through the simulation of a stochastic process. That is, these models transform noise variables into a

sample through an iterative process. In particular, we discuss the specific constructions and sampling processes of Flow Matching (Lipman et al., 2023; Liu et al., 2023; Liu, 2022; Albergo & Vanden-Eijnden, 2023) and Denoising Diffusion Models (Ho et al., 2020; Song et al., 2021b;a). The goal of this section is to provide background information on these methods.

Given random variables from an initial distribution $\bar{X}_0 \sim p_0 = \mathcal{N}(0, I)$, and $\bar{X}_1$ which are distributed according to some data distribution, we define the reference flow $\bar{\boldsymbol{X}} = (\bar{X}_t)_{t \in [0,1]}$ where

$$\bar{X}_t = \beta_t \bar{X}_0 + \alpha_t \bar{X}_1, \tag{2}$$

where $(\alpha_t)_{t \in [0,1]}, (\beta_t)_{t \in [0,1]}$ are functions such that $\alpha_0 = \beta_1 = 0$ and $\alpha_1 = \beta_0 = 1$. Diffusion models and Flow Matching construct generative Markov processes $X_t$ with initial distribution $X_0 \sim \mathcal{N}(0, I)$ that result in flows $\boldsymbol{X} = (X_t)_{t \in [0,1]}$ with the same time marginals as the reference flow $\bar{\boldsymbol{X}}$, *i.e.*, the random variables $X_t$ and $\bar{X}_t$ have identical distribution for all times $t \in [0, 1]$. This implies $X_1$ has the same distribution as the data distribution, so simulating the Markov process from random noise $X_0$ is a way to generate artificial samples[1].

**Flow Matching.** We focus on Flow Matching here, and defer the overview of denoising diffusion models (DDIM; Song et al. (2021a), DDPM; Ho et al. (2020)) to App. C.1. In its simplest form, the generative Markov process of a Flow Matching model is an ordinary differential equation (ODE) of the form:

$$\mathrm{d}X_t = v(X_t, t)\,\mathrm{d}t, \qquad X_0 \sim \mathcal{N}(0, I). \tag{3}$$

where $v(X_t, t)$ is a parametric velocity that is optimized to match the derivative of the reference flow, *i.e.*, $v(X_t, t) = \arg\min_{\hat{v}} \mathbb{E} \left\| \hat{v}(\bar{X}_t, t) - \frac{\mathrm{d}}{\mathrm{d}t} \bar{X}_t \right\|^2$ (see *e.g.* Lipman et al. (2023) for details on pre-training Flow Matching models). It can then be proven that the solution of the generative process (3) has the same time marginals as the reference flow (Lipman et al., 2023; Liu, 2022; Albergo & Vanden-Eijnden, 2023), and a commonly used choice is $\alpha_t = t$ and $\beta_t = 1 - t$. One can also consider a family of stochastic differential equations (SDEs) with an arbitrary state-independent diffusion coefficient[2]:

$$\mathrm{d}X_t = \left( v(X_t, t) + \frac{\sigma(t)^2}{2\beta_t(\frac{\dot{\alpha}_t}{\alpha_t}\beta_t - \dot{\beta}_t)} \left( v(X_t, t) - \frac{\dot{\alpha}_t}{\alpha_t} X_t \right) \right) \mathrm{d}t + \sigma(t)\,\mathrm{d}B_t, \qquad X_0 \sim \mathcal{N}(0, I), \tag{4}$$

where $(B_t)_{t \geq 0}$ is a Brownian motion. The generative processes in (3) and (4) have the same time marginals. This can be seen by writing down the Fokker-Planck equations for (3) and (4), and observing that they are the same up to a cancellation of terms (Maoutsa et al., 2020). The diffusion coefficient $\sigma(t)$ in (4) is compensated by the second term in the drift.

**Flow Matching in terms of the score function.** We can unify both the Flow Matching and continuous-time DDIM generative processes as:

$$\mathrm{d}X_t = b(X_t, t)\,\mathrm{d}t + \sigma(t)\,\mathrm{d}B_t, \qquad X_0 \sim \mathcal{N}(0, I), \tag{5}$$

$$\text{where } b(x, t) = \kappa_t x + \left( \frac{\sigma(t)^2}{2} + \eta_t \right) \mathfrak{s}(x, t), \quad \kappa_t = \frac{\dot{\alpha}_t}{\alpha_t}, \quad \eta_t = \beta_t \left( \frac{\dot{\alpha}_t}{\alpha_t} \beta_t - \dot{\beta}_t \right) \tag{6}$$

where $(\alpha_t, \beta_t)$ are coefficients of the reference flow (2), and $\mathfrak{s}(x, t)$ is the score function—defined as the gradient of the log density of the random variable $X_t$. See App. C.4 and App. C.5 for the derivation of (5)-(6) for DDIM and Flow Matching. In Subsec. 3.3, we rely on this characterization to derive our fine-tuning procedure. This expression has been written before for DDIM, e.g. Bartosh et al. (2024a;b).

## 3 FINE-TUNING AS "MEMORYLESS" STOCHASTIC OPTIMAL CONTROL

We now discuss the crux of the problem: how to produce a fine-tuned generative model that produces samples $X_1$ which follow the tilted distribution involving a reward model (1). An obvious direction is to construct a *fine-tuning objective* involving both the base generative model and the

---

[1]In our derivations, we assume the base model has been trained perfectly during the pre-training phase.

[2]We use the common short-hand "over-dot" notation to denote the time derivative, *i.e.*, $\dot{x}_t = \frac{\mathrm{d}}{\mathrm{d}t} x_t$.

reward model, where the optimal solution results in a fine-tuned generative model for the tilted distribution. However, as we will explain, this turns out to be non-trivial, because a naïve formulation will introduce bias into the solution.

In Subsec. 3.1, we discuss the problem formulation of stochastic optimal control, a general framework for optimizing SDEs, and its relation to the maximum entropy reinforcement learning framework commonly used for RLHF fine-tuning. Next, in Subsec. 3.2, we discuss the *initial value function bias* problem which plagues existing approaches and so far has seen no simple solution. Finally, in Subsec. 3.3, we propose a novel simple solution that circumvents the bias problem, by enforcing a particular diffusion coefficient, the *memoryless noise schedule*, to be used during fine-tuning. This results in an extremely simple fine-tuning objective that provably converges to a model which generates the tilted distribution (1) without any statistical bias.

## 3.1 PRELIMINARIES ON THE STOCHASTIC OPTIMAL CONTROL PROBLEM FORMULATION

Stochastic optimal control (SOC; Bellman (1957); Fleming & Rishel (2012); Sethi (2018)) considers general optimization problems over stochastic differential equations, but we only need to consider a common instantiation, the control-affine problem formulation:

$$\min_{u \in \mathcal{U}} \mathbb{E}\left[ \int_0^1 \left( \tfrac{1}{2}\|u(X_t^u, t)\|^2 + f(X_t^u, t) \right) \mathrm{d}t + g(X_1^u) \right], \tag{7}$$

$$\text{s.t. } \mathrm{d}X_t^u = \left( b(X_t^u, t) + \sigma(t)u(X_t^u, t) \right) \mathrm{d}t + \sigma(t)\mathrm{d}B_t, \qquad X_0^u \sim p_0 \tag{8}$$

where in (8), $X_t^u \in \mathbb{R}^d$ is the state of the stochastic process, $u : \mathbb{R}^d \times [0, 1] \to \mathbb{R}^d$ is commonly referred to as the control vector field, $b : \mathbb{R}^d \times [0, 1] \to \mathbb{R}^d$ is a base drift, and $\sigma : [0, 1] \to \mathbb{R}^{d \times d}$ is the diffusion coefficient. These jointly define the *controlled process* $\boldsymbol{X}^u \sim p^u$ that we are interested in optimizing; often both $b$ and $\sigma$ are fixed and we only optimize over the control $u$.

As part of the objective functional (7), we have an affine control cost $\tfrac{1}{2}\|u(X_t^u, t)\|^2$, a running state cost $f : \mathbb{R}^d \times [0, 1] \to \mathbb{R}$ and a terminal state cost $g : \mathbb{R}^d \to \mathbb{R}$.

The stochastic optimal control (SOC) objective (7) can be decomposed recursively from the final time value. It is common to define the *cost functional* which is the expected future cost starting from state $x$ at time $t$:

$$J(u; x, t) := \mathbb{E}_{\boldsymbol{X} \sim p^u} \left[ \int_t^1 \left( \tfrac{1}{2}\|u(X_s, s)\|^2 + f(X_s, s) \right) \mathrm{d}s + g(X_1) \mid X_t = x \right]. \tag{9}$$

From here, the *value function* is defined as the optimal value of the cost functional[3] : $V(x, t) := \min_{u \in \mathcal{U}} J(u; x, t) = J(u^*; x, t)$, where $u^*$ is the *optimal control*, *i.e.*, minimizer of (7). Furthermore, a classical result is that the value function can be expressed in terms of the *uncontrolled* base process $p^{\text{base}}$ (Kappen (2005), see Domingo-Enrich et al. 2023, Eq. 8, App. B for a self-contained proof):

$$V(x, t) = -\log \mathbb{E}_{\boldsymbol{X} \sim p^{\text{base}}} \left[ \exp(- \int_t^1 f(X_s, s)\mathrm{d}s - g(X_1)) \mid X_t = x \right]. \tag{10}$$

A useful expression for the optimal control (which we will make use of in deriving the Adjoint Matching objective in Sec. 4) is that it is related to the gradient of the value function:

$$u^*(x, t) = -\sigma(t)^\top \nabla_x V(x, t) = -\sigma(t)^\top \nabla_x J(u^*, x, t). \tag{11}$$

**Relation to MaxEnt RL.**  Stochastic optimal control with the control-affine formulation (7) is the continuous-time equivalence of maximum entropy reinforcement learning (MaxEnt RL; Todorov (2006); Ziebart et al. (2008)) with a KL regularization instead of only an entropy regularization. In particular, by the Girsanov theorem (Thm. 2), the affine control cost is equivalent to a Kullback–Leibler (KL) divergence between the base process $p^{\text{base}}$, when $u = 0$, and the controlled process $p^u$, when conditioned on the same initial state $X_0$ (see App. D.4):

$$D_{\text{KL}}\big(p^u(\boldsymbol{X}|X_0) \,\|\, p^{base}(\boldsymbol{X}|X_0)\big) = \mathbb{E}_{\boldsymbol{X}^u \sim p^u} \left[ \int_0^1 \tfrac{1}{2}\|u(X_t^u, t)\|^2 \mathrm{d}t \right], \tag{12}$$

---

[3]Note that there is a slight difference in terminology between SOC and reinforcement learning, where our cost functional is referred to as the state value function and our value function is the optimal state value function in RL.

resulting in the KL-regularized RL interpretation of (7):

$$\max_{u \in \mathcal{U}} \mathbb{E}_{X_0 \sim p_0} \left[ \mathbb{E}_{\boldsymbol{X} \sim p^u(\cdot|X_0)} \left[ \int_0^1 -f(X_t^u, t) \mathrm{d}t - g(X_1^u) \right] - D_{\mathrm{KL}}(p^u(\boldsymbol{X}|X_0) \parallel p^{base}(\boldsymbol{X}|X_0)) \right],$$
(13)

where the negative state costs correspond to intermediate and terminal rewards in the RL interpretation. The KL divergence forces the optimal process to stay close to the base process.

## 3.2 THE INITIAL VALUE FUNCTION BIAS PROBLEM

We next discuss why naïvely adding a KL regularization does not lead to the tilted distribution (1). From (13), we can show that the optimal distribution conditioned on $X_0$ is[4]

$$p^*(\boldsymbol{X}|X_0) \propto p^{\mathrm{base}}(\boldsymbol{X}|X_0) \exp \left( - \int_0^1 f(X_t, t) \, \mathrm{d}t - g(X_1) \right).$$
(14)

This is analogous to the exponentiated reward distribution in MaxEnt RL (Rawlik et al., 2013), but the entropy regularizer is generalized to a KL regularization with respect to a prior distribution $p^{\mathrm{base}}$. In order to relate this to the tilted distribution (1) that we want to achieve for fine-tuning, first notice that the normalization constant of the right-hand side (RHS) of (14) is exactly the value function at $t = 0$:

$$\mathbb{E}_{\boldsymbol{X} \sim p^{\mathrm{base}}(\boldsymbol{X}|X_0)} \left[ \exp \left( - \int_0^1 f(X_t, t) \, \mathrm{d}t - g(X_1) \right) \right] = \exp \left( -V(X_0, 0) \right),$$
(15)

where the equality is due to (10). Dividing the RHS of (14) by (15) and multiplying by $p_0(X_0)$, we obtain the normalized distribution over the full path $\boldsymbol{X}$,

$$p^*(\boldsymbol{X}) = p^{\mathrm{base}}(\boldsymbol{X}) \exp \left( - \int_0^1 f(X_t, t) \, \mathrm{d}t - g(X_1) + V(X_0, 0) \right).$$
(16)

Setting $f = 0$ and $g = -r$, we arrive at an expression for the optimal distribution

$$p^*(X_0, X_1) = p^{\mathrm{base}}(X_0, X_1) \exp \left( r(X_1) + V(X_0, 0) \right).$$
(17)

This unfortunately does not lead to the tilted distribution (1) because we have a bias in the optimal distribution that is due to the value function of the initial distribution $V(X_0, 0)$. That is to say, naïvely adding a KL regularization (12) to the fine-tuning objective in the sense of (13) leads to a biased distribution (16) after fine-tuning and is *not* equivalent to the tilted distribution (1). For instance, when the sampling procedure is noiseless, *i.e.*, $\sigma(t) = 0$, fine-tuning naïvely will not have any effect because $X_0$ completely determines $X_1$.

This is unlike the situation for large language models (Ouyang et al., 2022; Rafailov et al., 2023), where there is no dynamical process that samples $X_1$ iteratively and hence no dependence on the initial noise variable $X_0$. Although this KL regularization is a common objective for RLHF of large language models, it has seen seldom use in fine-tuning diffusion models, likely due to this issue of the initial value function bias. In the context of diffusion models, KL regularization (13) has been explored in prior works (Fan et al., 2024), but its behavior was not well-understood and they did not relate the fine-tuned model to the tilted distribution (1). Another direction that has been proposed is to learn the initial distribution $p_0$ to cancel out the bias (Uehara et al., 2024b; Tang, 2024) but this simply shifts the work into tilting the initial distribution and requires an auxiliary model for parameterizing the optimal initial distribution. In contrast, we show in the next section that it is possible to remove the value function bias by simply choosing a very particular noise schedule during the fine-tuning procedure.

## 3.3 THE MEMORYLESS NOISE SCHEDULE TO FINE-TUNE DYNAMICAL GENERATIVE MODELS

In this subsection, we propose a very simple method of turning (17) into the tilted distribution (1) through the use of *memoryless* noise schedules. We provide an intuitive explanation of why such noise schedules are sufficient for fine-tuning, and show that if we want to sample the fine-tuned model with an arbitrary noise schedule, we must use a particular memoryless noise schedule.

Intuitively, the main reason we cannot arrive at the tilted distribution from (17) is due to the $p^{\mathrm{base}}(X_0, X_1)$ distribution not factoring into $X_0$ and $X_1$. Hence, we define a memoryless generative process as follows:

---

[4]Note (14) is informal because densities over continuous-time processes are ill-defined; the formal statement is $\frac{\mathrm{d}\mathbb{P}^*}{\mathrm{d}\mathbb{P}^{\mathrm{base}}}(\boldsymbol{X}|X_0) = \exp(- \int_0^1 f(X_t, t) \, \mathrm{d}t - g(X_1))$, where $\frac{\mathrm{d}\mathbb{P}^*}{\mathrm{d}\mathbb{P}^{\mathrm{base}}}$ denotes the Radon-Nikodym derivative. We treat this formally in the proofs.

| | $\kappa_t$ | $\eta_t$ | Diffusion coefficient $\sigma(t)$ | Memoryless $X_t$ |
|---|---|---|---|---|
| Flow Matching (3) | $\frac{\dot{\alpha}_t}{\alpha_t}$ | $\beta_t\left(\frac{\dot{\alpha}_t}{\alpha_t}\beta_t - \dot{\beta}_t\right)$ | General (commonly 0) | No |
| Memoryless Flow Matching (4) | $\frac{\dot{\alpha}_t}{\alpha_t}$ | $\beta_t\left(\frac{\dot{\alpha}_t}{\alpha_t}\beta_t - \dot{\beta}_t\right)$ | $\sqrt{2\eta_t}$ | Yes |
| DDIM (29) | $\frac{\dot{\bar{\alpha}}_t}{2\bar{\alpha}_t}$ | $\frac{\dot{\bar{\alpha}}_t}{2\bar{\alpha}_t}$ | General (commonly 0) | No |
| DDPM (30) | $\frac{\dot{\bar{\alpha}}_t}{2\bar{\alpha}_t}$ | $\frac{\dot{\bar{\alpha}}_t}{2\bar{\alpha}_t}$ | $\sqrt{2\eta_t}$ | Yes |

Table 1: Diffusion coefficient $\sigma(t)$ and the factors $\kappa_t$, $\eta_t$ for the Flow Matching, Memoryless Flow Matching, DDIM, and DDPM generative processes. When the diffusion coefficient is $\sigma(t) = \sqrt{2\eta_t}$, the generative process is memoryless, *i.e.*, samples $X_1$ will be independent of the initial noise $X_0$.

**Definition 1** (Memoryless generative process)**.** *A generative process of the form* (5)-(6) *is memoryless if $X_0$ and $X_1$ are independent, i.e., $p^{base}(X_0, X_1) = p^{base}(X_0)p^{base}(X_1)$.*

When the base generative process is memoryless, this implies:

$$p^*(X_1) = \int p^{\text{base}}(X_0)p^{\text{base}}(X_1)\exp(r(X_1) + V(X_0, 0))\mathrm{d}X_0 \propto p^{\text{base}}(X_1)\exp(r(X_1)). \quad (18)$$

That is, solving the SOC problem (7)-(8) with a memoryless base model will result in a fine-tuned model that generates samples $p^*(X_1)$ according to the tilted distribution (1). This memoryless property is not satisfied generally by the family of generative processes captured by (7)-(8). For instance, the Flow Matching and DDIM generative processes with zero diffusion coefficient (*i.e.*, $\sigma(t) = 0$) are definitely not memoryless due to $X_0$ and $X_1$ being theoretically invertible. Below, we provide the sufficient and necessary condition for the noise schedule in order to have a memoryless generative process.

**Proposition 1** (Memoryless noise schedules)**.** *Within the family of generative processes* (5)-(6)*, a generative process is memoryless if and only if the noise schedule is chosen as:*

$$\sigma(t)^2 = 2\eta_t + \chi(t), \text{ where } \chi : [0, 1] \to \mathbb{R} \text{ is s.t. } \forall t \in (0, 1], \lim_{t' \to 0^+} \alpha_{t'}\exp\left(-\int_{t'}^t \frac{\chi(s)}{2\beta_s^2}\mathrm{d}s\right) = 0,$$
$$(19)$$

*where $\eta_t$ is defined in* (6)*. In particular, we refer to $\sigma(t) = \sqrt{2\eta_t}$ as the memoryless noise schedule.*

Due to the endpoint constraints of $(\alpha_t, \beta_t)$ for the reference flow (2), the memoryless noise schedule $\sigma(t)$ is infinite at $t = 0$ and approaches zero at $t = 1$. This provides a way for the generative process to mix when close to noise $X_0$ while stay steadying when close to the sample $X_1$. Hence, the sample will have no information about $X_0$ due to the enormous amount of mixing with a large diffusion coefficient. Furthermore, while we have intuitively justified the memoryless noise schedule through its independence property, our theoretical result is actually even stronger: all generative models of the form (5)-(6) *must* be fine-tuned using the memoryless noise schedule. We formalize this in the following theorem, which we prove in App. E.2:

**Theorem 1** (Fine-tuning recipe for general noise schedule sampling)**.** *Within the family of generative processes* (5)-(6)*, in order to allow the use of arbitrary noise schedules and still generate samples according to the tilted distribution* (1)*, the fine-tuning problem* (7)-(8) *with $f = 0$ and $g = -r$ must be done with the memoryless noise schedule $\sigma(t) = \sqrt{2\eta_t}$.*

Thm. 1 states that we *need* to use the memoryless noise schedule for fine-tuning with the SOC objective—or equivalently, the KL regularized reward objective (13). This is the only noise schedule that retains the relationship between the velocity and score function, allowing the conversion to arbitrary noise schedules (*e.g.*, $\sigma(t) = 0$) after fine-tuning. It is worth noting that when using the memoryless noise schedule for DDIM, this recovers what we derived as the continuous-time limit of the DDPM generative process (30). However, the DDPM sampler (Ho et al., 2020) is not commonly used as the DDIM sampler (Song et al., 2021a) and Flow Matching models typically generate samples using $\sigma(t) = 0$, so an explicit conversion to the memoryless noise schedule is necessary for fine-tuning. Tab. 1 summarizes the memoryless schedule for diffusion and Flow Matching models, which we refer to as Memoryless Flow Matching. In Fig. 2, we visualize fine-tuning a 1D model, where we see that constant $\sigma(t)$ leads to biased distributions whereas the memoryless noise schedule perfectly converges to the tilted distribution (1). In App. E.2.1, we express the base drift $b$ and the control $u$ in terms of the base and fine-tuned Flow Matching vector fields $v^{\text{base}}$ and $v^{\text{finetune}}$, and do the same for DDIM.

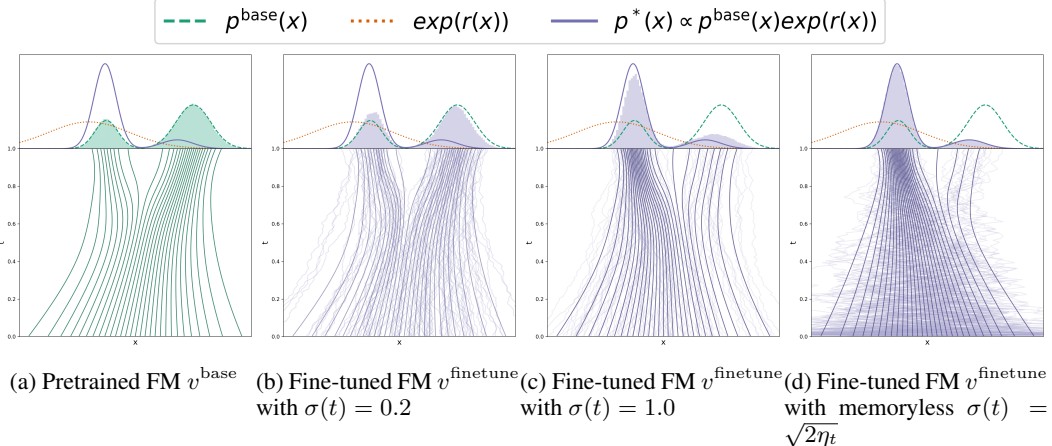

Figure 2: Visualization of Thm. 1 showing that fine-tuning must be done with the memoryless noise schedule to ensure convergence to the tilted distribution (1). (a) Shows the base Flow Matching model. (b, c) Fine-tuning using a constant $\sigma(t)$ leads to biased distributions. (d) Fine-tuning using the memoryless noise schedule leads to the correct tilted distribution. Note that sample generation can use any noise schedule after fine-tuning, including $\sigma(t) = 0$.

## 4 ADJOINT MATCHING FOR STOCHASTIC OPTIMAL CONTROL

Although several deep learning methods have been proposed to solve SOC problems using deep learning (Domingo-Enrich et al., 2023; Nüsken & Richter, 2021), the preferred approach is the adjoint method, which performs gradient-based optimization on the control objective (7) with respect to a parameterized control function. There are two approaches which yield the same gradient in the small step size limit (see App. F.1.1 for more detail): the *discrete adjoint method*, where the control objective is discretized and an automatic differentiation engine is used to backpropagate through it (Han & E, 2016), and the *continuous adjoint method*, where the continuous-time structure is exploited by solving the *adjoint ODE* backwards in time:

$$\tfrac{\mathrm{d}a}{\mathrm{d}t}(t; \boldsymbol{X}, u) = -\Big[a(t; \boldsymbol{X}, u)^{\mathsf{T}} \big(\nabla_{X_t}(b(X_t, t) + \sigma(t)u(X_t, t))\big) + \nabla_{X_t}\big(f(X_t, t) + \tfrac{1}{2}\|u(X_t, t)\|^2\big)\Big],$$
(20)

with initial condition $a(1; \boldsymbol{X}, u) = \nabla g(X_1)$. The gradient of the continuous adjoint loss is given by

$$\tfrac{\mathrm{d}\mathcal{L}}{\mathrm{d}\theta} = \tfrac{1}{2}\int_0^1 \tfrac{\partial}{\partial\theta}\|u(X_t, t)\|^2\mathrm{d}t + \int_0^1 \tfrac{\partial u(X_t, t)}{\partial\theta}^{\mathsf{T}}\sigma(t)^{\mathsf{T}}a(t; \boldsymbol{X}, u)\mathrm{d}t,$$
(21)

The following proposition introduces and studies a loss whose gradient is also (21).

**Proposition 2.** *Let us define, for now, the basic Adjoint Matching objective as:*

$$\mathcal{L}_{\text{Basic}-\text{Adj}-\text{Match}}(u; \boldsymbol{X}) := \tfrac{1}{2}\int_0^1 \big\|u(X_t, t) + \sigma(t)^{\mathsf{T}}a(t; \boldsymbol{X}, \bar{u})\big\|^2 \mathrm{d}t, \quad \boldsymbol{X} \sim p^{\bar{u}}, \quad \bar{u} = \mathit{stopgrad}(u),$$
(22)

*where $\bar{u} = \mathit{stopgrad}(u)$ means that the gradients of $\bar{u}$ with respect to the parameters $\theta$ of the control $u$ are artificially set to zero. The gradient of $\mathcal{L}_{\text{Basic}-\text{Adj}-\text{Match}}(u; \boldsymbol{X})$ with respect to $\theta$ is equal to the gradient $\tfrac{\mathrm{d}\mathcal{L}}{\mathrm{d}\theta}$ in equation (21). Importantly, the only critical point of $\mathbb{E}\big[\mathcal{L}_{\text{Basic}-\text{Adj}-\text{Match}}\big]$ is the optimal control $u^*$.*

Critical points of $\mathcal{L}$ are controls $u$ such that $\frac{\delta}{\delta u}\mathcal{L}(u) = 0$, where $\frac{\delta}{\delta u}\mathcal{L}$ denotes the first variation of the functional $\mathcal{L}$. In other words, Prop. 2 states that the only control that satisfies the first-order optimality condition for the basic Adjoint Matching objective is the optimal control, which provides theoretical grounding for gradient-based optimization algorithms. An intuitive way to understand the basic Adjoint Matching objective is that it is a *consistency loss*. The Adjoint Matching objective is based off of the observation that the optimal control $u^*(x, t)$ is the unique fixed-point of the relation $u(x, t) = -\sigma(t)^{\mathsf{T}}\nabla_x J(u; x, t)$ (see Lemma 6 in App. F.3) and so we are directly optimizing for a control that fits this relation, while using the adjoint state as a stochastic estimator of $\nabla_x J(u; x, t)$.

We can see that the basic Adjoint Matching objective produces the same gradient w.r.t. $\theta$ as the continuous adjoint method (21) by expanding the square in (22) and removing terms that do not depend on $\theta$. And while the basic Adjoint Matching is not entirely novel, it provides the means of deriving a simpler *leaner* objective function.

**The "Lean" Adjoint.** The minimizer of a least-squares objective is the conditional expectation of the regression target, so for the Adjoint Matching objective, at the optimum we have that

$$u^*(x,t) = \mathbb{E}_{\boldsymbol{X} \sim p^*}\left[-\sigma(t)^\mathsf{T} a(t; \boldsymbol{X}, u^*) | X_t = x\right]. \tag{23}$$

Multiplying both sides by the Jacobian $\nabla_x u^*(x,t)$ and re-arranging, we get the relation

$$\mathbb{E}_{\boldsymbol{X} \sim p^*}\left[u^*(x,t)^\mathsf{T} \nabla_x u^*(x,t) + a(t; \boldsymbol{X}, u^*)^\mathsf{T} \sigma(t) \nabla_x u^*(x,t) \mid X_t = x\right] = 0. \tag{24}$$

Notice that the terms inside the expectation in (24) show up as part of the adjoint ODE (20). Therefore, we motivate the definition of a *lean adjoint state* $\tilde{a}$ with the terms in (24) removed. Plugging this lean adjoint back into the least-squares objective, we obtain our final proposed Adjoint Matching objective:

$$\mathcal{L}_{\mathrm{Adj-Match}}(u; \boldsymbol{X}) := \tfrac{1}{2} \int_0^1 \left\| u(X_t, t) + \sigma(t)^\mathsf{T} \tilde{a}(t; \boldsymbol{X}) \right\|^2 \mathrm{d}t, \quad \boldsymbol{X} \sim p^{\bar{u}}, \ \bar{u} = \texttt{stopgrad}(u), \tag{25}$$

$$\text{where} \quad \tfrac{\mathrm{d}}{\mathrm{d}t}\tilde{a}(t; \boldsymbol{X}) = -(\tilde{a}(t; \boldsymbol{X})^\mathsf{T} \nabla_x b(X_t, t) + \nabla_x f(X_t, t)), \tag{26}$$

$$\tilde{a}(1; \boldsymbol{X}) = \nabla_x g(X_1). \tag{27}$$

Equations (26)-(27) define the *lean adjoint state*, and (25) is the complete Adjoint Matching objective. The unique critical point of $\mathbb{E}[\mathcal{L}_{\mathrm{Adj-Match}}]$ is the optimal control, which we prove relying on Prop. 2 and equation (24) (see Prop. 7 in App. F.4).

Compared to the adjoint method (App. F.1.1), Adjoint Matching produces a *different gradient in expectation than the continuous adjoint*. This is because the lean adjoint state is not related to the gradient of the cost functional anymore, *i.e.*, (171) is not true, except at the optimum when $u = u^*$. Even at the optimal solution, since Adjoint Matching removes terms that have expectation zero, it can potentially exhibit better convergence and lower variance than the continuous adjoint method. Additionally, computation of the lean adjoint state (26) also exhibits a smaller computational cost due to the removal of the extra terms (no longer need the Jacobian of the control $\nabla_x u$). We provide a rigorous derivation of Adjoint Matching and the above claims in App. F.4.

Adjoint Matching can be applied to reward fine-tuning of dynamical generative models through the memoryless SOC formulation discussed in Sec. 3. In App. F.5, we provide pseudo-code for Flow Matching models (Alg. 1) and for denoising diffusion models (Alg. 2).

## 5 EXPERIMENTS

We experimentally validate our proposed method on reward fine-tuning a Flow Matching base model (Lipman et al., 2023). In particular, we use the usual setup of pre-training an autoencoder for $512 \times 512$ resolution images, then training a text-conditional Flow Matching model on the latent variables with a U-net architecture (Long et al., 2015), similar to the setup in Rombach et al. (2022). We pre-trained our base model using a dataset of licensed text and image pairs. Then for fine-tuning, we consider the reward function: $r(x) := \lambda \times \texttt{RewardModel}(x)$ corresponding to a scaled version of the reward model, which we take to be ImageReward (Xu et al., 2023). Different values of $\lambda$ provide different tradeoffs between the KL regularization and the reward model (13). For evaluation and benchmarking purposes, we report metrics that separately quantify text-to-image consistency, human preference, and sample diversity, capturing the tradeoff between each aspect of generative models (Astolfi et al., 2024). For consistency, we make use of the standard ClipScore (Hessel et al., 2021) and PickScore (Kirstain et al., 2023); for generalization to unseen human preferences, we use the HPSv2 model (Wu et al., 2023b); and for diversity, we compute averages of pairwise distances of the DreamSim features (Fu et al., 2023). More details are provided in App. H.4. As our baselines, we consider the DPO (Wallace et al., 2023a), ReFL (Xu et al., 2023), and DRaFT-K algorithms (Clark et al., 2024). DPO does not use gradients from the reward function, while ReFL and DRaFT make use of heuristic gradient stopping approaches to stay close to the base generative model. Out

| | Fine-tuning Method | Fine-tuning $\sigma(t)$ | Sampling $\sigma(t)$ | ClipScore ↑ | PickScore ↑ | HPS v2 ↑ | DreamSim Diversity ↑ |
|---|---|---|---|---|---|---|---|
| | None (Base model) | N/A | $\sqrt{2\eta_t}$ | $24.15_{\pm 0.26}$ | $17.25_{\pm 0.06}$ | $16.19_{\pm 0.17}$ | $53.60_{\pm 1.37}$ |
| | | | 0 | $28.32_{\pm 0.22}$ | $18.15_{\pm 0.07}$ | $17.89_{\pm 0.16}$ | $\mathbf{56.53_{\pm 1.52}}$ |
| Baselines | DRaFT-1 | $\sqrt{2\eta_t}$ | $\sqrt{2\eta_t}$ | $30.18_{\pm 0.24}$ | $19.38_{\pm 0.08}$ | $24.61_{\pm 0.17}$ | $25.54_{\pm 0.99}$ |
| | | 0 | 0 | $30.95_{\pm 0.28}$ | $19.37_{\pm 0.06}$ | $24.37_{\pm 0.17}$ | $27.39_{\pm 1.14}$ |
| | DRaFT-40 | $\sqrt{2\eta_t}$ | $\sqrt{2\eta_t}$ | $26.94_{\pm 0.28}$ | $18.34_{\pm 0.19}$ | $19.98_{\pm 1.02}$ | $41.98_{\pm 2.14}$ |
| | | 0 | 0 | $30.07_{\pm 0.39}$ | $19.45_{\pm 0.08}$ | $24.06_{\pm 0.24}$ | $36.53_{\pm 1.69}$ |
| | DPO | $\sqrt{2\eta_t}$ | $\sqrt{2\eta_t}$ | $24.11_{\pm 0.22}$ | $17.24_{\pm 0.06}$ | $16.15_{\pm 0.14}$ | $53.27_{\pm 1.36}$ |
| | | 0 | 0 | $27.77_{\pm 0.18}$ | $17.92_{\pm 0.07}$ | $17.30_{\pm 0.20}$ | $54.11_{\pm 1.50}$ |
| | ReFL | $\sqrt{2\eta_t}$ | $\sqrt{2\eta_t}$ | $28.59_{\pm 0.31}$ | $18.68_{\pm 0.10}$ | $22.24_{\pm 0.46}$ | $32.71_{\pm 2.76}$ |
| | | 0 | 0 | $30.06_{\pm 0.63}$ | $19.07_{\pm 0.21}$ | $23.06_{\pm 0.41}$ | $32.69_{\pm 1.28}$ |
| Memoryless SOC | Cont. Adjoint $\lambda = 12500$ | $\sqrt{2\eta_t}$ | $\sqrt{2\eta_t}$ | $26.99_{\pm 0.43}$ | $18.33_{\pm 0.16}$ | $20.83_{\pm 0.63}$ | $46.59_{\pm 1.40}$ |
| | | | 0 | $29.49_{\pm 0.32}$ | $18.98_{\pm 0.16}$ | $21.34_{\pm 0.53}$ | $48.41_{\pm 1.44}$ |
| | Disc. Adjoint $\lambda = 12500$ | $\sqrt{2\eta_t}$ | $\sqrt{2\eta_t}$ | $28.04_{\pm 0.57}$ | $18.44_{\pm 0.21}$ | $20.04_{\pm 0.39}$ | $54.90_{\pm 2.03}$ |
| | | | 0 | $29.28_{\pm 0.17}$ | $18.82_{\pm 0.14}$ | $19.73_{\pm 0.17}$ | $53.36_{\pm 2.48}$ |
| | Adj.-Matching $\lambda = 1000$ | $\sqrt{2\eta_t}$ | $\sqrt{2\eta_t}$ | $30.36_{\pm 0.22}$ | $19.29_{\pm 0.08}$ | $24.12_{\pm 0.17}$ | $40.89_{\pm 1.50}$ |
| | | | 0 | $31.41_{\pm 0.22}$ | $19.57_{\pm 0.09}$ | $23.29_{\pm 0.18}$ | $43.10_{\pm 1.76}$ |
| | Adj.-Matching $\lambda = 2500$ | $\sqrt{2\eta_t}$ | $\sqrt{2\eta_t}$ | $30.59_{\pm 0.40}$ | $19.49_{\pm 0.10}$ | $24.85_{\pm 0.23}$ | $37.07_{\pm 1.47}$ |
| | | | 0 | $31.64_{\pm 0.21}$ | $19.71_{\pm 0.09}$ | $24.12_{\pm 0.27}$ | $39.88_{\pm 1.59}$ |
| | Adj.-Matching $\lambda = 12500$ | $\sqrt{2\eta_t}$ | $\sqrt{2\eta_t}$ | $30.62_{\pm 0.30}$ | $19.50_{\pm 0.09}$ | $\mathbf{24.95_{\pm 0.28}}$ | $34.50_{\pm 1.33}$ |
| | | | 0 | $\mathbf{31.65_{\pm 0.19}}$ | $\mathbf{19.76_{\pm 0.08}}$ | $24.49_{\pm 0.27}$ | $37.24_{\pm 1.57}$ |

Table 2: Evaluation metrics of different fine-tuning methods for text-to-image generation. The second and third columns show the noise schedules $\sigma(t)$ used for fine-tuning and for sampling: $\sigma(t) = \sqrt{2\eta_t}$ corresponds to Memoryless Flow Matching, and $\sigma(t) = 0$ to the Flow Matching ODE (3). We report standard errors estimated over 3 runs of the fine-tuning algorithm on random sets of 40000 training prompts, each evaluated over a random set of 1000 test prompts.

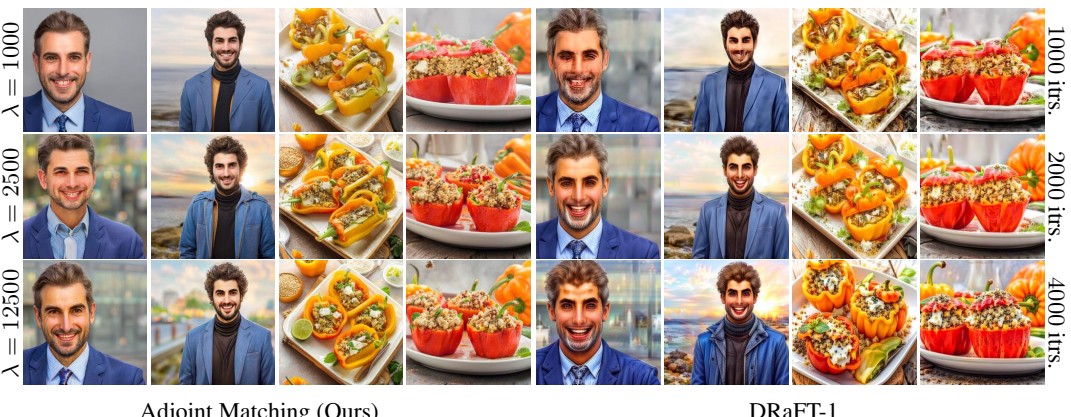

Figure 3: Our proposed Adjoint Matching using the memoryless SOC formulation introduces a much more principled way of trading off how close to stay to the base model while optimizing the reward model. In contrast, baseline methods such as DRaFT-1 only optimize the reward model and must rely on early stopping to perform this trade off, resulting in a much more sensitive hyperparameter. Samples are produced using $\sigma(t) = 0$ with the same noise sample. Text prompts: "*Handsome Smiling man in blue jacket portrait*" and "*Quinoa and Feta Stuffed Baby Bell Peppers*".

of these baseline methods, we find that DRaFT-1 performs the best, so we perform additional ablation experiments comparing to this method. Within the same SOC formulation, we also consider the discrete and continuous adjoint methods. We provide full experimental details in App. H; an important implementation detail is that we slightly offset $\sigma(t)$ in order to avoid division by zero.

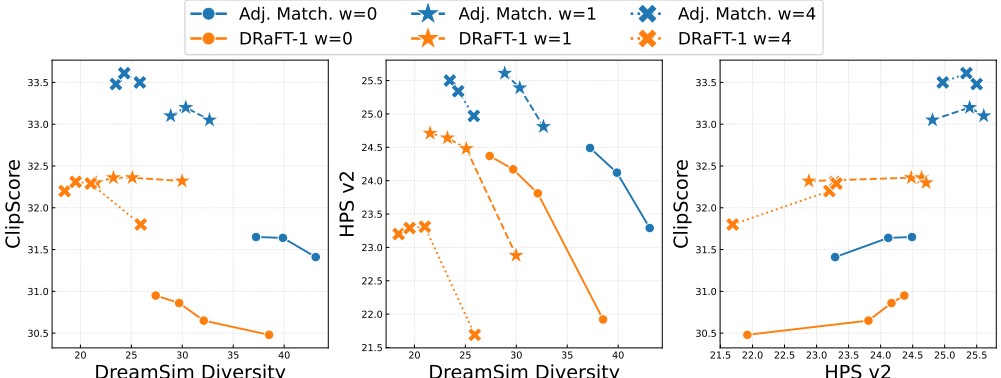

Figure 4: Tradeoffs between different aspects of generative models: text-to-image consistency (Clip-Score), sample diversity for each prompt (DreamSim Diversity), and generalization to unseen human preferences (HPS v2). Different points are obtained from varying values of $\lambda$ for Adjoint Matching and varying number of fine-tuning iterations for the DRaFT-1 baseline. Overall, we find our proposed method Adjoint Matching has the best Pareto fronts.

**Evaluation results.** In Tab. 2 we report the evaluation metrics for the baselines as well as our proposed Adjoint Matching approach. We compare each method at roughly the same wall clock time (see the times and number of iterations in Tab. 3, and comments in App. H.5). We find that across all metrics, our proposed memoryless SOC formulation outperforms existing baseline methods. The choice of SOC algorithms also obviously favors Adjoint Matching over continuous and discrete adjoint methods, which result in poorer consistency and human preference metrics.

**Ablation: base model vs. reward tradeoff.** We note that the scaling in front of the reward model $\lambda$ determines how strongly the we should prefer the reward model over the base model. As such, we see a natural tradeoff curve: higher $\lambda$ results in better consistency and human preference, but lower diversity in the generated samples. Overall, we find that Adjoint Matching performs stably across all values of $\lambda$. Our method of regularizing the fine-tuning procedure through memoryless SOC works much better than baseline methods which often must employ early stopping. We show the qualitative effect of varying $\lambda$ in Fig. 3, while for the DRaFT-1 baseline we show the effect of varying the number of fine-tuning iterations.

**Ablation: classifier-free guidance.** We note that it is possible to apply classifier-free guidance (CFG; Ho & Salimans (2022); Zheng et al. (2023)) after fine-tuning. We use the formula $(1 + w)v(x, t|y) - wv(x, t)$, where $w$ is the guidance weight, $v(x, t|y)$ is a fine-tuned text-to-image model while $v(x, t)$ is an unconditional image model. This is not principled as only the conditional model is fine-tuned, but generally it is unclear what distribution guided models sample from anyhow. In Fig. 4 we show the evaluation metrics with classifier-free guidance applied. Comparing three different guidance weight values, we see a higher weight does improve text-to-image consistency, and to some extent, human preference, but this comes at the cost of being worse in terms of diversity. We show qualitative differences in Fig. 6 (App. A).

# 6 CONCLUSION

We investigate the problem of fine-tuning dynamical generative models such as Flow Matching and propose the use of a stochastic optimal control (SOC) formulation with a memoryless noise schedule. This ensures we converge to the same tilted distribution that the large language modeling literature uses for learning from human feedback. In particular, the memoryless noise schedule corresponds to DDPM sampling for diffusion models and a new Memoryless Flow Matching generative process for flow models. In conjunction, we propose a novel training algorithm for solving stochastic optimal control problems, by casting SOC as a regression problem, which we call the Adjoint Matching objective. Empirically, we find that our memoryless SOC formulation works better than multiple existing works on fine-tuning diffusion models, and our Adjoint Matching algorithm outperforms related gradient-based methods. In summary, we are the first to provide a theoretically-driven algorithm for fine-tuning Flow Matching models, and we find that our approach significantly outperforms baseline methods across multiple axes of evaluation—text-to-image consistency, generalization to unseen human preference, and sample diversity—on large-scale text-to-image generation.

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

# Contents

# A    ADDITIONAL FIGURES & TABLES

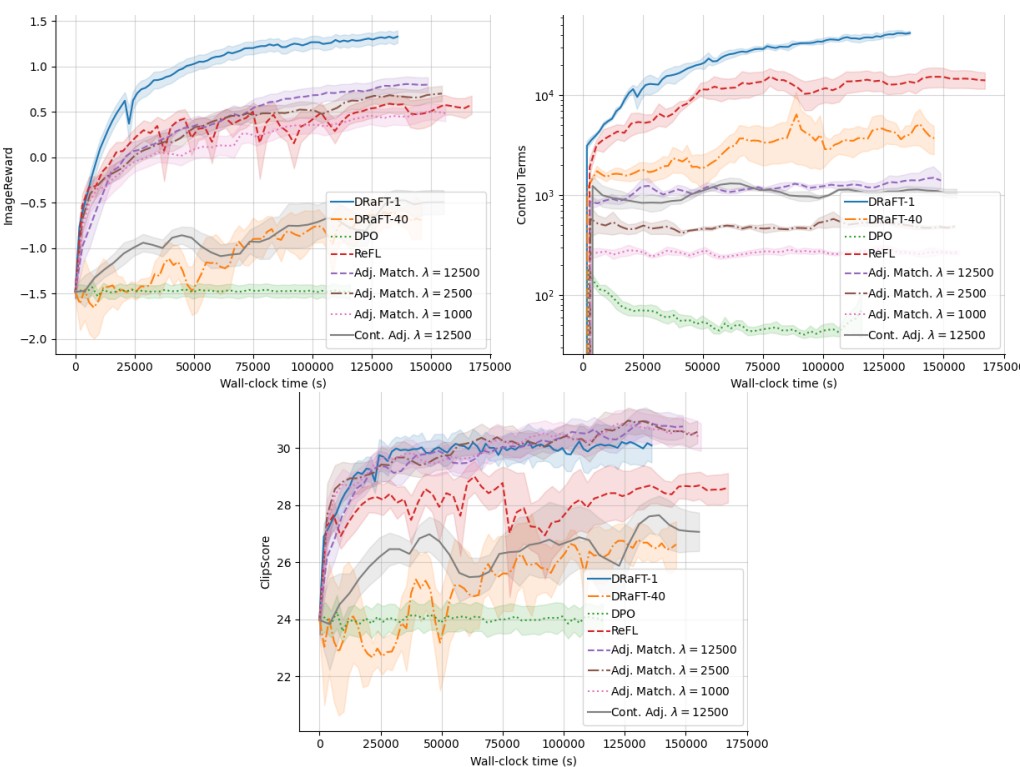

Figure 5: Average values of ImageReward (reward function), control cost ($\int_0^t \frac{1}{2}\|u(X_t^u, t)\|^2 \, \mathrm{d}t$), and ClipScore vs. wall-clock time for Adjoint Matching and our baselines. Lines show averages over three fine-tuning runs, evaluating on separate test datasets of size 200. Confidence intervals show standard errors of estimates.

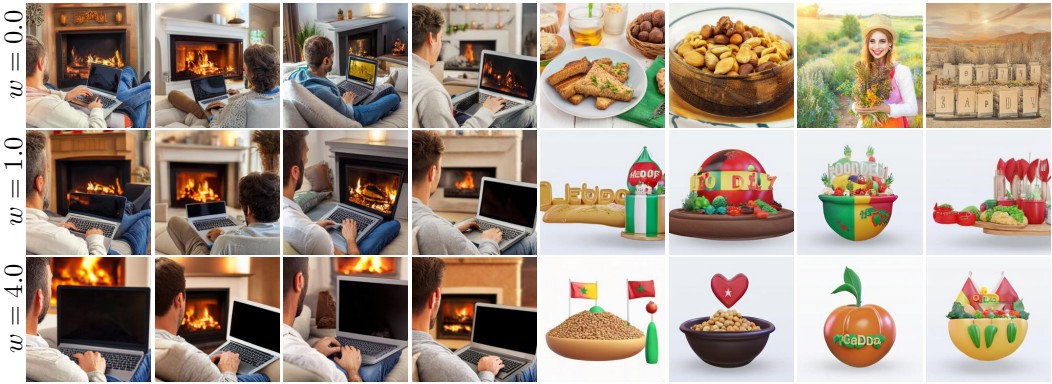

Text prompt: "*Man sitting on sofa at home in front of fireplace and using laptop computer, rear view*"

Text prompt: "*3D World Food Day Morocco*"

Figure 6: Generated samples from varying classifier-free guidance weight $w$, from an Adjoint Matching fine-tuned model. Higher guidance increases text-to-image consistency but loses diversity and has use cases for generating highly structured images such as 3D renderings. Corresponding samples from the base model can be found in Fig. 7.

| Fine-tuning loss | Fine-tuning $\sigma(t)$ | Sampling $\sigma(t)$ | ImageReward ↑ | ClipScore diversity ↑ | PickScore diversity ↑ | Total time (s) / # iterations |
|---|---|---|---|---|---|---|
| None (CFG = 1.0) | N/A | $\sqrt{2\eta_t}$ | $-1.384_{\pm 0.040}$ | $28.07_{\pm 1.40}$ | $1.63_{\pm 0.08}$ | N/A |
|  |  | $0$ | $-0.920_{\pm 0.042}$ | $30.29_{\pm 1.53}$ | $1.82_{\pm 0.09}$ |  |
| DRaFT-1 | $\sqrt{2\eta_t}$ | $\sqrt{2\eta_t}$ | $1.357_{\pm 0.039}$ | $16.86_{\pm 0.98}$ | $1.21_{\pm 0.07}$ | $140k_{\pm 5.9k}$ |
|  | $0$ | $0$ | $1.251_{\pm 0.040}$ | $16.76_{\pm 1.06}$ | $1.27_{\pm 0.07}$ | / 4000 |
| DRaFT-40 | $\sqrt{2\eta_t}$ | $\sqrt{2\eta_t}$ | $-0.560_{\pm 0.138}$ | $24.07_{\pm 1.37}$ | $1.64_{\pm 0.12}$ | $148k_{\pm 4.2k}$ |
|  | $0$ | $0$ | $0.424_{\pm 0.042}$ | $20.99_{\pm 1.54}$ | $1.67_{\pm 0.08}$ | / 1500 |
| DPO | $\sqrt{2\eta_t}$ | $\sqrt{2\eta_t}$ | $-1.386_{\pm 0.033}$ | $27.80_{\pm 1.40}$ | $1.62_{\pm 0.08}$ | $118k_{\pm 0.6k}$ |
|  | $0$ | $0$ | $-0.957_{\pm 0.040}$ | $29.81_{\pm 1.43}$ | $1.68_{\pm 0.10}$ | / 1000 |
| ReFL | $\sqrt{2\eta_t}$ | $\sqrt{2\eta_t}$ | $0.687_{\pm 0.085}$ | $19.49_{\pm 1.76}$ | $1.22_{\pm 0.08}$ | $173k_{\pm 10.9k}$ |
|  | $0$ | $0$ | $0.709_{\pm 0.080}$ | $18.39_{\pm 1.11}$ | $1.31_{\pm 0.10}$ | / 6000 |
| Cont. Adjoint $\lambda = 12500$ | $\sqrt{2\eta_t}$ | $\sqrt{2\eta_t}$ | $-0.448_{\pm 0.135}$ | $26.97_{\pm 1.37}$ | $1.82_{\pm 0.09}$ | $153k_{\pm 0.9k}$ |
|  |  | $0$ | $-0.249_{\pm 0.116}$ | $26.25_{\pm 1.30}$ | $1.90_{\pm 0.10}$ | / 750 |
| Disc. Adjoint $\lambda = 12500$ | $\sqrt{2\eta_t}$ | $\sqrt{2\eta_t}$ | $-0.557_{\pm 0.113}$ | $30.40_{\pm 2.39}$ | $1.91_{\pm 0.09}$ | $152k_{\pm 1.5k}$ |
|  |  | $0$ | $-0.552_{\pm 0.041}$ | $28.37_{\pm 2.26}$ | $1.97_{\pm 0.09}$ | / 1000 |
| Adj.-Matching $\lambda = 1000$ | $\sqrt{2\eta_t}$ | $\sqrt{2\eta_t}$ | $0.550_{\pm 0.043}$ | $23.00_{\pm 1.27}$ | $1.65_{\pm 0.08}$ |  |
|  |  | $0$ | $0.454_{\pm 0.055}$ | $22.76_{\pm 1.40}$ | $1.73_{\pm 0.09}$ |  |
| Adj.-Matching $\lambda = 2500$ | $\sqrt{2\eta_t}$ | $\sqrt{2\eta_t}$ | $0.755_{\pm 0.040}$ | $21.33_{\pm 1.71}$ | $1.55_{\pm 0.08}$ | $156k_{\pm 1.9k}$ |
|  |  | $0$ | $0.671_{\pm 0.047}$ | $21.42_{\pm 1.54}$ | $1.64_{\pm 0.08}$ | / 1000 |
| Adj.-Matching $\lambda = 12500$ | $\sqrt{2\eta_t}$ | $\sqrt{2\eta_t}$ | $0.882_{\pm 0.058}$ | $20.49_{\pm 1.48}$ | $1.50_{\pm 0.09}$ |  |
|  |  | $0$ | $0.778_{\pm 0.050}$ | $20.34_{\pm 1.49}$ | $1.57_{\pm 0.09}$ |  |

Table 3: Metrics for various fine-tuning methods for text-to-image generation. The second and third columns show the noise schedules $\sigma(t)$ used for fine-tuning and for inference: $\sigma(t) = \sqrt{2\eta_t}$ corresponds to Memoryless Flow Matching, and $\sigma(t) = 0$ to the Flow Matching ODE (3). Confidence intervals show standard errors of estimates; computed over 3 runs of the fine-tuning algorithm on separate fine-tuning prompt datasets of size 40000 each. Test prompt sets are of size 1000, and also different for each run.

| Fine-tun. loss | Fine-tun. $\sigma(t)$ | Generat. $\sigma(t)$ | ImageReward ↑ | ClipScore ↑ | PickScore ↑ | HPS v2 ↑ | DreamSim diversity ↑ | Runtime/ #iter. |
|---|---|---|---|---|---|---|---|---|
| ReFL | $\sqrt{2\eta_t}$ | $\sqrt{2\eta_t}$ | $0.459_{\pm 0.096}$ | $28.46_{\pm 0.25}$ | $18.77_{\pm 0.09}$ | $22.54_{\pm 0.17}$ | $37.51_{\pm 3.50}$ | $43k_{\pm 2.7k}$ |
|  | $0$ | $0$ | $0.330_{\pm 0.114}$ | $29.63_{\pm 0.61}$ | $19.08_{\pm 0.18}$ | $22.46_{\pm 0.77}$ | $39.51_{\pm 1.30}$ | / 1500 |
| DRaFT-1 | $\sqrt{2\eta_t}$ | $\sqrt{2\eta_t}$ | $0.913_{\pm 0.068}$ | $29.80_{\pm 0.22}$ | $19.16_{\pm 0.06}$ | $23.63_{\pm 0.16}$ | $35.21_{\pm 1.93}$ | $35k_{\pm 1.5k}$ |
|  | $0$ | $0$ | $0.626_{\pm 0.195}$ | $30.48_{\pm 0.32}$ | $18.91_{\pm 0.34}$ | $21.92_{\pm 1.63}$ | $38.52_{\pm 2.01}$ | / 1000 |
| Draft-40 | $\sqrt{2\eta_t}$ | $\sqrt{2\eta_t}$ | $-1.427_{\pm 0.267}$ | $23.39_{\pm 1.72}$ | $17.24_{\pm 0.45}$ | $15.72_{\pm 1.80}$ | $41.98_{\pm 2.14}$ | $49k_{\pm 1.4k}$ |
|  | $0$ | $0$ | $-0.097_{\pm 0.052}$ | $29.12_{\pm 0.41}$ | $18.97_{\pm 0.14}$ | $21.93_{\pm 0.20}$ | $46.35_{\pm 1.34}$ | / 500 |
| Adj.-Match. $\lambda = 1000$ | $\sqrt{2\eta_t}$ | $\sqrt{2\eta_t}$ | $0.107_{\pm 0.046}$ | $29.37_{\pm 0.25}$ | $19.05_{\pm 0.07}$ | $22.79_{\pm 0.20}$ | $46.38_{\pm 1.36}$ |  |
|  |  | $0$ | $0.051_{\pm 0.044}$ | $30.58_{\pm 0.17}$ | $19.31_{\pm 0.07}$ | $21.93_{\pm 0.23}$ | $48.12_{\pm 1.56}$ |  |
| Adj.-Match. $\lambda = 2500$ | $\sqrt{2\eta_t}$ | $\sqrt{2\eta_t}$ | $0.199_{\pm 0.068}$ | $29.27_{\pm 0.21}$ | $19.07_{\pm 0.10}$ | $22.98_{\pm 0.30}$ | $45.03_{\pm 1.61}$ | $39k_{\pm 0.5k}$ |
|  |  | $0$ | $0.106_{\pm 0.067}$ | $30.43_{\pm 0.24}$ | $19.32_{\pm 0.11}$ | $22.16_{\pm 0.33}$ | $47.61_{\pm 1.49}$ | / 250 |
| Adj.-Match. $\lambda = 12500$ | $\sqrt{2\eta_t}$ | $\sqrt{2\eta_t}$ | $0.299_{\pm 0.095}$ | $29.61_{\pm 0.37}$ | $19.26_{\pm 0.14}$ | $23.67_{\pm 0.27}$ | $43.36_{\pm 1.93}$ |  |
|  |  | $0$ | $0.224_{\pm 0.051}$ | $30.70_{\pm 0.23}$ | $19.52_{\pm 0.11}$ | $22.93_{\pm 0.21}$ | $44.62_{\pm 1.79}$ |  |
| Cont. Adj. $\lambda = 12500$ | $\sqrt{2\eta_t}$ | $\sqrt{2\eta_t}$ | $-0.910_{\pm 0.116}$ | $26.29_{\pm 0.44}$ | $18.06_{\pm 0.16}$ | $18.86_{\pm 0.88}$ | $51.60_{\pm 1.97}$ | $51k_{\pm 0.3k}$ |
|  |  | $0$ | $-0.681_{\pm 0.051}$ | $28.50_{\pm 0.19}$ | $18.69_{\pm 0.11}$ | $19.90_{\pm 0.50}$ | $50.87_{\pm 1.52}$ | / 250 |
| Disc. Adj. $\lambda = 12500$ | $\sqrt{2\eta_t}$ | $\sqrt{2\eta_t}$ | $-0.978_{\pm 0.123}$ | $26.68_{\pm 0.76}$ | $18.51_{\pm 0.11}$ | $18.53_{\pm 0.28}$ | $55.95_{\pm 1.70}$ | $38k_{\pm 0.4k}$ |
|  |  | $0$ | $-0.791_{\pm 0.065}$ | $28.66_{\pm 0.33}$ | $18.51_{\pm 0.11}$ | $18.53_{\pm 0.28}$ | $54.78_{\pm 2.00}$ | / 250 |

Table 4: Additional metrics for various fine-tuning methods for text-to-image generation, which complement the ones in Tab. 2 (both tables correspond to the same runs). The second and third columns show the noise schedules $\sigma(t)$ used for fine-tuning and for inference: $\sigma(t) = \sqrt{2\eta_t}$ corresponds to Memoryless Flow Matching, and $\sigma(t) = 0$ to the Flow Matching ODE (3).

| $w$ | Fine-tuning loss | #iter. / $\lambda$ | Fine-tun. $\sigma(t)$ | Sampl. $\sigma(t)$ | ImageReward ↑ | ClipScore ↑ | PickScore ↑ | HPS v2 ↑ | DreamSim diversity ↑ |
|---|---|---|---|---|---|---|---|---|---|
| 0.0 | None | N/A | N/A | $\sqrt{2\eta_t}$ | $-1.384\pm0.040$ | $24.15\pm0.26$ | $17.25\pm0.06$ | $16.19\pm0.17$ | $53.60\pm1.37$ |
|  |  |  |  | 0 | $-0.920\pm0.042$ | $28.32\pm0.22$ | $18.15\pm0.07$ | $17.89\pm0.16$ | $\mathbf{56.53\pm1.52}$ |
| 0.0 | DRaFT-1 | 1000 | $\sqrt{2\eta_t}$ | $\sqrt{2\eta_t}$ | $0.913\pm0.068$ | $29.80\pm0.22$ | $19.16\pm0.06$ | $23.63\pm0.16$ | $35.21\pm1.93$ |
|  |  |  | 0 | 0 | $0.626\pm0.195$ | $30.48\pm0.32$ | $18.91\pm0.34$ | $21.92\pm1.63$ | $38.52\pm2.01$ |
|  |  | 2000 | $\sqrt{2\eta_t}$ | $\sqrt{2\eta_t}$ | $1.204\pm0.046$ | $29.90\pm0.43$ | $19.29\pm0.12$ | $24.40\pm0.27$ | $28.51\pm1.68$ |
|  |  |  | 0 | 0 | $1.052\pm0.088$ | $30.65\pm0.24$ | $19.27\pm0.11$ | $23.81\pm0.44$ | $32.11\pm2.37$ |
|  |  | 3000 | $\sqrt{2\eta_t}$ | $\sqrt{2\eta_t}$ | $\mathbf{1.307\pm0.041}$ | $29.96\pm0.22$ | $19.31\pm0.06$ | $24.42\pm0.13$ | $26.57\pm1.32$ |
|  |  |  | 0 | 0 | $1.173\pm0.058$ | $30.86\pm0.25$ | $19.37\pm0.06$ | $24.17\pm0.23$ | $29.69\pm1.30$ |
|  |  | 4000 | $\sqrt{2\eta_t}$ | $\sqrt{2\eta_t}$ | $\mathbf{1.357\pm0.039}$ | $30.18\pm0.24$ | $19.38\pm0.08$ | $24.61\pm0.17$ | $25.54\pm0.99$ |
|  |  |  | 0 | 0 | $1.251\pm0.040$ | $30.95\pm0.28$ | $19.37\pm0.06$ | $24.37\pm0.17$ | $27.39\pm1.14$ |
| 0.0 | Adj.-Match. | 1000 | $\sqrt{2\eta_t}$ | $\sqrt{2\eta_t}$ | $0.550\pm0.043$ | $30.36\pm0.22$ | $19.29\pm0.08$ | $24.12\pm0.17$ | $40.89\pm1.50$ |
|  |  |  | 0 | 0 | $0.454\pm0.055$ | $31.41\pm0.22$ | $19.57\pm0.09$ | $23.29\pm0.18$ | $43.10\pm1.76$ |
|  |  | 2500 | $\sqrt{2\eta_t}$ | $\sqrt{2\eta_t}$ | $0.755\pm0.040$ | $30.59\pm0.40$ | $19.49\pm0.10$ | $24.85\pm0.23$ | $37.07\pm1.47$ |
|  |  |  | 0 | 0 | $0.671\pm0.047$ | $31.64\pm0.21$ | $19.71\pm0.09$ | $24.12\pm0.27$ | $39.88\pm1.59$ |
|  |  | 12500 | $\sqrt{2\eta_t}$ | $\sqrt{2\eta_t}$ | $0.882\pm0.058$ | $30.62\pm0.30$ | $19.50\pm0.09$ | $24.95\pm0.28$ | $34.50\pm1.33$ |
|  |  |  | 0 | 0 | $0.778\pm0.050$ | $31.65\pm0.19$ | $19.76\pm0.08$ | $24.49\pm0.27$ | $37.24\pm1.57$ |
| 1.0 | None | N/A | N/A | $\sqrt{2\eta_t}$ | $-0.269\pm0.050$ | $30.41\pm0.22$ | $18.74\pm0.07$ | $20.47\pm0.18$ | $43.82\pm1.24$ |
|  |  |  |  | 0 | $-0.123\pm0.041$ | $31.83\pm0.17$ | $19.28\pm0.07$ | $20.95\pm0.16$ | $42.59\pm1.23$ |
| 1.0 | DRaFT-1 | 1000 | $\sqrt{2\eta_t}$ | $\sqrt{2\eta_t}$ | $1.123\pm0.051$ | $32.06\pm0.19$ | $19.69\pm0.06$ | $24.56\pm0.17$ | $28.25\pm1.55$ |
|  |  |  | 0 | 0 | $0.856\pm0.167$ | $32.32\pm0.25$ | $19.38\pm0.34$ | $22.88\pm1.54$ | $29.98\pm1.86$ |
|  |  | 2000 | 0 | 0 | $1.177\pm0.053$ | $32.36\pm0.18$ | $19.67\pm0.08$ | $24.48\pm0.28$ | $25.09\pm1.82$ |
|  |  | 3000 | 0 | 0 | $1.255\pm0.038$ | $32.36\pm0.19$ | $19.70\pm0.06$ | $24.64\pm0.17$ | $23.24\pm1.19$ |
|  |  | 4000 | 0 | 0 | $\mathbf{1.296\pm0.033}$ | $32.30\pm0.19$ | $19.68\pm0.06$ | $24.71\pm0.14$ | $21.54\pm0.96$ |
| 1.0 | Adj.-Match. | 1000 | 0 | 0 | $0.782\pm0.044$ | $33.05\pm0.22$ | $20.20\pm0.09$ | $24.81\pm0.18$ | $32.67\pm1.26$ |
|  |  | 2500 | $\sqrt{2\eta_t}$ | $\sqrt{2\eta_t}$ | $1.027\pm0.038$ | $32.85\pm0.21$ | $20.08\pm0.08$ | $\mathbf{25.88\pm0.20}$ | $29.83\pm1.00$ |
|  |  |  | 0 | 0 | $0.910\pm0.040$ | $33.20\pm0.17$ | $20.29\pm0.09$ | $25.39\pm0.24$ | $30.34\pm1.51$ |
|  |  | 12500 | 0 | 0 | $0.985\pm0.041$ | $33.10\pm0.18$ | $20.28\pm0.08$ | $\mathbf{25.61\pm0.27}$ | $28.86\pm1.37$ |
| 4.0 | None | N/A | N/A | $\sqrt{2\eta_t}$ | $0.277\pm0.043$ | $32.68\pm0.18$ | $19.50\pm0.07$ | $22.29\pm0.16$ | $35.12\pm0.92$ |
|  |  |  |  | 0 | $0.209\pm0.046$ | $32.83\pm0.17$ | $19.79\pm0.07$ | $22.30\pm0.17$ | $32.05\pm1.05$ |
| 4.0 | DRaFT-1 | 1000 | $\sqrt{2\eta_t}$ | $\sqrt{2\eta_t}$ | $1.062\pm0.045$ | $32.29\pm0.16$ | $19.48\pm0.06$ | $23.67\pm0.13$ | $25.03\pm1.32$ |
|  |  |  | 0 | 0 | $0.604\pm0.395$ | $31.80\pm0.86$ | $19.09\pm0.53$ | $21.69\pm2.10$ | $25.92\pm2.57$ |
|  |  | 2000 | 0 | 0 | $1.112\pm0.046$ | $32.29\pm0.20$ | $19.34\pm0.11$ | $23.31\pm0.22$ | $21.02\pm1.67$ |
|  |  | 3000 | 0 | 0 | $1.151\pm0.036$ | $32.31\pm0.21$ | $19.36\pm0.06$ | $23.29\pm0.14$ | $19.53\pm1.24$ |
|  |  | 4000 | 0 | 0 | $1.172\pm0.040$ | $32.20\pm0.22$ | $19.30\pm0.07$ | $23.20\pm0.15$ | $18.45\pm1.06$ |
| 4.0 | Adj.-Match. | 1000 | 0 | 0 | $0.852\pm0.046$ | $\mathbf{33.50\pm0.22}$ | $20.31\pm0.08$ | $24.97\pm0.19$ | $25.83\pm0.82$ |
|  |  | 2500 | $\sqrt{2\eta_t}$ | $\sqrt{2\eta_t}$ | $1.052\pm0.039$ | $\mathbf{33.51\pm0.19}$ | $20.15\pm0.07$ | $\mathbf{25.56\pm0.18}$ | $26.21\pm0.73$ |
|  |  |  | 0 | 0 | $0.942\pm0.042$ | $\mathbf{33.61\pm0.19}$ | $\mathbf{20.35\pm0.08}$ | $25.34\pm0.21$ | $24.30\pm0.86$ |
|  |  | 12500 | 0 | 0 | $1.007\pm0.052$ | $\mathbf{33.48\pm0.20}$ | $20.29\pm0.08$ | $\mathbf{25.50\pm0.29}$ | $23.48\pm0.81$ |

Table 5: Evaluation metrics when using classifier-free guidance (CFG; Ho & Salimans (2022)).

| LR / Adam $\beta_1$ | Fine-tuning loss | Fine-tun. $\sigma(t)$ | Generat. $\sigma(t)$ | ImageReward ↑ | ClipScore ↑ | PickScore ↑ | HPS v2 ↑ | DreamSim diversity ↑ |
|---|---|---|---|---|---|---|---|---|
| $3\times10^{-5}$ / 0.97 | DRaFT-1 | $\sqrt{2\eta_t}$ | $\sqrt{2\eta_t}$ | $1.467\pm0.029$ | $30.28\pm0.56$ | $19.37\pm0.09$ | $24.70\pm0.15$ | $21.20\pm0.93$ |
|  | Adj.-Match. $\lambda=12500$ | $\sqrt{2\eta_t}$ | $\sqrt{2\eta_t}$ | $1.130\pm0.034$ | $31.01\pm0.27$ | $19.60\pm0.08$ | $25.01\pm0.25$ | $26.73\pm0.88$ |
| $2\times10^{-5}$ / 0.95 | Disc. Adj. $\lambda=12500$ | $\sqrt{2\eta_t}$ | $\sqrt{2\eta_t}$ | $-1.186\pm0.553$ | $21.95\pm4.29$ | $16.94\pm0.95$ | $12.34\pm4.40$ | $28.33\pm10.26$ |
|  |  | 0 | 0 | $-0.961\pm0.653$ | $24.07\pm4.71$ | $17.86\pm1.17$ | $15.93\pm5.80$ | $33.62\pm7.80$ |

Table 6: Metrics for alternative optimization hyperparameters (learning rate and Adam $\beta_1$).

| Fine-tuning loss | Fine-tuning $\sigma(t)$ | Generative $\sigma(t)$ | ImageReward↑ | ClipScore↑ | PickScore↑ | HPS v2↑ | DreamSim diversity↑ |
|---|---|---|---|---|---|---|---|
| Adj.-Matching $\lambda = 12500$ | 1 | 1 | 0.009±0.077 | 29.18±0.51 | 18.66±0.09 | 20.75±0.32 | 41.33±1.24 |
| | | 0 | 0.454±0.055 | 31.41±0.22 | 19.57±0.09 | 23.29±0.18 | 43.10±1.76 |
| Adj.-Matching $\lambda = 12500$ | $\sqrt{2\eta_t}$ | $\sqrt{2\eta_t}$ | 0.882±0.058 | 30.62±0.30 | 19.50±0.09 | 24.95±0.28 | 34.50±1.33 |
| | | 0 | 0.778±0.050 | 31.65±0.19 | 19.76±0.08 | 24.49±0.27 | 37.24±1.57 |

Table 7: Comparison with an alternative fine-tuning noise schedule $\sigma(t) = 1$. We see that the initial value function bias (Subsec. 3.2) results in the model not having a high reward function (ImageReward is the reward function used for fine-tuning). Its performance on other metrics are also lower than when fine-tuning with the memoryless noise schedule, except for diversity.

| #sampl. timesteps | Fine-tuning loss | Fine-tun. $\sigma(t)$ | Sampl. $\sigma(t)$ | ImageReward↑ | ClipScore↑ | PickScore↑ | HPS v2↑ | DreamSim diversity↑ |
|---|---|---|---|---|---|---|---|---|
| 10 | None (Base) | N/A | $\sqrt{2\eta_t}$ | −2.279±0.001 | 13.99±0.12 | 14.98±0.05 | 7.37±0.10 | 5.07±0.13 |
| | | | 0 | −1.386±0.040 | 26.26±0.24 | 17.64±0.07 | 14.92±0.17 | 51.26±1.38 |
| | DRaFT-1 | $\sqrt{2\eta_t}$ | $\sqrt{2\eta_t}$ | 1.033±0.051 | 25.98±0.25 | 18.28±0.07 | 22.08±0.18 | 14.47±0.67 |
| | | 0 | 0 | 1.236±0.038 | 31.54±0.27 | 19.53±0.07 | 24.47±0.19 | 24.78±0.88 |
| | Adj.-Match. $\lambda = 12500$ | $\sqrt{2\eta_t}$ | $\sqrt{2\eta_t}$ | −2.104±0.074 | 17.12±0.56 | 15.76±0.20 | 11.48±1.03 | 9.88±0.81 |
| | | | 0 | 0.607±0.055 | 31.36±0.20 | 19.56±0.08 | 23.23±0.28 | 33.75±1.48 |
| 20 | None (Base) | N/A | $\sqrt{2\eta_t}$ | −2.275±0.002 | 14.58±0.13 | 15.07±0.05 | 7.47±0.10 | 11.27±0.33 |
| | | | 0 | −1.017±0.055 | 27.92±0.19 | 18.01±0.07 | 17.17±0.15 | 54.69±1.45 |
| | DRaFT-1 | $\sqrt{2\eta_t}$ | $\sqrt{2\eta_t}$ | **1.301±0.039** | 27.09±0.24 | 18.93±0.07 | 23.78±0.20 | 21.05±1.12 |
| | | 0 | 0 | 1.255±0.038 | 31.14±0.25 | 19.43±0.06 | 24.52±0.16 | 26.15±1.11 |
| | Adj.-Match. $\lambda = 12500$ | $\sqrt{2\eta_t}$ | $\sqrt{2\eta_t}$ | −0.032±0.072 | 25.07±0.27 | 18.01±0.07 | 20.75±0.23 | 29.06±2.34 |
| | | | 0 | 0.768±0.048 | **31.70±0.17** | **19.73±0.08** | 24.30±0.26 | 35.90±1.52 |
| 40 | None (Base) | N/A | $\sqrt{2\eta_t}$ | −1.384±0.040 | 24.15±0.26 | 17.25±0.06 | 16.19±0.17 | 53.60±1.37 |
| | | | 0 | −0.920±0.042 | 28.32±0.22 | 18.15±0.07 | 17.89±0.16 | **56.53±1.52** |
| | DRaFT-1 | $\sqrt{2\eta_t}$ | $\sqrt{2\eta_t}$ | **1.357±0.039** | 30.18±0.24 | 19.38±0.08 | 24.61±0.17 | 25.54±0.99 |
| | | 0 | 0 | 1.251±0.040 | 30.95±0.28 | 19.37±0.06 | 24.37±0.17 | 27.39±1.14 |
| | Adj.-Match. $\lambda = 12500$ | $\sqrt{2\eta_t}$ | $\sqrt{2\eta_t}$ | 0.882±0.058 | 30.62±0.30 | 19.50±0.09 | **24.95±0.28** | 34.50±1.33 |
| | | | 0 | 0.778±0.050 | **31.65±0.19** | **19.76±0.08** | 24.49±0.27 | 37.24±1.57 |
| 100 | None (Base) | N/A | $\sqrt{2\eta_t}$ | −0.881±0.041 | 27.83±0.19 | 18.10±0.07 | 18.43±0.17 | **57.21±1.50** |
| | | | 0 | −0.881±0.036 | 28.65±0.18 | 18.22±0.06 | 18.20±0.17 | 57.73±1.68 |
| | DRaFT-1 | $\sqrt{2\eta_t}$ | $\sqrt{2\eta_t}$ | **1.343±0.040** | 30.64±0.20 | 19.38±0.08 | 24.37±0.17 | 25.51±1.10 |
| | | 0 | 0 | 1.239±0.037 | 30.74±0.28 | 19.33±0.06 | 24.24±0.17 | 28.70±1.11 |
| | Adj.-Match. $\lambda = 12500$ | $\sqrt{2\eta_t}$ | $\sqrt{2\eta_t}$ | 0.892±0.044 | 31.23±0.23 | **19.65±0.08** | **24.92±0.23** | 35.13±1.40 |
| | | | 0 | 0.779±0.048 | **31.64±0.17** | **19.76±0.08** | 24.57±0.25 | 38.26±1.65 |
| 200 | None (Base) | N/A | $\sqrt{2\eta_t}$ | −0.848±0.048 | 28.37±0.21 | 18.27±0.08 | 18.56±0.19 | **58.00±1.58** |
| | | | 0 | −0.871±0.036 | 28.50±0.18 | 18.23±0.06 | 18.25±0.14 | 57.84±1.60 |
| | DRaFT-1 | $\sqrt{2\eta_t}$ | $\sqrt{2\eta_t}$ | **1.331±0.044** | 30.69±0.23 | 19.36±0.07 | 24.21±0.17 | 26.41±1.18 |
| | | 0 | 0 | 1.222±0.042 | 30.77±0.27 | 19.32±0.06 | 24.18±0.16 | 29.09±1.07 |
| | Adj.-Match. $\lambda = 12500$ | $\sqrt{2\eta_t}$ | $\sqrt{2\eta_t}$ | 0.869±0.062 | 31.33±0.21 | **19.68±0.09** | **24.81±0.30** | 35.90±1.55 |
| | | | 0 | 0.766±0.050 | **31.61±0.16** | **19.75±0.08** | 24.52±0.24 | 38.60±1.38 |

Table 8: Performance metrics for different number of sampling steps. Only the number of sampling steps is ablated; the fine-tuned models used in all cases are the ones fine-tuned using 40 steps.

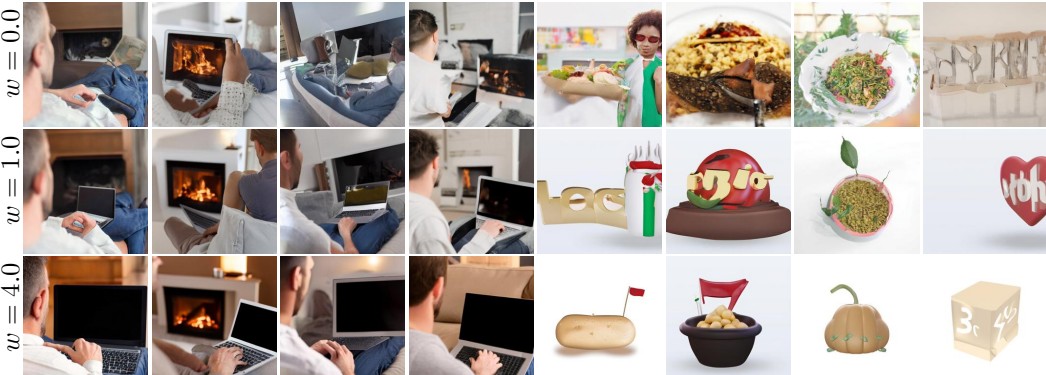

Text prompt: "*Man sitting on sofa at home in front of fireplace and using laptop computer, rear view*"

Text prompt: "*3D World Food Day Morocco*"

Figure 7: Generated samples from varying classifier-free guidance weights, from the pre-trained Flow Matching model. Corresponding samples from the fine-tuned model can be found in Fig. 6.

Base Flow Matching model      Adjoint Matching (Ours)      DRaFT-1

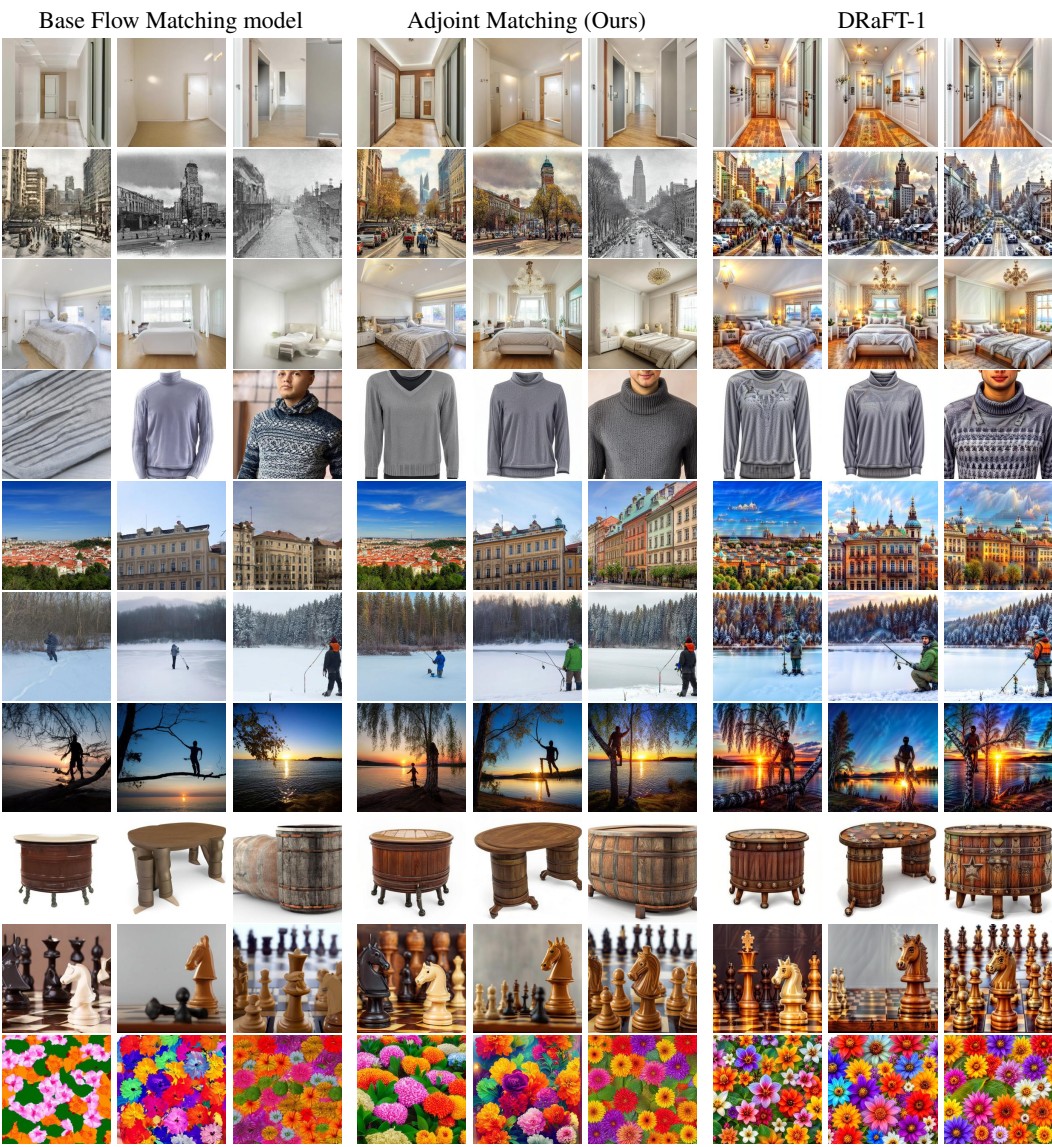

Figure 8: Generated samples with classifier-free guidance ($w = 1$) and $\sigma(t) = 0$ across ten selected prompts. Each row corresponds to a different prompt and each image corresponds to a different random seed consistent across models.

Base Flow Matching model          Adjoint Matching (Ours)          DRaFT-1

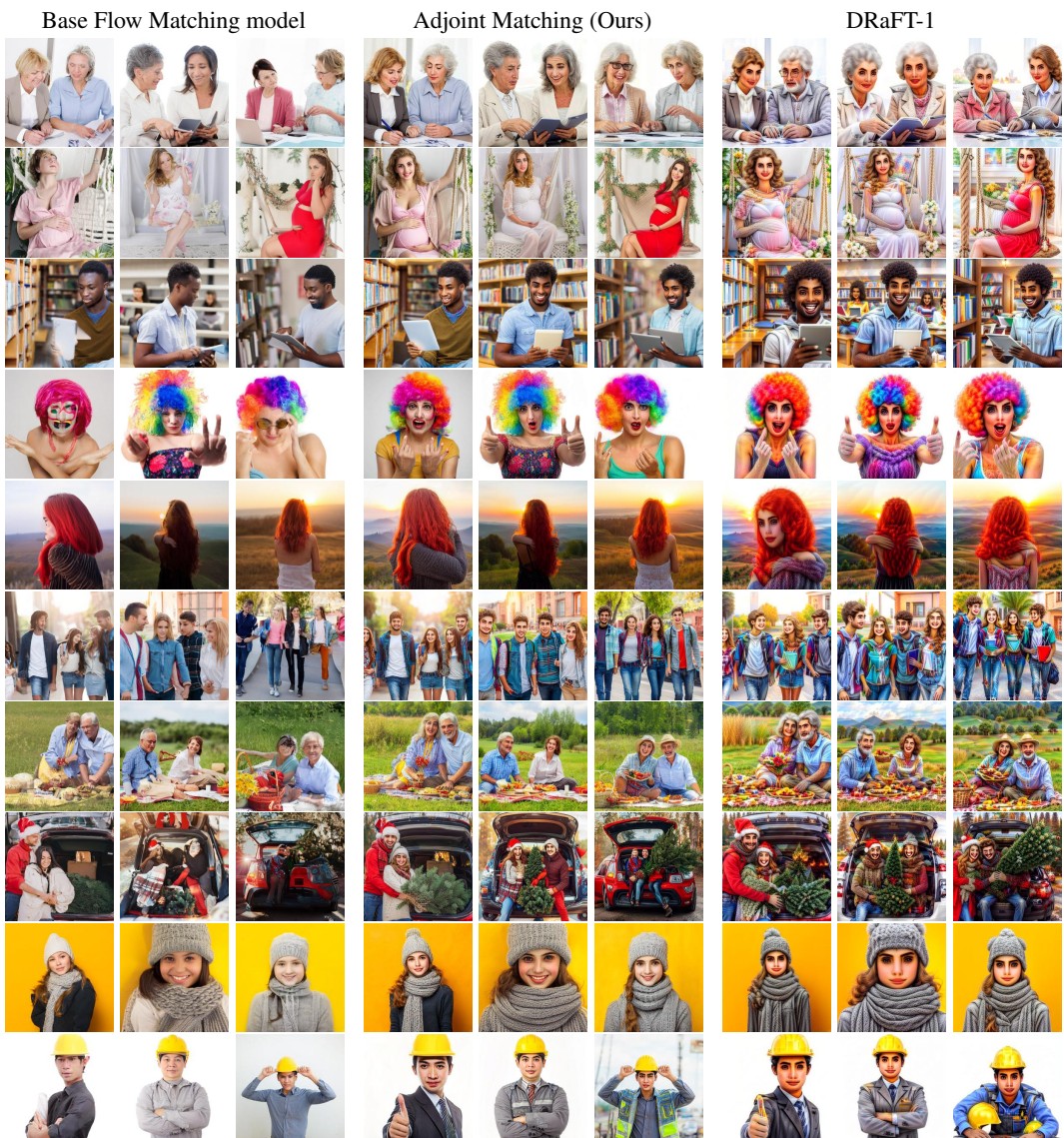

Figure 9: Generated samples with classifier-free guidance ($w = 1$) and $\sigma(t) = 0$ across ten selected prompts with people. Each row corresponds to a different prompt and each image corresponds to a different random seed consistent across models.

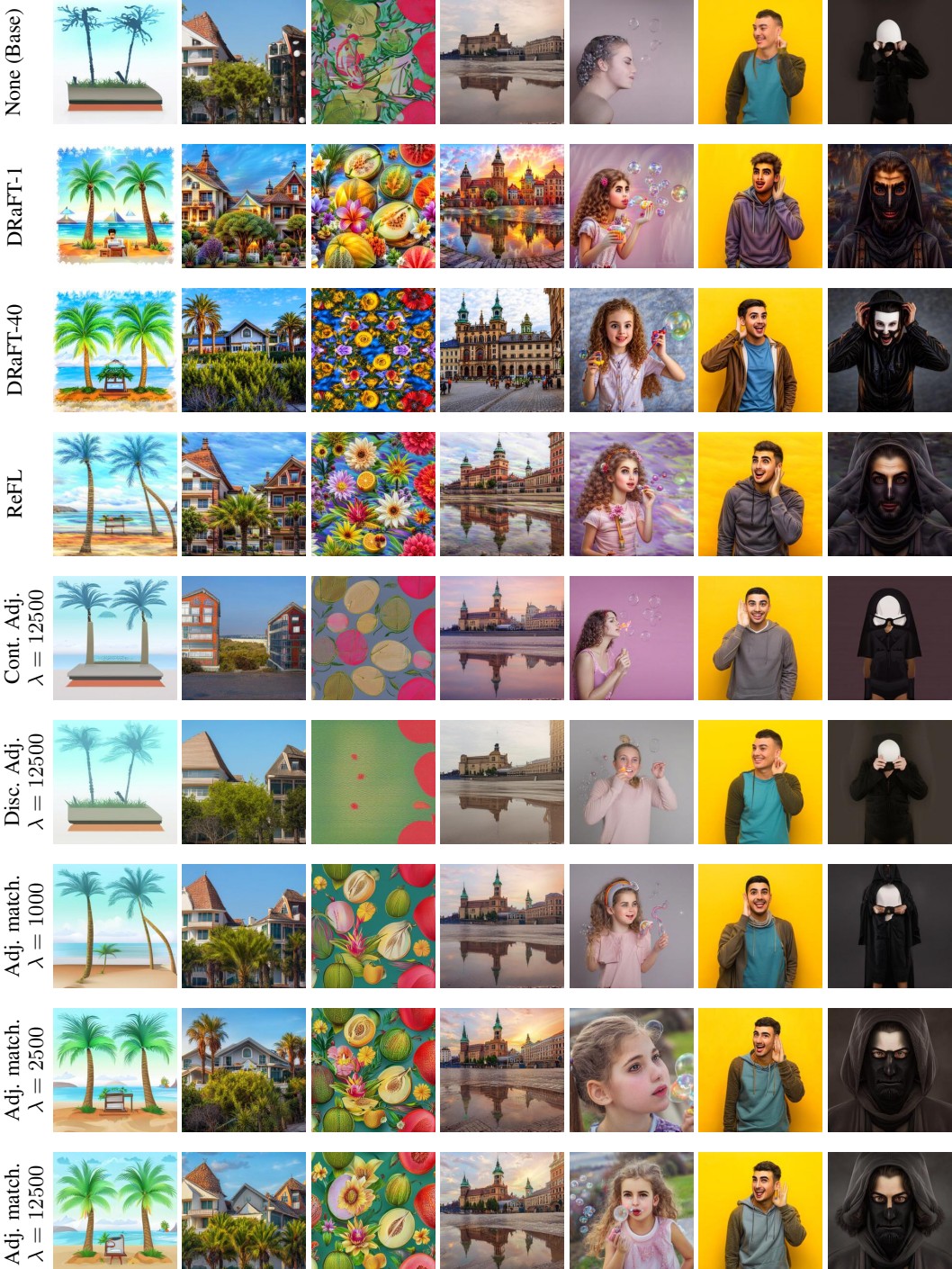

Figure 10: Generated samples without guidance ($w = 0$) and $\sigma(t) = 0$ across seven selected prompts. Each row corresponds to a different finetuning algorithm. Prompts: "*Seaside view poster with palm trees vector image*", "*Cayucos Beach Inn*", "*Happy Summer Life- Aloha Flowers and Melon - Pattern Metal Print*", "*Castle Square, Warsaw Old Town*", "*Funny girl blowing soap bubbles. High quality photo*", "*Colombian man with sweatshirt over yellow wall listening to something by putting hand on the ear*", "*man in the hood black mask masquerade*".

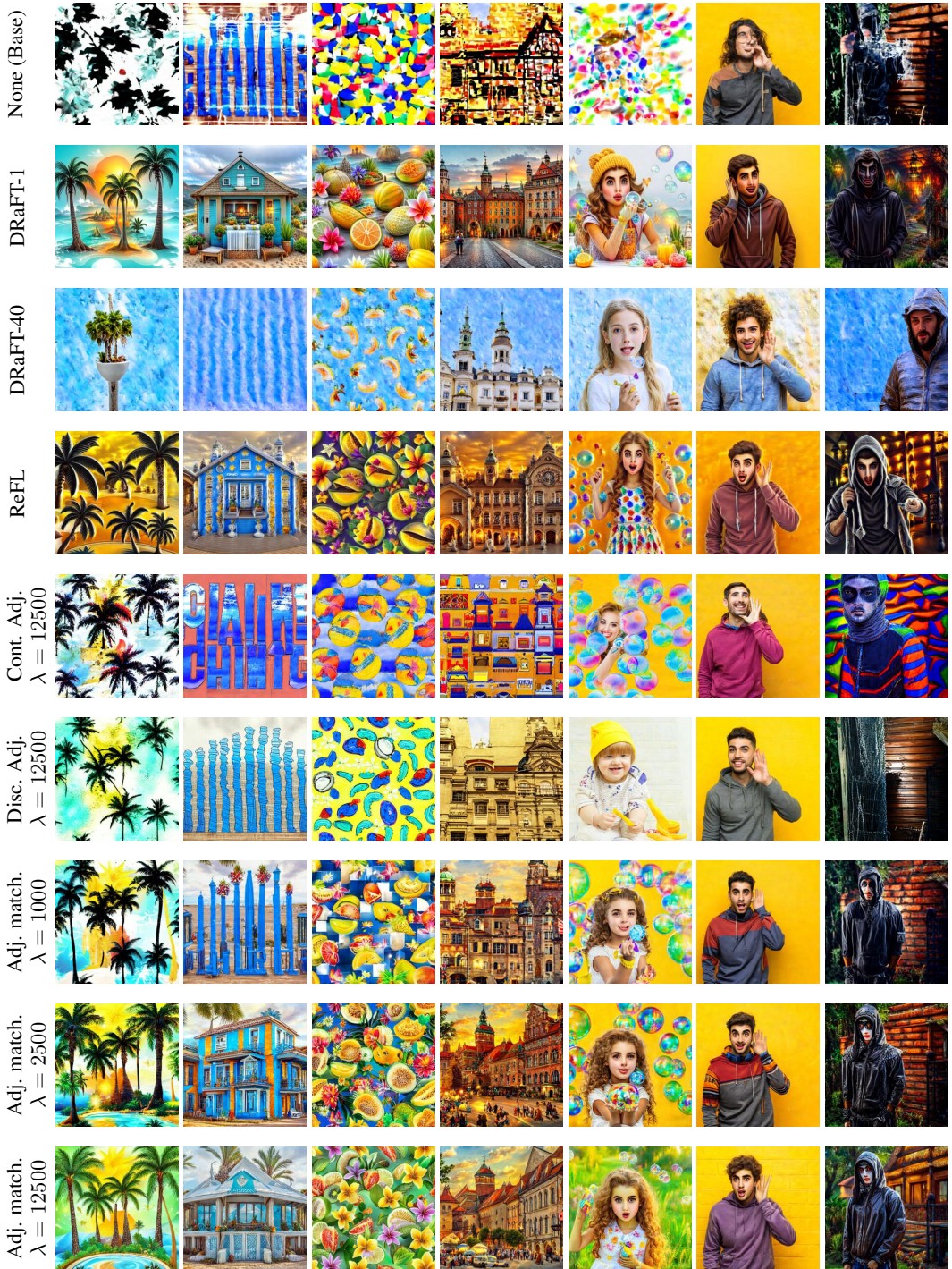

Figure 11: Generated samples without guidance ($w = 0$) and $\sigma(t) = \sqrt{2\eta_t}$ across seven selected prompts. Each row corresponds to a different finetuning algorithm. The prompts are the same as in Fig. 10.

## B    RELATED WORK

**Fine-tuning from human feedback.**    There are two main overarching approaches to RLHF: the *reward-based* approach (Ziegler et al., 2020; Stiennon et al., 2020; Ouyang et al., 2022; Bai et al., 2022) and *direct preference optimization* (DPO; Rafailov et al. (2023)). The reward-based approach (Ziegler et al., 2020; Stiennon et al., 2020; Ouyang et al., 2022; Bai et al., 2022) consists in learning the reward model $r(x)$ from human preference data, and then solving a maximum entropy RL problem with rewards produced by $r(x)$. DPO merges the two previous steps into one: there is no need to learn $r(x)$ as human preference data is directly used to fine-tune the model. However, DPO is typically only applied with a filtered dataset, and does not work explicitly with a reward model. Furthermore, for flow and diffusion models specifically, it is possible to differentiate the reward function, so there is a larger emphasis on reward-based approaches.

**Fine-tuning for diffusion models.**    Among existing reward-based diffusion fine-tuning methods, Fan & Lee (2023) interpret the denoising process as a multi-step decision-making task and use policy gradient algorithms to fine-tune diffusion samplers. Black et al. (2024) makes use of proximal policy gradients for fine-tuning but this does not make use of the differentiability of the reward model. Fan et al. (2023) also consider KL-regularized rewards (13) but do not make the critical connection to the tilted distribution (1) that we flesh out in Subsec. 3.2. The fine-tuning algorithms of Xu et al. (2023); Clark et al. (2024) directly take gradients of the reward model and use heuristics to try to stay close to the original base generative model, but their behavior is not well understood and unrelated to the tilted distribution: Xu et al. (2023) takes gradients of the reward applied on the denoised sample at different points in time, and Clark et al. (2024) backpropagates the reward function through all or part of the diffusion trajectory. Finally, Uehara et al. (2024b) also fine-tune diffusion models with the goal of sampling from the tilted distribution (1), but their approach is much more involved than ours as it requires learning a value function, and solving two stochastic optimal control problems. Additional reward fine-tuning works include Bruna & Han (2024), that provide theoretical guarantees to sample from the tilted distribution when the reward is a quadratic function, and Zhang et al. (2024), that propose a reward fine-tuning algorithm for the GFlowNet architecture.

**Inference-time optimization methods.**    Some have proposed methods that do not update the base model but instead modify the generation process directly. One approach is to add a guidance term to the velocity (Chung et al., 2022; Song et al., 2023; Pokle et al., 2023); however, this is a heuristic and it is not well-understood what particular distribution is being generated. Another approach is to directly optimize the initial noise distribution (Li, 2021; Wallace et al., 2023b; Ben-Hamu et al., 2024); this is taking an opposite approach to the inital value bias problem than us by moving all of the work into optimizing the initial distribution. A more computationally intensive approach is to perform online estimation of the optimal control, for the purpose of heuristically solving an optimal control problem within the sampling process (Huang et al., 2024; Rout et al., 2024); these approaches aim to solve a separate control problem for each generated sample, instead of performing amortization (Amos et al., 2023) to learn a fine-tuned generative model.

**Optimal control in generative modeling.**    Methods from optimal control have been used to train dynamical generative models parameterized by ODEs (Chen et al., 2018), SDEs (Li et al., 2020), and jump processes (**?**), enabled through the adjoint method. They can be used to train arbitrary generative processes, but for simplified constructions these have fallen in favor of simulation-free matching objectives such as denoising score matching (Vincent, 2011) and Flow Matching (Lipman et al., 2023). The optimal control formalism also has significance in sampling from un-normalized distributions (Zhang & Chen, 2022; Berner et al., 2023; Vargas et al., 2023; 2022; Richter & Berner, 2024; Tzen & Raginsky, 2019). The inclusion of a state cost has been used to solve transport problems where intermediate path distributions are of importance (Liu et al., 2024; Pooladian et al., 2024). These collective advances naturally lead to the consideration of the optimal control formalism for reward fine-tuning.

**Conditional sampling in inverse problems.**    Denker et al. (2024) and Wu et al. (2023a) independently consider a pre-trained diffusion model $p(x)$, and an observation $y$ on the generated sample $x$, as well as the analytic likelihood $p(y|x)$. Their aim is to sample from the posterior $p(x)p(y|x)$, and their applications include inpainting, class-conditional generation, super-resolution, phase retrieval,

non-linear deblurring, computed tomography, and protein design. Their setting reduces to a particular case of our reward fine-tuning framework by setting $r(x) = \log p(y|x)$. Denker et al. (2024) formulate an SOC problem, and they solve it via the log-variance loss (Richter et al. (2020); Nüsken & Richter (2021)), and the moment loss (Nüsken & Richter, 2021), which they refer to as the trajectory balance loss (Malkin et al., 2023). Wu et al. (2023a) propose Twisted Diffusion Sampler, an algorithm based on Sequential Monte Carlo that uses increased inference-time compute to reduce bias. A third work that also tackles the conditional sampling problem is Du et al. (2024), which use a Lagrangian formulation that they solve approximately using Gaussian paths.

## C    RESULTS ON DDIM AND FLOW MATCHING

### C.1    DENOISING DIFFUSION MODELS

We next discuss diffusion models, in particular the sampling scheme proposed by Denoising Diffusion Implicit Model (DDIM; Song et al. (2021a)) which we will later relate to Denoising Diffusion Probabilistic Models (DDPM; Ho et al. (2020)) as a particular case of the former. For sampling from a diffusion model, the DDIM update rule[5] (Song et al. (2021a), Eq. 12), typically stated in discrete time with $k \in \{0, \dots, K\}$, is:

$$X_{k+1} = \sqrt{\bar{\alpha}_{k+1}}\big(\tfrac{X_k - \sqrt{1-\bar{\alpha}_k}\epsilon(X_k,k)}{\sqrt{\bar{\alpha}_k}}\big) + \sqrt{1 - \bar{\alpha}_{k+1} - \sigma_k^2}\epsilon(X_k, k) + \sigma_k \varepsilon_k, \qquad (28)$$

where $\varepsilon_k \sim \mathcal{N}(0, I)$, $X_0 \sim \mathcal{N}(0, I)$, $(\bar{\alpha}_k)$ is an increasing sequence such that $\bar{\alpha}_0 = 0$, $\bar{\alpha}_K = 1$, and the sequence $\sigma_k$ is arbitrary. That is, one samples an initial Gaussian random variable $x_0$, and applies the stochastic update (28) iteratively $K$ times in order to obtain an artificial sample $X_K$. Updates can be interpreted as progressively denoising the iterate: $x_0$ is completely noisy and $x_K$ is fully denoised. The noise predictor model $\epsilon(x_k, k)$ is trained to predict the noise of $x_k$ (see *e.g.* Ho et al. (2020) for details on pre-training denoising diffusion models).

To convert DDIM to a continuous-time stochastic process, we can show that the DDIM update rule (28), up to a first-order approximation, is equivalent to the Euler-Maruyama discretization of the following SDE:

$$\mathrm{d}X_t = \big(\tfrac{\dot{\bar{\alpha}}_t}{2\bar{\alpha}_t}X_t - \big(\tfrac{\dot{\bar{\alpha}}_t}{2\bar{\alpha}_t} + \tfrac{\sigma(t)^2}{2}\big)\tfrac{\epsilon^{\text{base}}(X_t,t)}{\sqrt{1-\bar{\alpha}_t}}\big)\mathrm{d}t + \sigma(t)\mathrm{d}B_t, \qquad X_0 \sim \mathcal{N}(0, I). \qquad (29)$$

See App. C.2 for the full derivation. To go from (28) to (29), we assumed a uniform discretization of time, *i.e.* $t = \tfrac{k}{K}$. This results in identifying the discrete-time process $(X_k)_{k\in\{0,\dots,K\}}$ with a continuous-time process $(X_t)_{t\in[0,1]}$, where $\bar{\alpha}_k := \bar{\alpha}_t$, $\sigma_k := \tfrac{1}{\sqrt{K}}\sigma(t)$, and $\epsilon(X_k, k)$ with $\epsilon^{\text{base}}(X_k, t)$. In relation to the reference flow (2), the generative process in (29) has the same time marginals when $\alpha_t = \sqrt{\bar{\alpha}_t}$ and $\beta_t = \sqrt{1 - \bar{\alpha}_t}$ (Ho et al., 2020).

Furthermore, when viewed up to first order approximations, the DDPM sampling scheme (Ho et al. (2020); Algorithm 2) can be seen as special instance of the DDIM sampling scheme when $\sigma(t) = \sqrt{\dot{\bar{\alpha}}_t/\bar{\alpha}_t}$. This results in the following generative process:

$$\mathrm{d}X_t = \big(\tfrac{\dot{\bar{\alpha}}_t}{2\bar{\alpha}_t}X_t - \tfrac{\dot{\bar{\alpha}}_t}{\bar{\alpha}_t}\tfrac{\epsilon^{\text{base}}(X_t,t)}{\sqrt{1-\bar{\alpha}_t}}\big)\mathrm{d}t + \sqrt{\tfrac{\dot{\bar{\alpha}}_t}{\bar{\alpha}_t}}\mathrm{d}B_t, \qquad X_0 \sim \mathcal{N}(0, I), \qquad (30)$$

### C.2    THE CONTINUOUS-TIME LIMIT OF DDIM

The DDIM inference update (Song et al., 2021a, Eq. 12) is

$$x_{k+1} = \sqrt{\bar{\alpha}_{k+1}}\big(\tfrac{x_k - \sqrt{1-\bar{\alpha}_k}\epsilon(x_k,k)}{\sqrt{\bar{\alpha}_k}}\big) + \sqrt{1 - \bar{\alpha}_{k+1} - \sigma_k^2}\epsilon(x_k, k) + \sigma_k \epsilon_k, \qquad x_K \sim N(0, I). \tag{31}$$

If we let $\Delta\bar{\alpha}_k = \bar{\alpha}_{k+1} - \bar{\alpha}_k$, we have that

$$\sqrt{\tfrac{\bar{\alpha}_{k+1}}{\bar{\alpha}_k}} = \sqrt{\tfrac{\bar{\alpha}_k + \bar{\alpha}_{k+1} - \bar{\alpha}_k}{\bar{\alpha}_k}} = \sqrt{1 + \tfrac{\bar{\alpha}_{k+1} - \bar{\alpha}_k}{\bar{\alpha}_k}} = \sqrt{1 + \tfrac{\Delta\bar{\alpha}_k}{\bar{\alpha}_k}} \approx 1 + \tfrac{\Delta\bar{\alpha}_k}{2\bar{\alpha}_k}, \qquad (32)$$

---

[5]We slightly depart from the notation in Song et al. (2021a) by flipping the direction of time and using $\bar{\alpha}_k$ which corresponds to the $\alpha_k$ in Song et al. (2021a) while it corresponds to the $\bar{\alpha}_k$ in Ho et al. (2020).

where we used the first-order Taylor approximation of $\sqrt{1+x}$. And

$$-\sqrt{\frac{\bar{\alpha}_{k+1}}{\bar{\alpha}_k}(1-\bar{\alpha}_k)} + \sqrt{1-\bar{\alpha}_{k+1}-\sigma_k^2} = -\sqrt{\left(1+\frac{\Delta\bar{\alpha}_k}{\bar{\alpha}_k}\right)(1-\bar{\alpha}_k)} + \sqrt{1-\bar{\alpha}_{k+1}-\sigma_k^2}$$

$$= -\sqrt{1+\frac{\Delta\bar{\alpha}_k}{\bar{\alpha}_k}-\bar{\alpha}_k-\Delta\bar{\alpha}_k} + \sqrt{1-\bar{\alpha}_{k+1}-\sigma_k^2} = -\sqrt{1-\bar{\alpha}_{k+1}+\frac{\Delta\bar{\alpha}_k}{\bar{\alpha}_k}} + \sqrt{1-\bar{\alpha}_{k+1}-\sigma_k^2}$$

$$= \sqrt{1-\bar{\alpha}_{k+1}}\left(-\sqrt{1+\frac{\Delta\bar{\alpha}_k}{\bar{\alpha}_k(1-\bar{\alpha}_{k+1})}} + \sqrt{1-\frac{\sigma_k^2}{1-\bar{\alpha}_{k+1}}}\right)$$

$$\approx \sqrt{1-\bar{\alpha}_{k+1}}\left(-\left(1+\frac{\Delta\bar{\alpha}_k}{2\bar{\alpha}_k(1-\bar{\alpha}_{k+1})}\right) + 1 - \frac{\sigma_k^2}{2(1-\bar{\alpha}_{k+1})}\right) = -\left(\frac{\Delta\bar{\alpha}_k}{2\bar{\alpha}_k} + \frac{\sigma_k^2}{2}\right)\frac{1}{\sqrt{1-\bar{\alpha}_{k+1}}},$$

$$(33)$$

where we used the same first-order Taylor approximation. Thus, up to first-order approximations, (31) is equivalent to

$$x_{k-1} = \left(1+\frac{\Delta\bar{\alpha}_k}{2\bar{\alpha}_k}\right)x_k - \left(\frac{\Delta\bar{\alpha}_k}{2\bar{\alpha}_k} + \frac{\sigma_k^2}{2}\right)\frac{\epsilon(x_k,k)}{\sqrt{1-\bar{\alpha}_{k+1}}} + \sigma_k\epsilon_k, \qquad x_K \sim N(0,I). \qquad (34)$$

If we modify our notation slightly, we can rewrite this as

$$X_{(k+1)h} = \left(1-\frac{h\dot{\bar{\alpha}}_{kh}}{2\bar{\alpha}_{kh}}\right)X_{kh} + \left(\frac{h\dot{\bar{\alpha}}_{kh}}{2\bar{\alpha}_{kh}} - \frac{h\sigma(kh)^2}{2}\right)\frac{\epsilon(X_{kh},kh)}{\sqrt{1-\bar{\alpha}_{kh}}} + \sqrt{h}\sigma(kh)\epsilon_k, \qquad X_0 \sim N(0,I). \qquad (35)$$

To go from (34) to (35), we introduced a continuous time variable and a step size $h = 1/K$, and we regard the increment $h\bar{\alpha}_k$ as approximately equal to $h$ times the derivative of $\bar{\alpha}$. We also identified $\sigma_k$ with $\sqrt{h}\sigma(kh)$, where $\sigma(kh)$ plays the role of a diffusion coefficient. Note that equation (35) can be reverse-engineered as the Euler-Maruyama discretization of the SDE

$$\mathrm{d}X_t = \left(-\frac{\dot{\bar{\alpha}}_t}{2\bar{\alpha}_t} + \left(\frac{\dot{\bar{\alpha}}_t}{2\bar{\alpha}_t} - \frac{\sigma(t)^2}{2}\right)\frac{\epsilon(X_t,t)}{\sqrt{1-\bar{\alpha}_t}}\right)\mathrm{d}t + \sigma(t)\mathrm{d}B_t, \qquad X_0 \sim N(0,I). \qquad (36)$$

### C.3 Forward and backward stochastic differential equations

Let $(\kappa_t)_{t\in[0,1]}$ and $(\eta_t)_{t\in[0,1]}$ such that

$$\forall t \in [0,1], \quad \eta_t \geq 0, \qquad \int_0^1 \kappa_{1-s}\,\mathrm{d}s = +\infty, \qquad 2\int_0^1 \eta_{1-t'}\exp\left(-2\int_{t'}^t \kappa_{1-s}\,\mathrm{d}s\right)\mathrm{d}t' = 1. \qquad (37)$$

As shown in Tab. 1, DDIM corresponds to $\kappa_t = \frac{\dot{\bar{\alpha}}_t}{2\bar{\alpha}_t}$, $\eta_t = \frac{\dot{\bar{\alpha}}_t}{2\bar{\alpha}_t}$, and Flow Matching corresponds to $\kappa_t = \frac{\dot{\alpha}_t}{\alpha_t}$, $\eta_t = \beta_t\left(\frac{\dot{\alpha}_t}{\alpha_t}\beta_t - \dot{\beta}_t\right)$.

**Lemma 1** (DDIM and Flow Matching fulfill the conditions (37)). *The choices of $(\kappa_t)_{t\in[0,1]}$ and $(\eta_t)_{t\in[0,1]}$ for DDIM and Flow Matching fulfill the conditions* (37). *For DDIM, we have that*

$$\int_0^t \kappa_{1-s}\,\mathrm{d}s = -\tfrac{1}{2}\log\bar{\alpha}_{1-t} \implies \int_0^1 \kappa_{1-s}\,\mathrm{d}s = +\infty,$$

$$2\int_0^t \eta_{t'}\exp\left(-2\int_{t'}^t \kappa_s\,\mathrm{d}s\right)\mathrm{d}t' = 1-\bar{\alpha}_{1-t} \implies 2\int_0^1 \eta_{t'}\exp\left(-2\int_{t'}^t \kappa_s\,\mathrm{d}s\right)\mathrm{d}t' = 1. \qquad (38)$$

*For Flow Matching,*

$$\int_0^t \kappa_{1-s}\,\mathrm{d}s = -\log\alpha_{1-t} \implies \int_0^1 \kappa_{1-s}\,\mathrm{d}s = +\infty, \qquad (39)$$

$$2\int_0^t \eta_{t'}\exp\left(-2\int_{t'}^t \kappa_s\,\mathrm{d}s\right)\mathrm{d}t' = \beta_{1-t}^2 \implies 2\int_0^1 \eta_{t'}\exp\left(-2\int_{t'}^t \kappa_s\,\mathrm{d}s\right)\mathrm{d}t' = 1. \qquad (40)$$

**Forward and backward SDEs** Consider the forward and backward SDEs

$$\mathrm{d}\vec{X}_t = -\kappa_{1-t}\vec{X}_t\,\mathrm{d}t + \sqrt{2\eta_{1-t}}\,\mathrm{d}B_t, \qquad \vec{X}_0 \sim p_{\mathrm{data}}, \qquad (41)$$

$$\mathrm{d}X_t = \left(\kappa_t X_t + 2\eta_t\mathfrak{s}(X_t,t)\right)\mathrm{d}t + \sqrt{2\eta_t}\,\mathrm{d}B_t, \qquad X_0 \sim N(0,I), \qquad (42)$$

where we let $\vec{p}_t$ be the density of $\vec{X}_t$, and we define the score function as $\mathfrak{s}(x,t) := \nabla\log\vec{p}_{1-t}(x)$. Similarly, we let $p_t$ be the density of $X_t$. $\vec{p}_t$ and $p_t$ solve the Fokker-Planck equations:

$$\partial_t\vec{p}_t = \nabla\cdot\left(\kappa_{1-t}x\vec{p}_t\right) + \eta_{1-t}\Delta\vec{p}_t, \qquad \vec{p}_0 = p_{\mathrm{data}}, \qquad (43)$$

$$\partial_t p_t = \nabla\cdot\left(\left(-\kappa_t x - 2\eta_t\nabla\log\vec{p}_{1-t}(X_t)\right)p_t\right) + \eta_t\Delta p_t, \qquad p_0 = N(0,I). \qquad (44)$$

**Lemma 2** (Solution of the forward SDE). *Let $(\kappa_t)_{t\geq0}$, $(\eta_t)_{t\geq0}$ with $\eta_t \geq 0$, and $(\xi_t)_{t\geq0}$ be arbitrary. The solution $\vec{X}_t$ of the SDE*

$$\mathrm{d}\vec{X}_t = \left( -\kappa_{1-t}\vec{X}_t + \xi_t \right)\mathrm{d}t + \sqrt{2\eta_{1-t}}\,\mathrm{d}B_t, \qquad \vec{X}_0 \sim p_{\mathrm{data}} \tag{45}$$

*is*

$$\vec{X}_t = \vec{X}_0 \exp\left( -\int_0^t \kappa_{1-s}\,\mathrm{d}s \right) + \int_0^t \exp\left( -\int_{t'}^t \kappa_{1-s}\,\mathrm{d}s \right)\xi_{1-t'}\,\mathrm{d}t' + \int_0^t \sqrt{2\eta_{1-t'}}\exp\left( -\int_{t'}^t \kappa_{1-s}\,\mathrm{d}s \right)\mathrm{d}B_{t'}, \tag{46}$$

*which has the same distribution as the random variable*

$$\hat{X}_t = \vec{X}_0 \exp\left( -\int_0^t \kappa_{1-s}\,\mathrm{d}s \right) + \int_0^t \exp\left( -\int_{t'}^t \kappa_{1-s}\,\mathrm{d}s \right)\xi_{1-t'}\,\mathrm{d}t' + \sqrt{2\int_0^t \eta_{1-t'}\exp\left( -2\int_{t'}^t \kappa_{1-s}\,\mathrm{d}s \right)\mathrm{d}t'}\,\epsilon,$$
$$\epsilon \sim N(0, I). \tag{47}$$

Applying Lemma 2 with $\xi_t \equiv 0$, we obtain that $\vec{p}_1$ is also the distribution of

$$\hat{X}_1 = \vec{X}_0 \exp\left( -\int_0^t \kappa_{1-s}\,\mathrm{d}s \right) + \sqrt{2\int_0^t \eta_{1-t'}\exp\left( -2\int_{t'}^t \kappa_{1-s}\,\mathrm{d}s \right)\mathrm{d}t'}\,\epsilon = \epsilon, \tag{48}$$

where $\epsilon \sim N(0, I)$. The third equality in (48) holds by (37). Hence we obtain that $\vec{p}_1 = N(0, I)$. Note also that

$$\partial_t \vec{p}_{1-t} = -\nabla \cdot \left( \kappa_t x \vec{p}_{1-t} \right) - \eta_t \Delta \vec{p}_{1-t} = -\nabla \cdot \left( \left( -\kappa_t x - 2\eta_t \nabla \log \vec{p}_{1-t}(x) \right)\vec{p}_{1-t} \right) + \eta_t \Delta \vec{p}_{1-t} \tag{49}$$

Thus, $\vec{p}_{1-t}$ is a solution of the backward Fokker-Planck equation (44), which proves the following:

**Proposition 3** (Equality of marginal distributions). *For any time $t \in [0, 1]$, the densities of the solutions $\vec{X}_t$, $X_t$ of the forward and backward SDEs are equal up to a time flip: $p_t = \vec{p}_{1-t}$.*

**Forward and backward SDEs with arbitrary noise schedule** Next, we look at the following pair of forward-backward SDEs:

$$\mathrm{d}\vec{X}_t = \left( -\kappa_{1-t}\vec{X}_t + \left( \tfrac{\sigma(1-t)^2}{2} - \eta_{1-t} \right)\mathfrak{s}(\vec{X}_t, 1-t) \right)\mathrm{d}t + \sigma(1-t)\,\mathrm{d}B_t, \qquad \vec{X}_0 \sim p_{\mathrm{data}}, \tag{50}$$
$$\mathrm{d}X_t = \left( \kappa_t X_t + \left( \tfrac{\sigma(t)^2}{2} + \eta_t \right)\mathfrak{s}(X_t, t) \right)\mathrm{d}t + \sigma(t)\,\mathrm{d}B_t, \qquad X_0 \sim N(0, I), \tag{51}$$

Here, the score function $\mathfrak{s}$ is the same vector field as in (51). Remark that equations (41)-(42) are a particular case of (50)-(51) for which $\sigma(t) = \sqrt{2\eta_t}$. The Fokker-Planck equations for (50)-(51) are:

$$\partial_t \vec{p}_t = \nabla \cdot \left( \left( \kappa_{1-t} x + \left( -\tfrac{\sigma(1-t)^2}{2} + \eta_{1-t} \right)\mathfrak{s}(X_t, t) \right)\vec{p}_t \right) + \eta_{1-t}\Delta \vec{p}_t, \qquad \vec{p}_0 = p_{\mathrm{data}}, \tag{52}$$
$$\partial_t p_t = \nabla \cdot \left( \left( -\kappa_t x - \left( \tfrac{\sigma(t)^2}{2} + \eta_t \right)\mathfrak{s}(X_t, t) \right)p_t \right) + \tfrac{\sigma(t)^2}{2}\Delta p_t, \qquad p_0 = N(0, I). \tag{53}$$

It is straight-forward to see that for any $\sigma$, the solutions $\vec{p}_t$ and $p_t$ of (52)-(53) are also solutions of (43)-(44). Hence, the marginals $\vec{X}_t$ and $X_t$ are equally distributed for all noise schedules $\sigma$, and they are equal to each other up to a time flip.

**Equality of distributions over trajectories** The result in Prop. 3 can be made even stronger:

**Proposition 4** (Equality of distributions over trajectories). *Let $\vec{X}$, $X$ be the solutions of the SDEs (50)-(51) with arbitrary noise schedule. For any sequence of times $(t_i)_{0\leq i\leq I}$, the joint distribution of $(\vec{X}_{t_i})_{0\leq i\leq I}$ is equal to the joint distribution of $(X_{1-t_i})_{0\leq i\leq I}$, or equivalently, that the probability measures $\vec{\mathbb{P}}$, $\mathbb{P}$ of the forward and backward processes $\vec{X}$, $X$ are equal, up to a flip in the time direction.*

This result states that sampling trajectories from the backward process is equivalent to sampling them from the forward process and then flipping their order.

### C.3.1 PROOF OF LEMMA 1

As shown in Tab. 1, DDIM corresponds to $\kappa_t = \frac{\dot{\bar{\alpha}}_t}{2\bar{\alpha}_t}$, $\eta_t = \frac{\dot{\bar{\alpha}}_t}{2\bar{\alpha}_t}$. Thus, $\eta_t \geq 0$ because $\bar{\alpha}_t$ is increasing, and

$$\int_0^t \kappa_{1-s}\, ds = \int_0^t \frac{\dot{\bar{\alpha}}_{1-s}}{2\bar{\alpha}_{1-s}}\, ds = -\tfrac{1}{2}\int_0^t \partial_s \log \bar{\alpha}_{1-s}\, ds = -\tfrac{1}{2}\left(\log \bar{\alpha}_{1-t} - \log \bar{\alpha}_1\right) = -\tfrac{1}{2}\log \bar{\alpha}_{1-t},$$

$$\implies \int_0^1 \kappa_{1-s}\, ds = -\tfrac{1}{2}\log \bar{\alpha}_0 = +\infty$$

$$(54)$$

$$2\int_0^t \eta_{t'} \exp\left(-2\int_{t'}^t \kappa_s\, ds\right) dt' = \int_0^t \frac{\dot{\bar{\alpha}}_{1-t'}}{\bar{\alpha}_{1-t'}} \exp\left(-\int_{t'}^t \frac{\dot{\bar{\alpha}}_{1-s}}{\bar{\alpha}_{1-s}}\, ds\right) dt'$$

$$= \int_0^t \frac{\dot{\bar{\alpha}}_{1-t'}}{\bar{\alpha}_{1-t'}} \frac{\bar{\alpha}_{1-t}}{\bar{\alpha}_{1-t'}}\, dt' = \bar{\alpha}_{1-t}\int_0^t \partial_{t'}\left(\frac{1}{\bar{\alpha}_{1-t'}}\right) dt' = \bar{\alpha}_{1-t}\left(\frac{1}{\bar{\alpha}_{1-t}} - \frac{1}{\bar{\alpha}_1}\right) = 1 - \bar{\alpha}_{1-t}, \qquad (55)$$

$$\implies 2\int_0^1 \eta_{t'} \exp\left(-2\int_{t'}^t \kappa_s\, ds\right) dt' = 1 - \bar{\alpha}_0 = 1.$$

where we used that $\bar{\alpha}_1 = 1$ and $\bar{\alpha}_0 = 0$. And Flow Matching corresponds to $\kappa_t = \frac{\dot{\alpha}_t}{\alpha_t}$, $\eta_t = \beta_t\left(\frac{\dot{\alpha}_t}{\alpha_t}\beta_t - \dot{\beta}_t\right)$. We have that $\eta_t \geq 0$ because $\alpha_t$ is increasing and $\beta_t$ is decreasing, and

$$\int_0^t \kappa_{1-s}\, ds = \int_0^t \frac{\dot{\alpha}_{1-s}}{\alpha_{1-s}}\, ds = -\int_0^t \partial_s \log \alpha_{1-s}\, ds = -(\log \alpha_{1-t} - \log \alpha_1) = -\log \alpha_{1-t},$$

$$\implies \int_0^1 \kappa_{1-s}\, ds = -\log \alpha_0 = +\infty, \qquad (56)$$

and

$$2\int_0^t \eta_{1-t'} \exp\left(-2\int_{t'}^t \kappa_{1-s}\, ds\right) dt' = 2\int_0^t \beta_{1-t'}\left(\frac{\dot{\alpha}_{1-t'}}{\alpha_{1-t'}}\beta_{1-t'} - \dot{\beta}_{1-t'}\right) \exp\left(-2\int_{t'}^t \frac{\dot{\alpha}_{1-s}}{\alpha_{1-s}}\, ds\right) dt'$$

$$= 2\int_0^t \beta_{1-t'}\left(\frac{\dot{\alpha}_{1-t'}}{\alpha_{1-t'}}\beta_{1-t'} - \dot{\beta}_{1-t'}\right)\left(\frac{\alpha_{1-t}}{\alpha_{1-t'}}\right)^2 dt',$$

$$(57)$$

To develop the right-hand side, note that by integration by parts,

$$\int_0^t \dot{\beta}_{1-t'}\beta_{1-t'}\left(\frac{\alpha_{1-t}}{\alpha_{1-t'}}\right)^2 dt' = -\int_0^t \partial_{t'}\left(\frac{\beta_{1-t'}^2}{2}\right)\left(\frac{\alpha_{1-t}}{\alpha_{1-t'}}\right)^2 dt'$$

$$= -\left[\frac{\beta_{1-t'}^2}{2}\left(\frac{\alpha_{1-t}}{\alpha_{1-t'}}\right)^2\right]_0^1 + \int_0^t \frac{\beta_{1-t'}^2}{2}\partial_{t'}\left(\frac{\alpha_{1-t}}{\alpha_{1-t'}}\right)^2 dt' = -\left[\frac{\beta_{1-t'}^2}{2}\left(\frac{\alpha_{1-t}}{\alpha_{1-t'}}\right)^2\right]_0^t + \int_0^t \beta_{1-t'}^2 \frac{\alpha_{1-t}^2\dot{\alpha}_{1-t'}}{\alpha_{1-t'}^3}\, dt'. \qquad (58)$$

And if we plug this into the right-hand side of (57), we obtain

$$2\int_0^t \eta_{1-t'} \exp\left(-2\int_{t'}^t \kappa_{1-s}\, ds\right) dt' = \left[\beta_{1-t'}^2\left(\frac{\alpha_{1-t}}{\alpha_{1-t'}}\right)^2\right]_0^t = \beta_{1-t}^2 - \beta_1^2\left(\frac{\alpha_{1-t}}{\alpha_1}\right)^2 = \beta_{1-t}^2, \qquad (59)$$

$$\implies 2\int_0^1 \eta_{1-t'} \exp\left(-2\int_{t'}^t \kappa_{1-s}\, ds\right) dt' = \beta_1^2 = 1. \qquad (60)$$

where we used that $\beta_1 = 0$, $\alpha_1 = 1$.

### C.3.2 PROOF OF LEMMA 2

We can solve this equation by variation of parameters. To simplify the notation, we replace $\kappa_{1-s}$, $\eta_{1-s}$ and $\xi_{1-s}$ by $\kappa_s$, $\eta_s$ and $\xi_s$. Defining $f(\vec{X}_t, t) = \vec{X}_t \exp\left(\int_0^t \kappa_{1-s}\, ds\right)$, we get that

$$df(\vec{X}_t, t) = \kappa_{1-t}\vec{X}_t \exp\left(\int_0^t \kappa_{1-s}\, ds\right) dt + \exp\left(\int_0^t \kappa_{1-s}\, ds\right) d\vec{X}_t$$

$$= \kappa_{1-t}\vec{X}_t \exp\left(\int_0^t \kappa_{1-s}\, ds\right) dt + \exp\left(\int_0^t \kappa_{1-s}\, ds\right)\left((-\kappa_{1-t}\vec{X}_t + \xi_{1-t})\, dt + \sqrt{2\eta_{1-t}}\, dB_t\right)$$

$$= \exp\left(\int_0^t \kappa_{1-s}\, ds\right)\xi_{1-t}\, dt + \sqrt{2\eta_t}\exp\left(\int_0^t \kappa_{1-s}\, ds\right) dB_t. \qquad (61)$$

Integrating from 0 to $t$, we get that

$$\vec{X}_t \exp\left(\int_0^t \kappa_{1-s}\, ds\right) = \vec{X}_0 + \int_0^t \exp\left(\int_0^{t'} \kappa_{1-s}\, ds\right)\xi_{1-t'}\, dt' + \int_0^t \sqrt{2\eta_{1-t'}}\exp\left(\int_0^{t'} \kappa_{1-s}\, ds\right) dB_{t'}, \qquad (62)$$

$$\iff \vec{X}_t = \vec{X}_0 \exp\left(-\int_0^t \kappa_{1-s}\, ds\right) + \int_0^t \exp\left(-\int_{t'}^t \kappa_{1-s}\, ds\right)\xi_{1-t'}\, dt' \qquad (63)$$

$$+ \int_0^t \sqrt{2\eta_{1-t'}}\exp\left(-\int_{t'}^t \kappa_{1-s}\, ds\right) dB_{t'}. \qquad (64)$$

Since

$$\mathbb{E}\big[\big(\int_0^t \sqrt{2\eta_{1-t'}}\exp\big(-\int_{t'}^t \kappa_{1-s}\,\mathrm{d}s\big)\,\mathrm{d}B_{t'}\big)^2\big] = 2\int_0^t \eta_{1-t'}\exp\big(-2\int_{t'}^t \kappa_{1-s}\,\mathrm{d}s\big)\,\mathrm{d}t', \qquad (65)$$

we obtain that $\int_0^t \sqrt{2\eta_{1-t'}}\exp\big(-\int_{t'}^t \kappa_{1-s}\,\mathrm{d}s\big)\,\mathrm{d}B_{t'}$ has the same distribution as $\sqrt{2\int_0^t \eta_{1-t'}\exp\big(-2\int_{t'}^t \kappa_{1-s}\,\mathrm{d}s\big)\,\mathrm{d}t'}\,\epsilon$, where $\epsilon \sim N(0,1)$.

### C.3.3 PROOF OF PROP. 4

This is a result that has been used by previous works, e.g. (De Bortoli et al., 2021, Sec. 2.1), but their derivation lacks rigor as it uses some unexplained approximations. While natural, the result is not common knowledge in the area. We provide a derivation which is still in discrete time, and hence not completely formal, but that corrects the gaps in the proof of De Bortoli et al. (2021).

We introduce the short-hand

$$\vec{b}(x,t) = -\kappa_{1-t}x + \big(\tfrac{\sigma(1-t)^2}{2} - \eta_{1-t}\big)\mathfrak{s}(x, 1-t), \qquad (66)$$

$$b(x,t) = \kappa_t X_t + \big(\tfrac{\sigma(t)^2}{2} + \eta_t\big)\mathfrak{s}(X_t, t), \qquad (67)$$

$$\vec{\sigma}(t) = \sigma(1-t). \qquad (68)$$

Remark that $b(x,t) = -\vec{b}(x, 1-t) + \sigma(t)^2\mathfrak{s}(X_t, t)$.

Suppose that we discretize the forward process $\vec{X}$ using $K+1$ equispaced timesteps:

$$x_{k+1} = x_k + h\vec{b}(x_k, kh) + \sqrt{h}\vec{\sigma}(kh)\epsilon_k, \qquad \text{with } \epsilon_k \sim N(0,1). \qquad (69)$$

It is important to remark that $x_{k+1} - x_k = O(h^{1/2})$. Throughout the proof we will keep track of all terms up to linear order in $h$, while neglecting terms of order $O(h^{3/2})$ and higher. The distribution of the discretized forward process is:

$$\vec{p}(x_{0:K}) = \vec{p}_0(x_0)\prod_{k=0}^{K-1}\vec{p}_{k+1|k}(x_{k+1}|x_k), \quad \text{where } \vec{p}_{k+1|k}(x_{k+1}|x_k) = \frac{\exp\big(-\frac{\|x_{k+1}-x_k-h\vec{b}(x_k,kh)\|^2}{2h\vec{\sigma}(kh)^2}\big)}{(2\pi h\vec{\sigma}(kh)^2)^{d/2}}. \qquad (70)$$

Using telescoping products, we have that

$$\vec{p}(x_{0:K}) = \vec{p}_K(x_K)\prod_{k=0}^{K-1}\vec{p}_{k+1|k}(x_{k+1}|x_k)\frac{\vec{p}_k(x_k)}{\vec{p}_{k+1}(x_{k+1})}$$
$$= \vec{p}_K(x_K)\prod_{k=0}^{K-1}\vec{p}_{k+1|k}(x_{k+1}|x_k)\exp\big(\log(\vec{p}_k(x_k)) - \log(\vec{p}_{k+1}(x_{k+1}))\big) \qquad (71)$$

We can use a discrete time version of Ito's lemma:

$$\log\vec{p}(x_{k+1}, (k+1)h) \approx \log\vec{p}(x_k, kh) + h\big(\partial_t\log\vec{p}(x_k, kh) + \tfrac{\vec{\sigma}(kh)^2}{2}\Delta\log\vec{p}(x_k, kh)\big) \qquad (72)$$
$$+ \langle\nabla\log\vec{p}(x_k, kh), x_{k+1} - x_k\rangle + O(h^{3/2}). \qquad (73)$$

Using equation (69) and a Taylor approximation, observe that

$$\langle\nabla\log p(x_k, kh), x_{k+1} - x_k\rangle$$
$$= \langle\nabla\log p(x_{k+1}, (k+1)h) - \nabla^2\log p(x_{k+1}, (k+1)h)(x_{k+1} - x_k), x_{k+1} - x_k\rangle + O(h^{3/2})$$
$$= \langle\nabla\log p(x_{k+1}, (k+1)h), x_{k+1} - x_k\rangle$$
$$\quad - \langle h\vec{b}(x_k, kh) + \sqrt{h}\vec{\sigma}(kh)\epsilon_k, \nabla^2\log p(x_{k+1}, (k+1)h)\big(h\vec{b}(x_k, kh) + \sqrt{h}\vec{\sigma}(kh)\epsilon_k\big)\rangle + O(h^{3/2})$$
$$= \langle\nabla\log p(x_{k+1}, (k+1)h), x_{k+1} - x_k\rangle - h\vec{\sigma}(kh)^2\Delta\log p(x_{k+1}, (k+1)h) + O(h^{3/2}). \qquad (74)$$

And since $\vec{p}$ satisfies the Fokker-Planck equation

$$\partial_t\vec{p}_t = \nabla\cdot\big((-\vec{b}(x,t) + \tfrac{\vec{\sigma}(t)^2}{2}\nabla\log\vec{p}_t(x))\vec{p}_t\big), \qquad (75)$$

we have that

$$\partial_t\log\vec{p}_t = \frac{\partial_t\vec{p}_t}{\vec{p}_t} = \frac{\nabla\cdot\big((-\vec{b}(x,t) + \tfrac{\vec{\sigma}(t)^2}{2}\nabla\log\vec{p}_t(x))\vec{p}_t\big)}{\vec{p}_t}$$
$$= -\nabla\cdot\vec{b}(x,t) + \tfrac{\vec{\sigma}(t)^2}{2}\Delta\log\vec{p}_t(x) + \langle-\vec{b}(x,t) + \tfrac{\vec{\sigma}(t)^2}{2}\nabla\log\vec{p}_t(x), \nabla\log\vec{p}_t(x)\rangle. \qquad (76)$$

Hence,

$$\partial_t \log p(x_k, kh) = \partial_t \log p(x_{k+1}, (k+1)h) + O(h^{1/2})$$

$$= -\nabla \cdot \vec{b}(x_{k+1}, (k+1)h) + \frac{\vec{\sigma}((k+1)h)^2}{2} \Delta \log \vec{p}(x_{k+1}, (k+1)h)$$

$$+ \langle -\vec{b}(x_{k+1}, (k+1)h) + \frac{\vec{\sigma}((k+1)h)^2}{2} \nabla \log \vec{p}(x_{k+1}, (k+1)h), \nabla \log \vec{p}(x_{k+1}, (k+1)h) \rangle + O(h^{1/2}). \tag{77}$$

If we plug (74) and (77) into (72), we obtain

$$\log p(x_{k+1}, (k+1)h) - \log p(x_k, kh)$$

$$= h\big(-\nabla \cdot \vec{b}(x_{k+1}, (k+1)h) + \langle -\vec{b}(x_{k+1}, (k+1)h)$$

$$+ \frac{\vec{\sigma}((k+1)h)^2}{2} \nabla \log \vec{p}(x_{k+1}, (k+1)h), \nabla \log \vec{p}(x_{k+1}, (k+1)h) \rangle\big)$$

$$+ \langle \nabla \log p(x_{k+1}, (k+1)h), x_{k+1} - x_k \rangle + O(h^{3/2}) \tag{78}$$

$$= \frac{\langle 2h\vec{\sigma}(kh)^2 \nabla \log p(x_{k+1}, (k+1)h), x_{k+1} - x_k - h\vec{b}(x_{k+1}, (k+1)h) \rangle}{2h\vec{\sigma}(kh)^2}$$

$$+ h\big(-\nabla \cdot \vec{b}(x_{k+1}, (k+1)h) + \frac{\vec{\sigma}((k+1)h)^2}{2} \|\nabla \log \vec{p}(x_{k+1}, (k+1)h)\|^2\big) + O(h^{3/2}).$$

Applying a discrete time version of Ito's lemma again, we have that

$$\vec{b}(x_k, kh) = \vec{b}(x_{k+1}, (k+1)h) - h\big(\partial_t \vec{b}(x_{k+1}, (k+1)h) + \frac{\vec{\sigma}((k+1)h)^2}{2} \Delta \vec{b}(x_{k+1}, (k+1)h)\big)$$

$$+ \nabla \vec{b}(x_{k+1}, (k+1)h)^\top (x_k - x_{k+1}) + O(h^{3/2})$$

$$= \vec{b}(x_{k+1}, (k+1)h) + \nabla \vec{b}(x_{k+1}, (k+1)h)^\top (x_k - x_{k+1}) + O(h). \tag{79}$$

where $\Delta \vec{b}$ denotes the component-wise Laplacian of $\vec{b}$. Thus,

$$\log \vec{p}_{k+1|k}(x_{k+1}|x_k)$$

$$= -\frac{d}{2} \log\big(2\pi h\vec{\sigma}(kh)^2\big) - \frac{\|x_{k+1} - x_k - h\vec{b}(x_k, kh)\|^2}{2h\vec{\sigma}(kh)^2}$$

$$= -\frac{d}{2} \log\big(2\pi h\vec{\sigma}(kh)^2\big) - \frac{\|x_{k+1} - x_k - h(\vec{b}(x_{k+1}, (k+1)h) + \nabla \vec{b}(x_{k+1}, (k+1)h)^\top (x_k - x_{k+1}))\|^2}{2h\vec{\sigma}(kh)^2} + O(h^{3/2})$$

$$= -\frac{d}{2} \log\big(2\pi h\vec{\sigma}(kh)^2\big) - \frac{\|x_{k+1} - x_k - h\vec{b}(x_{k+1}, (k+1)h)\|^2}{2h\vec{\sigma}(kh)^2} + \frac{\langle x_{k+1} - x_k, \nabla \vec{b}(x_{k+1}, (k+1)h)^\top (x_k - x_{k+1}) \rangle}{\vec{\sigma}(kh)^2} + O(h^{3/2})$$

$$= -\frac{d}{2} \log\big(2\pi h\vec{\sigma}(kh)^2\big) - \frac{\|x_{k+1} - x_k - h\vec{b}(x_{k+1}, (k+1)h)\|^2}{h\vec{\sigma}(kh)^2} - \frac{h\vec{\sigma}(kh)^2 \langle \epsilon_k, \nabla \vec{b}(x_{k+1}, (k+1)h)^\top \epsilon_k \rangle}{\vec{\sigma}(kh)^2} + O(h^{3/2})$$

$$= -\frac{d}{2} \log\big(2\pi h\vec{\sigma}(kh)^2\big) - \frac{\|x_{k+1} - x_k - h\vec{b}(x_{k+1}, (k+1)h)\|^2}{h\vec{\sigma}(kh)^2} - h\Delta \vec{b}(x_{k+1}, (k+1)h) + O(h^{3/2}) \tag{80}$$

Combining (78) and (80), we obtain that

$$\log \vec{p}_{k+1|k}(x_{k+1}|x_k) - \big(\log p(x_{k+1}, (k+1)h) - \log p(x_k, kh)\big)$$

$$= -\frac{d}{2} \log\big(2\pi h\vec{\sigma}(kh)^2\big) - \frac{\|x_{k+1} - x_k - h\vec{b}(x_{k+1}, (k+1)h) + h\vec{\sigma}(kh)^2 \nabla \log p(x_{k+1}, (k+1)h)\|^2}{h\vec{\sigma}(kh)^2} + O(h^{3/2})$$

$$= -\frac{d}{2} \log\big(2\pi h\vec{\sigma}((k+1)h)^2\big) - \frac{\|x_{k+1} - x_k - h\vec{b}(x_{k+1}, (k+1)h) + h\vec{\sigma}((k+1)h)^2 \nabla \log p(x_{k+1}, (k+1)h)\|^2}{h\vec{\sigma}((k+1)h)^2} + O(h^{3/2}). \tag{81}$$

By Bayes rule, and taking the exponential of this equation, we obtain

$$\vec{p}_{k+1|k}(x_{k+1}|x_k) := \vec{p}_{k+1|k}(x_{k+1}|x_k) \frac{\vec{p}_k(x_k)}{\vec{p}_{k+1}(x_{k+1})}$$

$$= \frac{\exp\big(-\frac{\|x_k - x_{k+1} + h\vec{b}(x_{k+1}, (k+1)h) - h\vec{\sigma}((k+1)h)^2 \nabla \log p(x_{k+1}, (k+1)h)\|^2}{2h\vec{\sigma}((k+1)h)^2}\big)}{(2\pi h\vec{\sigma}((k+1)h)^2)^{d/2}} + O(h^{3/2}). \tag{82}$$

Up to the $O(h^{3/2})$ term, the right-hand side is the conditional Gaussian corresponding to the update

$$x_k = x_{k+1} + h\big(-\vec{b}(x_{k+1}, (k+1)h) + \vec{\sigma}((k+1)h)^2 \nabla \log p(x_{k+1}, (k+1)h)\big)$$

$$+ \sqrt{h}\vec{\sigma}((k+1)h)\epsilon_{k+1}, \qquad \epsilon_{k+1} \sim N(0, I). \tag{83}$$

If we define $y_k = x_{K-k}$, and we use that $b(x,t) = -\vec{b}(x, 1-t) + \vec{\sigma}(t)^2 \nabla \log p(x, 1-t)$, we can rewrite (83) as

$$
\begin{aligned}
y_{K-k} &= y_{K-k-1} + h\big(-\vec{b}(y_{K-k-1}, (K-k-1)h) + \vec{\sigma}((K-k-1)h)^2 \nabla \log p(y_{K-k-1}, (K-k-1)h)\big) \\
&\quad + \sqrt{h}\vec{\sigma}((K-k-1)h)\epsilon_k = y_{K-k-1} + hb(y_{K-k-1}, kh) + \sqrt{h}\sigma(kh)\epsilon_{K-k-1}, \\
&\implies y_{k+1} = y_k + hb(y_k, kh) + \sqrt{h}\sigma(kh)\epsilon_k.
\end{aligned}
$$
$$(84)$$

And this is the Euler-Maruyama discretization of the backward process $\overleftarrow{X}$. If we plug (82) into (71), we obtain that

$$
\vec{p}(x_{0:K}) \approx \vec{p}_K(x_K) \prod_{k=0}^{K-1} \vec{p}_{k+1|k}(x_{k+1}|x_k). \tag{85}
$$

which concludes the proof, as $\vec{p}_K(x_K)$ is the initial distribution of the backward process, and $\vec{p}_{k+1|k}(x_{k+1}|x_k)$ are its transition kernels.

## C.4 THE RELATIONSHIP BETWEEN THE NOISE PREDICTOR $\epsilon$ AND THE SCORE FUNCTION

Applying Lemma 2 with the choices of $(\kappa_t)_{t\geq0}$ and $(\eta_t)_{t\geq0}$ for DDIM, we obtain that $\vec{X}_t$ has the same distribution as

$$
\hat{X}_t = \sqrt{\bar{\alpha}_{1-t}}\vec{X}_0 + \sqrt{1-\bar{\alpha}_{1-t}}\epsilon, \qquad \epsilon \sim N(0,1). \tag{86}
$$

Since $\vec{X}_t$ and $\hat{X}_t$ have the same distribution, predicting the noise of $\vec{X}_t$ is equivalent to predicting the noise of $\hat{X}_t$. The noise predictor $\epsilon$ can be written as:

$$
\epsilon(x,t) := \mathbb{E}[\epsilon|\hat{X}_{1-t} = x] = \mathbb{E}\big[\epsilon|\sqrt{\bar{\alpha}_t}\vec{X}_0 + \sqrt{1-\bar{\alpha}_t}\epsilon = x\big] = \mathbb{E}\big[\tfrac{x-\sqrt{\bar{\alpha}_t}\vec{X}_0}{\sqrt{1-\bar{\alpha}_t}}|\sqrt{\bar{\alpha}_t}\vec{X}_0 + \sqrt{1-\bar{\alpha}_t}\epsilon = x\big]. \tag{87}
$$

And the score function $\mathfrak{s}(x,t) := \nabla \log \vec{p}_{1-t}(x)$ admits the expression

$$
\mathfrak{s}(x,t) := \nabla \log \vec{p}_{1-t}(x) = \tfrac{\nabla \vec{p}_{1-t}(x)}{\vec{p}_{1-t}(x)} = \tfrac{\nabla \mathbb{E}[\vec{p}_{1-t|0}(x|\vec{X}_0)]}{\vec{p}_{1-t}(x)} = \tfrac{\mathbb{E}[\nabla \log \vec{p}_{1-t|0}(x|\vec{X}_0)\vec{p}_{1-t|0}(x|\vec{X}_0)]}{\vec{p}_{1-t}(x)}, \tag{88}
$$

where

$$
\vec{p}_{1-t|0}(x|\vec{X}_0) = \tfrac{\exp(-\|x-\sqrt{\bar{\alpha}_t}Y_1\|^2/(2(1-\bar{\alpha}_t)))}{(2\pi(1-\bar{\alpha}_t))^{d/2}} \implies \nabla \log \vec{p}_{t|1}(x|Y_1) = -\tfrac{x-\sqrt{\bar{\alpha}_t}Y_1}{1-\bar{\alpha}_t}. \tag{89}
$$

Plugging this into the right-hand side of (88) and using Bayes' rule, we get

$$
\mathfrak{s}(x,t) = \mathbb{E}\big[-\tfrac{x-\sqrt{\bar{\alpha}_t}\vec{X}_0}{1-\bar{\alpha}_t}|\sqrt{\bar{\alpha}_t}\vec{X}_0 + \sqrt{1-\bar{\alpha}_t}\epsilon = x\big]. \tag{90}
$$

Comparing the right-hand sides of (87) and (90), we obtain that $\mathfrak{s}(x,t) = -\tfrac{\epsilon(x,t)}{\sqrt{1-\bar{\alpha}_t}}$.

## C.5 THE RELATIONSHIP BETWEEN THE VECTOR FIELD $v$ AND THE SCORE FUNCTION

By construction (Lipman et al., 2023; Albergo & Vanden-Eijnden, 2023; Albergo et al., 2023), we have that

$$
\begin{aligned}
v(x,t) &= \mathbb{E}[\dot{\alpha}_t Y_1 + \dot{\beta}_t Y_0 | x = \alpha_t Y_1 + \beta_t Y_0] \\
&= \mathbb{E}\big[\tfrac{\dot{\alpha}_t(x-\beta_t Y_0)}{\alpha_t} + \dot{\beta}_t Y_0 | x = \alpha_t Y_1 + \beta_t Y_0\big] \\
&= \tfrac{\dot{\alpha}_t}{\alpha_t}x + (\dot{\beta}_t - \tfrac{\dot{\alpha}_t}{\alpha_t}\beta_t)\mathbb{E}[Y_0 | x = \alpha_t Y_1 + \beta_t Y_0],
\end{aligned} \tag{91}
$$

where we used that $Y_1 = (x - \beta_t Y_0)/\alpha_t$. Also, we can write the score as follows

$$
\mathfrak{s}(x,t) := \nabla \log p_t(x) = \tfrac{\nabla p_t(x)}{p_t(x)} = \tfrac{\nabla \mathbb{E}[p_{t|1}(x|Y_1)]}{p_t(x)} = \tfrac{\mathbb{E}[\nabla p_{t|1}(x|Y_1)]}{p_t(x)} = \tfrac{\mathbb{E}[p_{t|1}(x|Y_1)\nabla \log p_{t|1}(x|Y_1)]}{p_t(x)}, \tag{92}
$$

where

$$
p_{t|1}(x|Y_1) = \tfrac{\exp(-\|x-\alpha_t Y_1\|^2/(2\beta_t^2))}{(2\pi\beta_t^2)^{d/2}} \implies \nabla \log \vec{p}_{t|1}(x|Y_1) = -\tfrac{x-\alpha_t Y_1}{\beta_t^2} \tag{93}
$$

Plugging this back into the right-hand side of (92), we obtain

$$\mathfrak{s}(x,t) = -\frac{\mathbb{E}[p_{t|1}(x|Y_1)\frac{x-\alpha_t Y_1}{\beta_t^2}]}{p_t(x)} = -\frac{\int \vec{p}_{t|1}(x|Y_1)p_1(Y_1)\frac{x-\alpha_t Y_1}{\beta_t^2}\,dY_1}{\vec{p}_t(x)}$$

$$= -\int p_{1|t}(Y_1|x)\frac{x-\alpha_t Y_1}{\beta_t^2}\,dY_1 = -\mathbb{E}[\frac{x-\alpha_t Y_1}{\beta_t^2}|x=\alpha_t Y_1+\beta_t Y_0] = -\frac{\mathbb{E}[Y_0|x=\alpha_t Y_1+\beta_t Y_0]}{\beta_t}$$

(94)

The last equality holds because $(x-\alpha_t Y_1)/\beta_t = Y_0$. Putting together (91) and (94), we obtain that

$$v(x,t) = \frac{\dot{\alpha}_t}{\alpha_t}x + \beta_t(\frac{\dot{\alpha}_t}{\alpha_t}\beta_t - \dot{\beta}_t)\mathfrak{s}(x,t) \iff \mathfrak{s}(x,t) = \frac{1}{\beta_t(\frac{\dot{\alpha}_t}{\alpha_t}\beta_t - \dot{\beta}_t)}\big(v(x,t) - \frac{\dot{\alpha}_t}{\alpha_t}x\big)$$

(95)

Thus, the ODE (3) can be rewritten like this:

$$\frac{\mathrm{d}X_t}{\mathrm{d}t} = \frac{\dot{\alpha}_t}{\alpha_t}X_t + \beta_t(\frac{\dot{\alpha}_t}{\alpha_t}\beta_t - \dot{\beta}_t)\mathfrak{s}(X_t,t), \qquad X_0 \sim p_0.$$

(96)

To allow for an arbitrary diffusion coefficient, we need to add a correction term to the drift:

$$\mathrm{d}X_t = \big(\frac{\dot{\alpha}_t}{\alpha_t}X_t + \big(\frac{\sigma(t)^2}{2} + \beta_t(\frac{\dot{\alpha}_t}{\alpha_t}\beta_t - \dot{\beta}_t)\big)\mathfrak{s}(X_t,t)\big)\mathrm{d}t + \sigma(t)\mathrm{d}B_t, \qquad X_0 \sim p_0.$$

(97)

This can be easily shown by writing down the Fokker-Planck equations for (96) and (97), and observing that they are the same up to a cancellation of terms. Finally, if we plug the right-hand side of (95) into (97), we obtain the SDE for Flow Matching with arbitrary noise schedule (equation (4)).

# D STOCHASTIC OPTIMAL CONTROL AS MAXIMUM ENTROPY RL IN CONTINUOUS SPACE AND TIME

In this section, we bridge KL-regularized (or MaxEnt) reinforcement learning and stochastic optimal control. We show that when the action space is Euclidean and the transition probabilities are conditional Gaussians, taking the limit in which the step size goes to zero on the KL-regularized RL problem gives rise to the SOC problem. A consequence of this connection is that all algorithms for KL-regularized RL admit an analog for diffusion fine-tuning. This is not novel, but it may be useful for researchers that are familiar with RL fine-tuning formulations.

App. D.4 is providing a more direct, rigorous, continuous-time connection between SOC and MaxEnt RL, as it shows that the expected control cost is equal to the KL divergence between the distributions over trajectories, conditioned on the starting points (see equation (12)).

## D.1 MAXIMUM ENTROPY RL

Several diffusion fine-tuning methods (Black et al., 2024; Uehara et al., 2024b) are based on KL-regularized RL, also known as maximum entropy RL, which we review in the following. In the classical reinforcement learning (RL) setting, we have an agent that, starting from state $s_0 \sim p_0$, iteratively observes a state $s_k$, takes an action $a_k$ according to a policy $\pi(a_k; s_k, k)$ which leads to a new state $s_{k+1}$ according to a fixed transition probability $p(s_{k+1}|a_k, s_k)$, and obtains rewards $r_k(s_k, a_k)$. This can be summarized into a trajectory $\tau = ((s_k, a_k))_{k=0}^K$. The goal is to optimize the policy $\pi$ in order to maximize the expected total reward, i.e. $\max_\pi \mathbb{E}_{\tau \sim \pi, p}[\sum_{k=0}^K r_k(s_k, a_k)]$.

Maximum entropy RL (MaxEnt RL; Ziebart et al. (2008)) amounts to adding the entropy $H(\pi)$ of the policy $\pi(\cdot; s_k, k)$ to the reward for each step $k$, in order to encourage exploration and improve robustness to changes in the environment: $\max_\pi \mathbb{E}_{\tau \sim \pi, p}[\sum_{k=0}^K r_k(s_k, a_k) + \sum_{k=0}^{K-1} H(\pi(\cdot; s_k, k))]$ [6]. As a generalization, one can regularize using the negative KL divergence between $\pi(\cdot; s_k, k)$ and a base policy $\pi_{\text{base}}(\cdot; s_k, k)$:

$$\max_\pi \mathbb{E}_{\tau \sim \pi, p}[\sum_{k=0}^K r_k(s_k, a_k) - \sum_{k=0}^{K-1} \text{KL}(\pi(\cdot; s_k, k)||\pi_{\text{base}}(\cdot; s_k, k))],$$

(98)

which prevents the learned policy to deviate too much from the base policy. Each policy $\pi$ induces a distribution $q(\tau)$ over trajectories $\tau$, and the MaxEnt RL problem (98) can be expressed solely in terms of such distributions (Lemma 3 in App. D.3):

$$\max_q \mathbb{E}_{\tau \sim q}[\sum_{k=0}^K r_k(s_k, a_k)] - \text{KL}(q||q^{\text{base}}),$$

(99)

---

[6]The entropy terms are usually multiplied by a factor to tune their magnitude, but one can equivalently rescale the rewards, which is why we do not add any factor.

where $q^{\text{base}}$ is the distribution induced by the base policy $\pi_{\text{base}}$, and the maximization is over all distributions $q$ such that their marginal for $s_0$ is $p_0$. We can further recast this problem as (Lemma 4 in App. D.3):

$$\min_q \text{KL}(q||q^*), \qquad \text{where } q^*(\tau) := q^{\text{base}}(\tau) \exp \left( \sum_{k=0}^K r_k(s_k, a_k) - \mathcal{V}(s_0, 0) \right), \qquad (100)$$

where

$$\mathcal{V}(s_k, k) := \log \left( \mathbb{E}_{\tau \sim \pi_{\text{base}}, p}[\exp \left( \sum_{k'=k}^K r_{k'}(s_{k'}, a_{k'}) \right) | s_k] \right)$$
$$= \max_\pi \mathbb{E}_{\tau \sim \pi, p} \left[ \sum_{k'=k}^K r_{k'}(s_{k'}, a_{k'}) - \sum_{k'=k}^{K-1} \text{KL}(\pi(\cdot; s_{k'}, k')||\pi_{\text{base}}(\cdot; s_{k'}, k')) | s_k \right]$$
$$(101)$$

is the value function. Problem (100) directly implies that the distribution induced by the optimal policy $\pi^*$ is the tilted distribution $q^*$ (which has initial marginal $p_0$).

## D.2 FROM MAXIMUM ENTROPY RL TO STOCHASTIC OPTIMAL CONTROL

The following well-known result, which we prove in App. D.3, shows that in a natural sense, the continuous-time continuous-space version of MaxEnt RL is the SOC framework introduced in Subsec. 3.1. In particular, when states and actions are vectors in $\mathbb{R}^d$, policies are specified by a vector field $u$ (the control), and transition probabilities are conditional Gaussians, the MaxEnt RL problem becomes an SOC problem when the number of timesteps grows to infinity.

**Proposition 5.** *Suppose that*

(i) *The state space and the action space are $\mathbb{R}^d$,*

(ii) *Policies $\pi$ are specified as $\pi(a_k; s_k, k) = \delta(a_k - u(s_k, kh))$, where $u : \mathbb{R}^d \times [0, T] \to \mathbb{R}^d$ is a vector field, and $\delta$ denotes the Dirac delta,*

(iii) *Transition probabilities are conditional Gaussian densities: $p(s_{k+1}|a_k, s_k) = N(s_k + h(b(s_k, kh) + \sigma(kh)a_k), h\sigma(kh)\sigma(kh)^\top)$, where $h = T/K$ is the step size, and $b$ and $\sigma$ are defined as in Subsec. 3.1.*

*Then, in the limit in which the number of steps $K$ grows to infinity, the problem* (98) *is equivalent to the SOC problem* (7)-(8), *identifying*

- *the sequence of states $(s_k)_{k=0}^k$ with the trajectory $\boldsymbol{X}^u = (X_t^u)_{t \in [0,1]}$,*

- *the running reward $\sum_{k=0}^{K-1} r_k(s_k, a_k)$ with the negative running cost $- \int_0^T f(X_t^u, t) \, dt$,*

- *the terminal reward $r_K(s_K, a_K)$ with the negative terminal cost $-g(X_T^u)$,*

- *the KL regularization $\mathbb{E}_{\tau \sim \pi, p}[\sum_{k=0}^{K-1} \text{KL}(\pi(\cdot; s_k, k)||\pi_{\text{base}}(\cdot; s_k, k))]$ with $\frac{1}{2}$ times the expected $L^2$ norm of the control $\frac{1}{2}\mathbb{E}\left[ \int_0^T \|u(X_t^u, t)\|^2 \, dt \right]$,*

- *and the value function $\mathcal{V}(s_k, k)$ defined in* (101) *with the negative value function $-V(x, t)$ defined in Subsec. 3.1.*

A first consequence of this result is that every loss function designed for generic MaxEnt RL problems has a corresponding loss function for SOC problems. The geometric structure of the latter allows for additional losses that do not have an analog in the classical MaxEnt RL setting; in particular, we can differentiate the state and terminal costs.

A second consequence of Prop. 5 is that the characterization (100) can be translated to the SOC setting. The analogs of the distributions $q^*, q^{\text{base}}$ induced by the optimal policy $\pi^*$ and the base policy $\pi^{\text{base}}$ are the distributions $p^*, p^{\text{base}}$ induced by the optimal control $u^*$ and the null control. For an arbitrary trajectory $\boldsymbol{X} = (X_t)_{t \in [0,T]}$, the relation between $\mathbb{P}^*$ and $\mathbb{P}^{\text{base}}$ is given by

$$\frac{d\mathbb{P}^*}{d\mathbb{P}^{\text{base}}}(\boldsymbol{X}) = \exp(- \int_0^T f(X_t, t) \, dt - g(X_T) + V(X_0, 0)) \qquad (102)$$

where $V$ is the value function as defined in Subsec. 3.1. Note that this matches the statement in (16).

### D.3 Proof of Prop. 5: from MaxEnt RL to SOC

Since the transition $p(s_{k+1}|a_k, s_k)$ is fixed, for each $\pi$ we can define

$$\tilde{\pi}(a_k, s_{k+1}; s_k, k) = \pi(a_k; s_k, k)p(s_{k+1}|a_k, s_k)$$
$$\text{and } \tilde{\pi}_{\text{base}}(a_k, s_{k+1}; s_k, k) = \pi_{\text{base}}(a_k; s_k, k)p(s_{k+1}|a_k, s_k), \quad (103)$$

and reexpress (98) as (see Lemma 3)

$$\min_{\tilde{\pi}} \mathbb{E}_{\tau \sim \tilde{\pi}}[\sum_{k=0}^{K} r_k(s_k, a_k) - \sum_{k=0}^{K-1} \text{KL}(\tilde{\pi}(\cdot, \cdot; s_k, k)||\tilde{\pi}_{\text{base}}(\cdot, \cdot; s_k, k))]. \quad (104)$$

Using the hypothesis of the proposition, we can write

$$\tilde{\pi}(a_k, s_{k+1}; s_k, k) = \delta(a_k - u(s_k, k\eta))N(s_k + \eta(b(s_k, k\eta) + \sigma(k\eta)a_k), \eta\sigma(k\eta)\sigma(k\eta)^\top)$$
$$= \delta(a_k - u(s_k, k\eta))\tilde{\pi}(s_{k+1}; s_k, k), \quad (105)$$

where $\tilde{\pi}(s_{k+1}; s_k, k) = N(s_k + \eta(b(s_k, k\eta) + \sigma(k\eta)u(s_k, k\eta)), \eta\sigma(k\eta)\sigma(k\eta)^\top)$ is the state transition kernel. We set the base policy as $\pi_{\text{base}}(a_k; s_k, k) = \delta(a_k)$, and we obtain analogously that $\tilde{\pi}(a_k, s_{k+1}; s_k, k) = \delta(a_k)\tilde{\pi}_{\text{base}}(s_{k+1}; s_k, k)$ with $\tilde{\pi}_{\text{base}}(s_{k+1}; s_k, k) = N(s_k + \eta b(s_k, k\eta), \eta\sigma(k\eta)\sigma(k\eta)^\top)$. Now, if we take $K$ large, the trajectory $(s_k)_{k=0}^{K}$ generated by $\tilde{\pi}$ can be regarded as the Euler-Maruyama discretization of a solution $X^u$ of the controlled SDE (8), while the trajectory generated by $\tilde{\pi}_{\text{base}}$ is the discretization of the uncontrolled process $X^0$ obtained by setting $u = 0$. As a consequence

$$\lim_{K \to \infty} \mathbb{E}_{\tau \sim \tilde{\pi}}[\sum_{k=0}^{K-1} \text{KL}(\tilde{\pi}(\cdot, \cdot; s_k, k)||\tilde{\pi}_{\text{base}}(\cdot, \cdot; s_k, k))]$$
$$= \lim_{K \to \infty} \mathbb{E}_{\tau \sim \tilde{\pi}}[\sum_{k=0}^{K-1} \text{KL}(\tilde{\pi}(\cdot; s_k, k)||\tilde{\pi}_{\text{base}}(\cdot; s_k, k))] = \mathbb{E}_{X^u \sim \mathbb{P}^u}[\log \frac{d\mathbb{P}^u}{d\mathbb{P}^0}(X^u)], \quad (106)$$

where $\mathbb{P}^u$ and $\mathbb{P}^0$ are the measures of the processes $X^u$ and $X^0$, respectively. The Girsanov theorem (Thm. 2) implies that $\log \frac{d\mathbb{P}^u}{d\mathbb{P}^0}(X^u) = -\int_0^T \langle u(X_t^u, t), dB_t \rangle - \frac{1}{2}\int_0^T \|u(X_t^u, t)\|^2 dt$, which implies that $\mathbb{E}_{X^u \sim \mathbb{P}^u}[\log \frac{d\mathbb{P}^u}{d\mathbb{P}^0}(X^u)] = -\frac{1}{2}\mathbb{E}_{X^u \sim \mathbb{P}^u}[\int_0^T \|u(X_t^u, t)\|^2 dt]$. Setting the rewards $r_k(a_k, s_k) = \eta f(s_k, k\eta)$ for $k \in \{0, \ldots, K-1\}$ and $r_K(a_K, s_K) = \eta g(s_k)$, where $f$ and $g$ are as in Subsec. 3.1, yields the following limiting object:

$$\lim_{K \to \infty} \mathbb{E}_{\tau \sim \tilde{\pi}}[\sum_{k=0}^{K} r_k(s_k, a_k)] = \mathbb{E}_{X^u \sim \mathbb{P}^u}[\int_0^T f(X_t^u, t) dt + g(X_T^u)]. \quad (107)$$

Hence, the limit of the MaxEnt RL loss (104) is the SOC loss (7).

**Lemma 3.** *Let* $\tilde{\pi}(a_k, s_{k+1}; s_k, k)$ *and* $\tilde{\pi}_{\text{base}}(a_k, s_{k+1}; s_k, k)$ *be as defined in* (103). $\text{KL}(\tilde{\pi}(\cdot, \cdot; s_k, k)||\tilde{\pi}_{\text{base}}(\cdot, \cdot; s_k, k))]$ *and* $\text{KL}(\pi(\cdot; s_k, k)||\pi_{\text{base}}(\cdot; s_k, k))]$ *are equal. Moreover, if* $q, q^{\text{base}}$ *denote the distributions over trajectories induced by* $\pi, \pi_{\text{base}}$, *we have that*

$$\text{KL}(q||q^{\text{base}}) = \mathbb{E}[\sum_{k=0}^{K-1} \text{KL}(\pi(\cdot; s_k, k)||\pi_{\text{base}}(\cdot; s_k, k))]. \quad (108)$$

*Proof.* We have that

$$\text{KL}(\tilde{\pi}(\cdot, \cdot; s_k, k)||\tilde{\pi}_{\text{base}}(\cdot, \cdot; s_k, k))] = \sum_{a_k, s_{k+1}} \tilde{\pi}(a_k, s_{k+1}; s_k, k) \log \frac{\tilde{\pi}(a_k, s_{k+1}; s_k, k)}{\tilde{\pi}_{\text{base}}(a_k, s_{k+1}; s_k, k)}$$
$$= \sum_{a_k, s_{k+1}} \pi(a_k; s_k, k)p(s_{k+1}|a_k, s_k) \log \frac{\pi(a_k; s_k, k)p(s_{k+1}|a_k, s_k)}{\pi_{\text{base}}(a_k; s_k, k)p(s_{k+1}|a_k, s_k)}$$
$$= \sum_{a_k, s_{k+1}} \pi(a_k; s_k, k)p(s_{k+1}|a_k, s_k) \log \frac{\pi(a_k; s_k, k)}{\pi_{\text{base}}(a_k; s_k, k)} \quad (109)$$
$$= \sum_{a_k} \pi(a_k; s_k, k)\big(\sum_{s_{k+1}} p(s_{k+1}|a_k, s_k)\big) \log \frac{\pi(a_k; s_k, k)}{\pi_{\text{base}}(a_k; s_k, k)}$$
$$= \sum_{a_k} \pi(a_k; s_k, k) \log \frac{\pi(a_k; s_k, k)}{\pi_{\text{base}}(a_k; s_k, k)} = \text{KL}(\pi(\cdot; s_k, k)||\pi_{\text{base}}(\cdot; s_k, k)).$$

To prove (108), by construction we can write

$$q(\tau) = p_0(s_0)\prod_{k=0}^{K-1} \tilde{\pi}(a_k, s_{k+1}; s_k, k), \qquad q^{\text{base}}(\tau) = p_0(s_0)\prod_{k=0}^{K-1} \tilde{\pi}_{\text{base}}(a_k, s_{k+1}; s_k, k),$$
$$\quad (110)$$

which means that

$$
\begin{aligned}
\mathrm{KL}(q||q^{\mathrm{base}}) &= \mathbb{E}_{\tau \sim q}[\log \tfrac{q(\tau)}{q^{\mathrm{base}}(\tau)}] = \mathbb{E}_{\tau \sim q}[\textstyle\sum_{k=0}^{K-1} \log \tfrac{\tilde{\pi}(a_k, s_{k+1}; s_k, k)}{\tilde{\pi}_{\mathrm{base}}(a_k, s_{k+1}; s_k, k)}] \\
&= \textstyle\sum_{k=0}^{K-1} \mathbb{E}_{\tau \sim q^{0:(k+1)}}[\log \tfrac{\tilde{\pi}(a_k, s_{k+1}; s_k, k)}{\tilde{\pi}_{\mathrm{base}}(a_k, s_{k+1}; s_k, k)}] \\
&= \textstyle\sum_{k=0}^{K-1} \mathbb{E}_{\tau \sim q^{0:k}}[\textstyle\sum_{a_k, s_{k+1}} \tilde{\pi}(a_k, s_{k+1}; s_k, k) \log \tfrac{\tilde{\pi}(a_k, s_{k+1}; s_k, k)}{\tilde{\pi}_{\mathrm{base}}(a_k, s_{k+1}; s_k, k)}] \\
&= \textstyle\sum_{k=0}^{K-1} \mathbb{E}_{\tau \sim q^{0:k}}[\mathrm{KL}(\tilde{\pi}(\cdot, \cdot; s_k, k)||\tilde{\pi}_{\mathrm{base}}(\cdot, \cdot; s_k, k))] \\
&= \textstyle\sum_{k=0}^{K-1} \mathbb{E}_{\tau \sim q^{0:k}}[\mathrm{KL}(\pi(\cdot; s_k, k)||\pi_{\mathrm{base}}(\cdot; s_k, k))] \\
&= \mathbb{E}_{\tau \sim q^{0:k}}[\textstyle\sum_{k=0}^{K-1} \mathrm{KL}(\pi(\cdot; s_k, k)||\pi_{\mathrm{base}}(\cdot; s_k, k))]
\end{aligned}
\tag{111}
$$

Here, the notation $q^{0:k}$ denotes the trajectory $q$ up to the state $s_k$. $\qquad\square$

**Lemma 4.** *The distribution-based MaxEnt RL formulation in* (99) *is equivalent to the the following problem:*

$$
\min_q \mathrm{KL}(q||q^*), \qquad \text{where } q^*(\tau) := \frac{q^{\mathrm{base}}(\tau) \exp\left(\sum_{k=0}^K r_k(s_k, a_k)\right)}{\frac{1}{p_0(s_0)} \sum_{\{\tau'|s_0'=s_0\}} q^{\mathrm{base}}(\tau') \exp\left(\sum_{k=0}^K r_k(s_k', a_k')\right)},
\tag{112}
$$

*where the minimization is over $q$ with marginal $p_0$ at step zero. The optimum of the problem is $q^*$, which satisfies the marginal constraint. The following alternative characterization of $q^*$ holds:*

$$
q^*(\tau) = q^{\mathrm{base}}(\tau) \exp\left(\textstyle\sum_{k=0}^K r_k(s_k, a_k) - \mathcal{V}(s_0, 0)\right),
\tag{113}
$$

*where $\mathcal{V}(x, k) = \max_\pi \mathbb{E}_{\tau \sim \pi, p}\left[\sum_{k'=k}^K r_{k'}(s_{k'}, a_{k'}) - \sum_{k'=k}^{K-1} \mathrm{KL}(\pi(\cdot; s_{k'}, k')||\pi_{\mathrm{base}}(\cdot; s_{k'}, k'))|s_k = x\right]$.*
$$\tag{114}$$

*Proof.* Let us expand $\mathrm{KL}(q||q^*)$:

$$
\begin{aligned}
\mathrm{KL}(q||q^*) &= \mathbb{E}_{\tau \sim q}\left[\log \tfrac{q(\tau)}{q^*(\tau)}\right] \\
&= \mathbb{E}_{\tau \sim q}\big[\log q(\tau) - \log q^{\mathrm{base}}(\tau) - \textstyle\sum_{k=0}^K r_k(s_k, a_k) \\
&\qquad + \log\left(\tfrac{1}{p_0(s_0)} \textstyle\sum_{\{\tau'|s_0'=s_0\}} q^{\mathrm{base}}(\tau') \exp\left(\textstyle\sum_{k=0}^K r_k(s_k', a_k')\right)\right)\big] \\
&= \mathrm{KL}(q||q^{\mathrm{base}}) - \mathbb{E}_{\tau \sim q}\big[\textstyle\sum_{k=0}^K r_k(s_k, a_k)\big] \\
&\qquad + \mathbb{E}_{s_0 \sim p_0}\big[\log\left(\tfrac{1}{p_0(s_0)} \textstyle\sum_{\{\tau'|s_0'=s_0\}} q^{\mathrm{base}}(\tau') \exp\left(\textstyle\sum_{k=0}^K r_k(s_k', a_k')\right)\right)\big],
\end{aligned}
\tag{115}
$$

where the third equality holds because the marginal of $q$ at step zero is $p_0$ by hypothesis. Since the third term in the right-hand side is independent of $q$, this proves the equivalence between (99) and (112).

Next, we prove that the marginal of $q^*$ at step zero is $p_0$:

$$
\textstyle\sum_{\{\tau|s_0=x\}} q^*(\tau) := \textstyle\sum_{\{\tau|s_0=x\}} \frac{q^{\mathrm{base}}(\tau) \exp\left(\sum_{k=0}^K r_k(s_k, a_k)\right)}{\frac{1}{p_0(x)} \sum_{\{\tau'|s_0'=x\}} q^{\mathrm{base}}(\tau') \exp\left(\sum_{k=0}^K r_k(s_k', a_k')\right)} = p_0(x).
\tag{116}
$$

Now, for an arbitrary $s_0$, let $q_{s_0}, q_{s_0}^*$ be the distributions $q, q^*$ conditioned on the initial state being $s_0$. We can write an analog to equation (115) for $q_{s_0}, q_{s_0}^*$:

$$
\begin{aligned}
\mathrm{KL}(q_{s_0}||q_{s_0}^*) &= \mathbb{E}_{\tau \sim q_{s_0}}\left[\log \tfrac{q_{s_0}(\tau)}{q_{s_0}^*(\tau)}\right] \\
&= \mathbb{E}_{\tau \sim q_{s_0}}\big[\log q_{s_0}(\tau) - \log q_{s_0}^{\mathrm{base}}(\tau) - \textstyle\sum_{k=0}^K r_k(s_k, a_k) \\
&\qquad + \log\left(\tfrac{1}{p_0(s_0)} \textstyle\sum_{\{\tau'|s_0'=s_0\}} q_{s_0}^{\mathrm{base}}(\tau') \exp\left(\textstyle\sum_{k=0}^K r_k(s_k', a_k')\right)\right)\big] \\
&= \mathrm{KL}(q_{s_0}||q_{s_0}^{\mathrm{base}}) - \mathbb{E}_{\tau \sim q_{s_0}}\big[\textstyle\sum_{k=0}^K r_k(s_k, a_k)\big] \\
&\qquad + \log\left(\tfrac{1}{p_0(s_0)} \textstyle\sum_{\{\tau'|s_0'=s_0\}} q^{\mathrm{base}}(\tau') \exp\left(\textstyle\sum_{k=0}^K r_k(s_k', a_k')\right)\right),
\end{aligned}
\tag{117}
$$

Hence,

$$
\begin{aligned}
0 = \min_{q_{s_0}} \mathrm{KL}(q_{s_0}||q_{s_0}^*) &= -\max_{q_{s_0}}\{\mathbb{E}_{\tau \sim q_{s_0}}\big[\textstyle\sum_{k=0}^K r_k(s_k, a_k)\big] - \mathrm{KL}(q_{s_0}||q_{s_0}^{\mathrm{base}})\} \\
&\qquad + \log\left(\tfrac{1}{p_0(s_0)} \textstyle\sum_{\{\tau'|s_0'=s_0\}} q^{\mathrm{base}}(\tau') \exp\left(\textstyle\sum_{k=0}^K r_k(s_k', a_k')\right)\right).
\end{aligned}
\tag{118}
$$

And applying (108) from (108), we obtain that

$$\log\big(\tfrac{1}{p_0(s_0)}\textstyle\sum_{\{\tau'|s_0'=s_0\}} q^{\mathrm{base}}(\tau')\exp\big(\textstyle\sum_{k=0}^K r_k(s_k',a_k')\big)\big)$$
$$= \max_\pi \mathbb{E}_{\tau\sim\pi,p}\big[\textstyle\sum_{k=0}^K r_k(s_k,a_k) - \sum_{k=0}^{K-1}\mathrm{KL}(\pi(\cdot;s_k,k)||\pi_{\mathrm{base}}(\cdot;s_k,k))|s_0\big] = \mathcal{V}(s_0,0), \tag{119}$$

which concludes the proof. $\qquad\square$

### D.4 PROOF OF EQUATION (12): THE CONTROL COST IS A KL REGULARIZER

**Theorem 2** (Girsanov theorem for SDEs). *If the two SDEs*

$$\mathrm{d}X_t = b_1(X_t,t)\,\mathrm{d}t + \sigma(X_t,t)\,\mathrm{d}B_t, \qquad X_0 = x_{\mathrm{init}} \tag{120}$$
$$dY_t = (b_1(Y_t,t) + b_2(Y_t,t))\,\mathrm{d}t + \sigma(Y_t,t)\,\mathrm{d}B_t, \qquad Y_0 = x_{\mathrm{init}} \tag{121}$$

*admit unique strong solutions on $[0,T]$, then for any bounded continuous functional $\Phi$ on $C([0,T])$, we have that*

$$\mathbb{E}[\Phi(\boldsymbol{X})] = \mathbb{E}\big[\Phi(\boldsymbol{Y})\exp\big(-\textstyle\int_0^T \sigma(Y_t,t)^{-1}b_2(Y_t,t)\,\mathrm{d}B_t - \tfrac{1}{2}\int_0^T \|\sigma(Y_t,t)^{-1}b_2(Y_t,t)\|^2\,\mathrm{d}t\big)\big]$$
$$= \mathbb{E}\big[\Phi(\boldsymbol{Y})\exp\big(-\textstyle\int_0^T \sigma(Y_t,t)^{-1}b_2(Y_t,t)\,d\tilde{B}_t + \tfrac{1}{2}\int_0^T \|\sigma(Y_t,t)^{-1}b_2(Y_t,t)\|^2\,\mathrm{d}t\big)\big], \tag{122}$$

*where $\tilde{B}_t = B_t + \int_0^t \sigma(Y_s,s)^{-1}b_2(Y_s,s)\,\mathrm{d}s$. More generally, $b_1$ and $b_2$ can be random processes that are adapted to filtration of $\boldsymbol{B}$.*

Consider the SDEs

$$\mathrm{d}X_t = b(X_t,t)\,\mathrm{d}t + \sigma(t)\mathrm{d}B_t, \qquad\qquad X_0 = x_0, \tag{123}$$
$$\mathrm{d}X_t^u = \big(b(X_t^u,t) + \sigma(t)u(X_t^u,t)\big)\,\mathrm{d}t + \sigma(t)\mathrm{d}B_t, \qquad\qquad X_0^u = x_0. \tag{124}$$

If we let $\mathbb{P}|_{x_0}$, $\mathbb{P}^u|_{x_0}$ be the probability measures of the solutions of (123) and (124), Thm. 2 implies that

$$\log\frac{\mathrm{d}\mathbb{P}|_{x_0}}{\mathrm{d}\mathbb{P}^u|_{x_0}}(\boldsymbol{X}^u) = -\textstyle\int_0^1 u(X_t^u,t)\,\mathrm{d}B_t - \tfrac{1}{2}\int_0^1 \|u(X_t^u,t)\|^2\,\mathrm{d}t. \tag{125}$$

Hence,

$$D_{\mathrm{KL}}\big(\mathbb{P}^u|_{x_0}\,\big\|\,\mathbb{P}|_{x_0}\big) = \mathbb{E}\big[\log\tfrac{\mathrm{d}\mathbb{P}^u|_{x_0}}{\mathrm{d}\mathbb{P}|_{x_0}}(\boldsymbol{X}^u)|X_0^u = x_0\big] = -\mathbb{E}\big[\log\tfrac{\mathrm{d}\mathbb{P}|_{x_0}}{\mathrm{d}\mathbb{P}^u|_{x_0}}(\boldsymbol{X}^u)|X_0^u = x_0\big]$$
$$= \mathbb{E}\big[\textstyle\int_0^1 u(X_t^u,t)\,\mathrm{d}B_t + \tfrac{1}{2}\int_0^1 \|u(X_t^u,t)\|^2\,\mathrm{d}t|X_0^u = x_0\big] \tag{126}$$
$$= \mathbb{E}\big[\tfrac{1}{2}\textstyle\int_0^1 \|u(X_t^u,t)\|^2\,\mathrm{d}t|X_0^u = x_0\big],$$

where we used that stochastic integrals are martingales.

## E PROOFS OF SUBSEC. 3.3: MEMORYLESS NOISE SCHEDULE AND FINE-TUNING RECIPE

### E.1 PROOF OF PROP. 1: THE MEMORYLESS NOISE SCHEDULE

We consider the forward-backward SDEs (50)-(51) with arbitrary noise schedule. By Prop. 4, the trajectories $\vec{\boldsymbol{X}}$, $\boldsymbol{X}$ of these two processes are equally distributed up to a time flip, which also means that their marginals satisfy $\vec{p}_t = p_{1-t}$, for all $t \in [0,1]$. First, we develop an explicit expression for the score function $s(x,t) = \nabla\log p_t(x)$. By the properties of flow matching, we know that $p_t$ is the distribution of the interpolation variable $\bar{X}_t = \beta_t\bar{X}_0 + \alpha_t\bar{X}_1$, where $\bar{X}_0 \sim N(0,I)$, $\bar{X}_1 \sim p^{\mathrm{data}}$ are independent. Thus, $\frac{\bar{X}_t - \alpha_t\bar{X}_1}{\beta_t} \sim N(0,\mathrm{I})$, which means that we can express the density $p_t$ as

$$p_t(x) = \int_{\mathbb{R}^d} \frac{\exp\big(-\frac{\|x-\alpha_t y\|^2}{2\beta_t^2}\big)}{(2\pi\beta_t^2)^{d/2}} p^{\mathrm{data}}(y)\,\mathrm{d}y. \tag{127}$$

Thus,

$$s(x,t) = \nabla \log p_t(x) = -\frac{x}{\beta_t^2} + \frac{\alpha_t}{\beta_t^2} \frac{\int_{\mathbb{R}^d} y \exp\left(-\frac{\|x-\alpha_t y\|^2}{2\beta_t^2}\right) p^{\text{data}}(y)\, dy}{\int_{\mathbb{R}^d} \exp\left(-\frac{\|x-\alpha_t y\|^2}{2\beta_t^2}\right) p^{\text{data}}(y)\, dy} := -\frac{x - \alpha_t \xi_t(x)}{\beta_t^2}, \qquad (128)$$

where we defined

$$\xi_t(x) = \frac{\int_{\mathbb{R}^d} y \exp\left(-\frac{\|x-\alpha_t y\|^2}{2\beta_t^2}\right) p^{\text{data}}(y)\, dy}{\int_{\mathbb{R}^d} \exp\left(-\frac{\|x-\alpha_t y\|^2}{2\beta_t^2}\right) p^{\text{data}}(y)\, dy}. \qquad (129)$$

Hence, we can rewrite the forward SDE (50) as

$$d\vec{X}_t = \left(-\kappa_{1-t}\vec{X}_t - \left(\frac{\sigma(1-t)^2}{2} - \eta_{1-t}\right)\frac{\vec{X}_t - \alpha_{1-t}\xi_{1-t}(\vec{X}_t)}{\beta_{1-t}^2}\right) dt + \sigma(1-t)\, dB_t, \qquad \vec{X}_0 \sim p_{\text{data}} \qquad (130)$$

Hence, if we substitute $\kappa_{1-t} \leftarrow \kappa_{1-t} + \frac{\sigma(1-t)^2 - 2\eta_{1-t}}{2\beta_{1-t}^2}$, $\xi_{1-t} \leftarrow \frac{\alpha_{1-t}(\sigma(1-t)^2 - 2\eta_{1-t})}{2\beta_{1-t}^2}\xi_{1-t}(\vec{X}_t)$ (where we ignore the dependency on $\vec{X}_t$), $\sqrt{2\eta_{1-t}} \leftarrow \sigma(1-t)$, we can apply Lemma 2, which yields

$$\vec{X}_t = \vec{X}_0 \exp\left(-\int_0^t \left(\kappa_{1-s} + \frac{\sigma(1-s)^2 - 2\eta_{1-s}}{2\beta_{1-s}^2}\right) ds\right)$$
$$+ \int_0^t \exp\left(-\int_{t'}^t \left(\kappa_{1-s} + \frac{\sigma(1-s)^2 - 2\eta_{1-s}}{2\beta_{1-s}^2}\right) ds\right)\frac{\alpha_{1-t'}(\sigma(1-t')^2 - 2\eta_{1-t'})}{2\beta_{1-t'}^2}\xi_{1-t'}(\vec{X}_{t'})\, dt' \quad (131)$$
$$+ \int_0^t \sigma(1-t')\exp\left(-\int_{t'}^t \left(\kappa_{1-s} + \frac{\sigma(1-s)^2 - 2\eta_{1-s}}{2\beta_{1-s}^2}\right) ds\right) dB_{t'}.$$

We simplify the recurring expression:

$$\kappa_{1-s} + \frac{\sigma(1-s)^2 - 2\eta_{1-s}}{2\beta_{1-s}^2} = \frac{\dot{\alpha}_{1-s}}{\alpha_{1-s}} + \frac{\sigma(1-s)^2 - 2\beta_{1-s}\left(\frac{\dot{\alpha}_{1-s}}{\alpha_{1-s}}\beta_{1-s} - \dot{\beta}_{1-s}\right)}{2\beta_{1-s}^2} = \frac{\sigma(1-s)^2}{2\beta_{1-s}^2} + \frac{\dot{\beta}_{1-s}}{\beta_{1-s}} \qquad (132)$$

Thus,

$$\int_{t'}^t \left(\kappa_{1-s} + \frac{\sigma(1-s)^2 - 2\eta_{1-s}}{2\beta_{1-s}^2}\right) ds = \int_{t'}^t \left(\frac{\sigma(1-s)^2}{2\beta_{1-s}^2} - \partial_s \log \beta_{1-s}\right) ds = \int_{t'}^t \frac{\sigma(1-s)^2}{2\beta_{1-s}^2}\, ds - \left(\log \beta_{1-t} - \log \beta_{1-t'}\right), \qquad (133)$$

which means that

$$\exp\left(-\int_{t'}^t \left(\kappa_{1-s} + \frac{\sigma(1-s)^2 - 2\eta_{1-s}}{2\beta_{1-s}^2}\right) ds\right) = \exp\left(-\int_{t'}^t \frac{\sigma(1-s)^2}{2\beta_{1-s}^2}\, ds\right)\frac{\beta_{1-t}}{\beta_{1-t'}}, \qquad (134)$$

$$\frac{\alpha_{1-t'}(\sigma(1-t')^2 - 2\eta_{1-t'})}{2\beta_{1-t'}^2}\xi_{1-t'}(\vec{X}_{t'}) = \alpha_{1-t'}\left(\frac{\sigma(1-t')^2}{2\beta_{1-t'}^2} + \frac{\dot{\beta}_{1-t'}}{\beta_{1-t'}} - \frac{\dot{\alpha}_{1-t'}}{\alpha_{1-t'}}\right)\xi_{1-t'}(\vec{X}_{t'}). \quad (135)$$

If we define $\chi(1-s)$ such that $\sigma^2(1-s) = 2\beta_{1-s}\left(\frac{\dot{\alpha}_{1-s}}{\alpha_{1-s}}\beta_{1-s} - \dot{\beta}_{1-s}\right) + \chi(1-s)$, we obtain that

$$\exp\left(-\int_{t'}^t \frac{\sigma(1-s)^2}{2\beta_{1-s}^2}\, ds\right)\frac{\beta_{1-t}}{\beta_{1-t'}} = \exp\left(-\int_{t'}^t \left(\frac{\dot{\alpha}_{1-s}}{\alpha_{1-s}} - \frac{\dot{\beta}_{1-s}}{\beta_{1-s}} + \frac{\chi(1-s)}{2\beta_{1-s}^2}\right) ds\right)\frac{\beta_{1-t}}{\beta_{1-t'}}$$
$$= \exp\left(\int_{t'}^t \left(\partial_s \log \alpha_{1-s} - \partial_s \log \beta_{1-s} - \frac{\chi(1-s)}{2\beta_{1-s}^2}\right) ds\right)\frac{\beta_{1-t}}{\beta_{1-t'}} = \exp\left(-\int_{t'}^t \frac{\chi(1-s)}{2\beta_{1-s}^2}\, ds\right)\frac{\alpha_{1-t}}{\alpha_{1-t'}}, \qquad (136)$$

$$\alpha_{1-t'}\left(\frac{\sigma(1-t')^2}{2\beta_{1-t'}^2} + \frac{\dot{\beta}_{1-t'}}{\beta_{1-t'}} - \frac{\dot{\alpha}_{1-t'}}{\alpha_{1-t'}}\right)\xi_{1-t'}(\vec{X}_{t'}) = \frac{\alpha_{1-t'}\chi(1-t')}{2\beta_{1-t'}^2}\xi_{1-t'}(\vec{X}_{t'}) \qquad (137)$$

If we plug equations (136)-(137) into (134)-(135), and then those into (131), we obtain that

$$\vec{X}_t = \vec{X}_0 \exp\left(-\int_0^t \frac{\chi(1-s)}{2\beta_{1-s}^2}\, ds\right)\frac{\alpha_{1-t}}{\alpha_1} + \alpha_{1-t}\int_0^t \exp\left(-\int_{t'}^t \frac{\chi(1-s)}{2\beta_{1-s}^2}\, ds\right)\frac{\chi(1-t')}{2\beta_{1-t'}^2}\xi_{1-t'}(\vec{X}_{t'})\, dt'$$
$$+ \int_0^t \sqrt{2\beta_{1-t'}\left(\frac{\dot{\alpha}_{1-t'}}{\alpha_{1-t'}}\beta_{1-t'} - \dot{\beta}_{1-t'}\right) + \chi(1-t')}\exp\left(-\int_{t'}^t \frac{\chi(1-s)}{2\beta_{1-s}^2}\, ds\right)\frac{\alpha_{1-t}}{\alpha_{1-t'}}\, dB_{t'}. \qquad (138)$$

and if we take the limit $t \to 1^-$ and use that $\alpha_1 = 1$,

$$\vec{X}_1 = \vec{X}_0 \big( \lim_{t \to 1^-} \exp \big( - \int_0^t \tfrac{\chi(1-s)}{2\beta_{1-s}^2} \, \mathrm{d}s \big) \alpha_{1-t} \big) + \lim_{t \to 1^-} \alpha_{1-t} \int_0^t \exp \big( - \int_{t'}^t \tfrac{\chi(1-s)}{2\beta_{1-s}^2} \, \mathrm{d}s \big) \tfrac{\chi(1-t')}{2\beta_{1-t'}^2} \xi_{1-t'}(\vec{X}_{t'}) \, \mathrm{d}t'$$
$$+ \lim_{t \to 1^-} \int_0^t \big( 2\beta_{1-t'} \big( \tfrac{\dot{\alpha}_{1-t'}}{\alpha_{1-t'}} \beta_{1-t'} - \dot{\beta}_{1-t'} \big) + \chi(1-t') \big) \exp \big( - \int_{t'}^t \tfrac{\chi(1-s)}{2\beta_{1-s}^2} \, \mathrm{d}s \big) \tfrac{\alpha_{1-t}}{\alpha_{1-t'}} \, \mathrm{d}B_{t'}.$$

$$(139)$$

The assumption on $\chi$ in (19) is equivalent, up to a rearrangement of the notation and a flip in the time variable, to the statement that for all $t' \in [0,1)$,

$$\lim_{t \to 1^-} \exp \big( - \int_{t'}^t \tfrac{\chi(1-s)}{2\beta_{1-s}^2} \, \mathrm{d}s \big) \alpha_{1-t} = 0. \tag{140}$$

Hence, under assumption (19), the factor accompanying $\vec{X}_0$ in equation (139) is zero. Moreover, this assumption also implies that

$$\lim_{t \to 1^-} \alpha_{1-t} \int_0^t \exp \big( - \int_{t'}^t \tfrac{\chi(1-s)}{2\beta_{1-s}^2} \, \mathrm{d}s \big) \tfrac{\chi(1-t')}{2\beta_{1-t'}^2} \xi_{1-t'}(\vec{X}_{t'}) \, \mathrm{d}t'$$
$$= \int_0^1 \big( \lim_{t \to 1^-} \exp \big( - \int_{t'}^t \tfrac{\chi(1-s)}{2\beta_{1-s}^2} \, \mathrm{d}s \big) \alpha_{1-t} \big) \tfrac{\chi(1-t')}{2\beta_{1-t'}^2} \xi_{1-t'}(\vec{X}_{t'}) \, \mathrm{d}t' = 0. \tag{141}$$

If we plug (140) and (141) into (139), we obtain that

$$\vec{X}_1 = \lim_{t \to 1^-} \int_0^t \big( 2\beta_{1-t'} \big( \tfrac{\dot{\alpha}_{1-t'}}{\alpha_{1-t'}} \beta_{1-t'} - \dot{\beta}_{1-t'} \big) + \chi(1-t') \big) \exp \big( - \int_{t'}^t \tfrac{\chi(1-s)}{2\beta_{1-s}^2} \, \mathrm{d}s \big) \tfrac{\alpha_{1-t}}{\alpha_{1-t'}} \, \mathrm{d}B_{t'},$$

$$(142)$$

which shows that $\vec{X}_1$ is independent of $\vec{X}_0$. Next, we leverage that $\vec{X}$ and $X$ have equal distributions over trajectories (Prop. 4). In particular, the joint distribution of $(\vec{X}_0, \vec{X}_1)$ is equal to the joint distribution of $(X_1, X_0)$. We conclude that $X_1$ and $X_0$ are independent, which is the definition of the memorylessness property. Hence, the assumption (19) is sufficient for memorylessness to hold.

It remains to prove that the assumption (19) is necessary. Looking at equation (138) we deduce that generally, for any $t \in [0,1)$, $\vec{X}_0$ and $\vec{X}_t$ are not independent, because the first two terms in (138) are different from zero. Thus, if there existed a $t' \in [0,1)$ such that the limit (140) is different from zero, then $\vec{X}_1$ would not be independent from $\vec{X}_{t'}$, which means that in general it would not be independent of $\vec{X}_0$ either.

### E.2 FINE-TUNING RECIPE FOR GENERAL NOISE SCHEDULES

#### E.2.1 EXPRESSING $b$, $u$ IN TERMS OF $v$ OR $\epsilon$

Before proving the result we clarify the relation between the DDIM and Flow Matching vector fields $\epsilon$, $v$, and the base drift $b$ and the control $u$. Let $\epsilon^{\text{base}}$, $v^{\text{base}}$ denote the pre-trained vector fields and $\epsilon^{\text{finetune}}$, $v^{\text{finetune}}$ the fine-tuned vector fields. Then we have the following expressions for the full drift $b(x,t) + \sigma(t)u(x,t)$ and control $u(x,t)$ when $\sigma(t) = \sqrt{2\eta_t}$:

*DDIM / DDPM*:

$$b(x,t) + \sigma(t)u(x,t) = \tfrac{\dot{\bar{\alpha}}_t}{2\bar{\alpha}_t} x - \tfrac{\dot{\bar{\alpha}}_t}{\bar{\alpha}_t} \tfrac{\epsilon^{\text{finetune}}(x,t)}{\sqrt{1-\bar{\alpha}_t}}, \quad u(x,t) = -\sqrt{\tfrac{\dot{\bar{\alpha}}_t}{\bar{\alpha}_t(1-\bar{\alpha}_t)}} \big( \epsilon^{\text{finetune}}(x,t) - \epsilon^{\text{base}}(x,t) \big).$$

$$(143)$$

*Memoryless Flow Matching*:

$$b(x,t) + \sigma(t)u(x,t) = 2v^{\text{finetune}}(x,t) - \tfrac{\dot{\alpha}_t}{\alpha_t} x, \quad u(x,t) = \sqrt{\tfrac{2}{\beta_t \left( \tfrac{\dot{\alpha}_t}{\alpha_t} \beta_t - \dot{\beta}_t \right)}} \big( v^{\text{finetune}}(x,t) - v^{\text{base}}(x,t) \big).$$

$$(144)$$

Thus, to solve the SOC problem (7)-(8) in practice, we parameterize the control $u$ in terms of $\epsilon^{\text{finetune}}$ or $v^{\text{finetune}}$ and optimize these vector fields instead. After plugging in (143)-(144), the SOC problem (7)-(8) can then be solved using any SOC algorithm in order to perform fine-tuning, and we propose an especially effective algorithm in Sec. 4: Adjoint Matching. After fine-tuning, $\epsilon^{\text{finetune}}$ and $v^{\text{finetune}}$ can simply be plugged back into their respective generative processes (3)-(30) to sample from the tilted distribution (1) using any choice of diffusion coefficient.

### E.2.2 PROOF OF THM. 1

The proof of Thm. 1 relies heavily on the properties of the Hamilton-Jacobi-Bellman equation:

**Theorem 3** (Hamilton-Jacobi-Bellman equation). *If we define the infinitesimal generator*

$$\mathcal{L} := \tfrac{1}{2}\sum_{i,j=1}^d (\sigma\sigma^\top)_{ij}(t)\partial_{x_i}\partial_{x_j} + \sum_{i=1}^d b_i(x,t)\partial_{x_i}, \tag{145}$$

*the value function $V$ for the SOC problem* (7)-(8) *solves the following Hamilton-Jacobi-Bellman (HJB) partial differential equation:*

$$\partial_t V(x,t) = -\mathcal{L}V(x,t) + \tfrac{1}{2}\|(\sigma^\top\nabla V)(x,t)\|^2 - f(x,t),$$
$$V(x,T) = g(x). \tag{146}$$

Consider forward SDEs like (50), starting from the distributions $p^{\text{base}}$ and $p^*$, where $p^*(x) \propto p^{\text{base}}(x)\exp(r(x))$.

$$\mathrm{d}\vec{X}_t = \vec{b}(\vec{X}_t, t)\,\mathrm{d}t + \sigma(t)\,\mathrm{d}B_t, \qquad \vec{X}_0 \sim p^{\text{base}}, \tag{147}$$
$$\mathrm{d}\vec{X}_t^* = \vec{b}^*(\vec{X}_t^*, t)\,\mathrm{d}t + \sigma(t)\,\mathrm{d}B_t, \qquad \vec{X}_0^* \sim p^*. \tag{148}$$

where the drifts are defined as

$$\vec{b}(x,t) = -\kappa_{1-t}x + \big(\tfrac{\sigma(1-t)^2}{2} - \eta_{1-t}\big)\mathfrak{s}(x,1-t) = -\kappa_{1-t}x + \big(\tfrac{\sigma(1-t)^2}{2} - \eta_{1-t}\big)\nabla\log\vec{p}_t(x),$$
$$\vec{b}^*(x,t) = -\kappa_{1-t}x + \big(\tfrac{\sigma(1-t)^2}{2} - \eta_{1-t}\big)\mathfrak{s}^*(x,1-t) = -\kappa_{1-t}x + \big(\tfrac{\sigma(1-t)^2}{2} - \eta_{1-t}\big)\nabla\log\vec{p}_t^*(x), \tag{149}$$

and $\vec{p}_t, \vec{p}_t^*$ are the densities of $X_t, \vec{X}_t$, respectively. $\vec{p}_t, \vec{p}_t^*$ satisfy Fokker-Planck equations:

$$\partial_t\vec{p}_t = \nabla\cdot(\vec{b}(x,t)\vec{p}_t) + \nabla\cdot\big(\tfrac{\sigma(1-t)^2}{2}\nabla\vec{p}_t\big), \qquad \vec{p}_0 = p^{\text{base}},$$
$$\partial_t\vec{p}_t^* = \nabla\cdot(\vec{b}^*(x,t)\vec{p}_t^*) + \nabla\cdot\big(\tfrac{\sigma(1-t)^2}{2}\nabla\vec{p}_t^*\big), \qquad \vec{p}_0 = p^*. \tag{150}$$

Plugging (149) into (150), we obtain

$$\partial_t\vec{p}_t = \nabla\cdot(\kappa_{1-t}x\vec{p}_t) + \nabla\cdot\big(\eta_{1-t}\nabla\vec{p}_t\big), \qquad \vec{p}_0 = p^{\text{base}},$$
$$\partial_t\vec{p}_t^* = \nabla\cdot(\kappa_{1-t}x\vec{p}_t^*) + \nabla\cdot\big(\eta_{1-t}\nabla\vec{p}_t^*\big), \qquad \vec{p}_0 = p^*. \tag{151}$$

We apply the Hopf-Cole transformation to obtain PDEs for $-\log\vec{p}_t$ (and $-\log\vec{p}_t^*$ analogously):

$$-\partial_t(-\log\vec{p}_t) = \frac{\partial_t p_t}{p_t} = \frac{\nabla\cdot(\kappa_{1-t}x\vec{p}_t) + \nabla\cdot\big(\eta_{1-t}\nabla\vec{p}_t\big)}{p_t}$$
$$= \kappa_{1-t}\nabla\cdot x + \kappa_{1-t}\langle x, \nabla\log\vec{p}_t\rangle + \eta_{1-t}\frac{\nabla\cdot(\nabla\log\vec{p}_t\exp(\log p_t))}{p_t} \tag{152}$$
$$= \kappa_{1-t}d + \kappa_{1-t}\langle x, \nabla\log\vec{p}_t\rangle + \eta_{1-t}\big(\Delta\log\vec{p}_t + \|\nabla\log\vec{p}_t\|^2\big).$$

Hence, if we define $\mathscr{V}(x,t) = -\log\vec{p}_t(x)$, $\mathscr{V}^*(x,t) = -\log\vec{p}_t^*(x)$, then $\mathscr{V}$ and $\mathscr{V}^*$ satisfy the following Hamilton-Jacobi-Bellman equations:

$$-\partial_t\mathscr{V} = \kappa_{1-t}d - \kappa_{1-t}\langle x, \nabla\mathscr{V}\rangle + \eta_{1-t}\big(-\Delta\mathscr{V} + \|\nabla\mathscr{V}\|^2\big), \qquad \mathscr{V}(x,0) = -\log p^{\text{base}}(x), \tag{153}$$

$$-\partial_t\mathscr{V}^* = \kappa_{1-t}d - \kappa_{1-t}\langle x, \nabla\mathscr{V}^*\rangle + \eta_{1-t}\big(-\Delta\mathscr{V}^* + \|\nabla\mathscr{V}^*\|^2\big), \qquad \mathscr{V}^*(x,0) = -\log p^*(x). \tag{154}$$

Now, define $\hat{\mathscr{V}}(x,t) = \mathscr{V}^*(x,t) - \mathscr{V}(x,t)$. Subtracting (154) from (153), we obtain

$$-\partial_t\hat{\mathscr{V}} = -\kappa_{1-t}\langle x, \nabla\hat{\mathscr{V}}\rangle + \eta_{1-t}\big(-\Delta\hat{\mathscr{V}} + \|\nabla\mathscr{V}^*\|^2 - \|\nabla\mathscr{V}\|^2\big)$$
$$= -\kappa_{1-t}\langle x, \nabla\hat{\mathscr{V}}\rangle + \eta_{1-t}\big(-\Delta\hat{\mathscr{V}} + \|\nabla(\hat{\mathscr{V}} + \mathscr{V})\|^2 - \|\nabla\mathscr{V}\|^2\big)$$
$$= -\kappa_{1-t}\langle x, \nabla\hat{\mathscr{V}}\rangle + \eta_{1-t}\big(-\Delta\hat{\mathscr{V}} + \|\nabla\hat{\mathscr{V}}\|^2 + 2\langle\nabla\mathscr{V}, \nabla\hat{\mathscr{V}}\rangle\big) \tag{155}$$
$$= \langle-\kappa_{1-t}x + 2\eta_{1-t}\nabla\mathscr{V}, \nabla\hat{\mathscr{V}}\rangle + \eta_{1-t}\big(-\Delta\hat{\mathscr{V}} + \|\nabla\hat{\mathscr{V}}\|^2\big)$$
$$= \langle-\kappa_{1-t}x - 2\eta_{1-t}\mathfrak{s}(x,1-t), \nabla\hat{\mathscr{V}}\rangle + \eta_{1-t}\big(-\Delta\hat{\mathscr{V}} + \|\nabla\hat{\mathscr{V}}\|^2\big),$$
$$\hat{\mathscr{V}}(x,0) = -\log p^*(x) + \log p^{\text{base}}(x) = -r(x) + \log\big(\int p^{\text{base}}(y)\exp(r(y))\,\mathrm{d}y\big).$$

Hence, $\hat{\mathcal{V}}$ also satisfies a Hamilton-Jacobi-Bellman equation. If we define $V$ such that $\hat{\mathcal{V}}(x,t) = V(x, 1-t)$, we have that

$$
\partial_t V = \langle -\kappa_t x - 2\eta_t \mathfrak{s}(x,t), \nabla V \rangle + \eta_t \left( -\Delta V + \|\nabla V\|^2 \right),
$$
$$
V(x,1) = r(x) - \log \left( \int p^{\text{base}}(y) \exp(r(y)) \, dy \right). \tag{156}
$$

Using Thm. 3, we can reverse-engineer $V$ as the value function of the following SOC problem:

$$
\min_{u \in \mathcal{U}} \mathbb{E} \left[ \tfrac{1}{2} \int_0^1 \|u(X_t^u, t)\|^2 \, dt - r(x) + \log \left( \int p^{\text{base}}(y) \exp(r(y)) \, dy \right) \right], \tag{157}
$$
$$
\text{s.t. } dX_t^u = \left( \kappa_t x + 2\eta_t \mathfrak{s}(x,t) + \sqrt{2\eta_t} u(X_t^u, t) \right) dt + \sqrt{2\eta_t} dB_t, \qquad X_0^u \sim p_0. \tag{158}
$$

Note that this SOC problem is equal to the problem (7)-(8) with the choices $f = 0$, $g = -r$, and $\sigma(t) = \sqrt{2\eta_t}$. By equation (11), the optimal control of the problem (157)-(158) is of the form:

$$
u^*(x,t) = -\sqrt{2\eta_t} \nabla V(x,t) = -\sqrt{2\eta_t} \nabla \hat{\mathcal{V}}(x, 1-t) = -\sqrt{2\eta_t} \left( \nabla \mathcal{V}^*(x, 1-t) - \nabla \mathcal{V}(x, 1-t) \right)
$$
$$
= -\sqrt{2\eta_t} \left( -\nabla \log \vec{p}_{1-t}^*(x) + \nabla \log \vec{p}_{1-t}(x) \right) = \sqrt{2\eta_t} \left( \mathfrak{s}^*(x,t) - \mathfrak{s}(x,t) \right), \tag{159}
$$
$$
\iff \mathfrak{s}^*(x,t) = \mathfrak{s}(x,t) + u^*(x,t)/\sqrt{2\eta_t}. \tag{160}
$$

As in (51), the backward SDEs corresponding to the forward SDEs (148) take the following form:

$$
dX_t^* = \left( \kappa_t X_t^* + \left( \tfrac{\sigma(t)^2}{2} + \eta_t \right) \mathfrak{s}^*(X_t^*, t) \right) dt + \sigma(t) \, dB_t, \qquad X_0^* \sim N(0, I). \tag{161}
$$

If we plug (160) into this equation, we obtain

$$
dX_t^* = \left( \kappa_t X_t^* + \left( \tfrac{\sigma(t)^2}{2} + \eta_t \right) \left( \mathfrak{s}(X_t^*, t) + \tfrac{u^*(X_t^*, t)}{\sqrt{2\eta_t}} \right) \right) dt + \sigma(t) \, dB_t, \quad X_0^* \sim N(0, I), \tag{162}
$$
$$
\iff dX_t^* = \left( b(X_t^*, t) + \tfrac{\frac{\sigma(t)^2}{2} + \eta_t}{\sqrt{2\eta_t}} u^*(X_t^*, t) \right) dt + \sigma(t) \, dB_t, \qquad X_0^* \sim N(0, I). \tag{163}
$$

where we used that $b(x,t) = \kappa_t x + \left( \tfrac{\sigma(t)^2}{2} + \eta_t \right) \mathfrak{s}(x,t)$ by definition in equation (6).

**The fine-tuned inference SDE for DDIM** Now, for DDIM, we have that $u^*(x,t) = -\sqrt{\frac{\dot{\alpha}_t}{\alpha_t(1-\alpha_t)}} \left( \epsilon^*(x,t) - \epsilon^{\text{base}}(x,t) \right)$ by (143). Hence,

$$
\tfrac{\frac{\sigma(t)^2}{2} + \eta_t}{\sqrt{2\eta_t}} u^*(x,t) = -\tfrac{\frac{\sigma(t)^2}{2} + \frac{\dot{\alpha}_t}{2\alpha_t}}{\sqrt{\frac{\dot{\alpha}_t}{\alpha_t}}} \sqrt{\tfrac{\dot{\alpha}_t}{\alpha_t(1-\alpha_t)}} \left( \epsilon^*(x,t) - \epsilon^{\text{base}}(x,t) \right) = -\tfrac{\frac{\sigma(t)^2}{2} + \frac{\dot{\alpha}_t}{2\alpha_t}}{\sqrt{1-\alpha_t}} \left( \epsilon^*(x,t) - \epsilon^{\text{base}}(x,t) \right),
$$
$$
\tag{164}
$$

$$
\implies b(x,t) + \tfrac{\frac{\sigma(t)^2}{2} + \eta_t}{\sqrt{2\eta_t}} u^*(x,t) = \tfrac{\dot{\alpha}_t}{2\alpha_t} X_t - \left( \tfrac{\dot{\alpha}_t}{2\alpha_t} + \tfrac{\sigma(t)^2}{2} \right) \tfrac{\epsilon^{\text{base}}(X_t, t)}{\sqrt{1-\alpha_t}} - \tfrac{\frac{\sigma(t)^2}{2} + \frac{\dot{\alpha}_t}{2\alpha_t}}{\sqrt{1-\alpha_t}} \left( \epsilon^*(x,t) - \epsilon^{\text{base}}(x,t) \right)
$$
$$
= \tfrac{\dot{\alpha}_t}{2\alpha_t} X_t - \left( \tfrac{\dot{\alpha}_t}{2\alpha_t} + \tfrac{\sigma(t)^2}{2} \right) \tfrac{\epsilon^*(X_t, t)}{\sqrt{1-\alpha_t}}. \tag{165}
$$

We obtain that the fine-tuned inference SDE for DDIM is

$$
dX_t^* = \left( \tfrac{\dot{\alpha}_t}{2\alpha_t} X_t^* - \left( \tfrac{\dot{\alpha}_t}{2\alpha_t} + \tfrac{\sigma(t)^2}{2} \right) \tfrac{\epsilon^*(X_t^*, t)}{\sqrt{1-\alpha_t}} \right) dt + \sigma(t) \, dB_t, \qquad X_0^* \sim N(0, I), \tag{166}
$$

which is matches the SDE (29) with the choice $\epsilon = \epsilon^*$.

**The fine-tuned inference SDE for Flow Matching** For Flow Matching, we have that $u^*(x,t) = \sqrt{\frac{2}{\beta_t(\frac{\dot{\alpha}_t}{\alpha_t}\beta_t - \dot{\beta}_t)}}(v^*(x,t) - v^{\text{base}}(x,t))$ by (144). Hence,

$$
\begin{aligned}
\frac{\frac{\sigma(t)^2}{2} + \eta_t}{\sqrt{2\eta_t}} u^*(x,t) &= \frac{\frac{\sigma(t)^2}{2} + \beta_t(\frac{\dot{\alpha}_t}{\alpha_t}\beta_t - \dot{\beta}_t)}{\sqrt{2\beta_t(\frac{\dot{\alpha}_t}{\alpha_t}\beta_t - \dot{\beta}_t)}} \sqrt{\frac{2}{\beta_t(\frac{\dot{\alpha}_t}{\alpha_t}\beta_t - \dot{\beta}_t)}}(v^*(x,t) - v^{\text{base}}(x,t)) \\
&= \big(1 + \frac{\sigma(t)^2}{2\beta_t(\frac{\dot{\alpha}_t}{\alpha_t}\beta_t - \dot{\beta}_t)}\big)(v^*(x,t) - v^{\text{base}}(x,t)).
\end{aligned}
\tag{167}
$$

$$
\begin{aligned}
\implies b(x,t) + \frac{\frac{\sigma(t)^2}{2} + \eta_t}{\sqrt{2\eta_t}} u^*(x,t) &= v^{\text{base}}(x,t) + \frac{\sigma(t)^2}{2\beta_t(\frac{\dot{\alpha}_t}{\alpha_t}\beta_t - \dot{\beta}_t)}\big(v^{\text{base}}(x,t) - \frac{\dot{\alpha}_t}{\alpha_t}x\big) \\
&\quad + \big(1 + \frac{\sigma(t)^2}{2\beta_t(\frac{\dot{\alpha}_t}{\alpha_t}\beta_t - \dot{\beta}_t)}\big)(v^*(x,t) - v^{\text{base}}(x,t)) \\
&= v^*(x,t) + \frac{\sigma(t)^2}{2\beta_t(\frac{\dot{\alpha}_t}{\alpha_t}\beta_t - \dot{\beta}_t)}\big(v^*(x,t) - \frac{\dot{\alpha}_t}{\alpha_t}x\big).
\end{aligned}
\tag{168}
$$

We obtain that the fine-tuned inference SDE for Flow Matching is

$$
\mathrm{d}X_t^* = \big(v(X_t^*, t) + \frac{\sigma(t)^2}{2\beta_t(\frac{\dot{\alpha}_t}{\alpha_t}\beta_t - \dot{\beta}_t)}\big(v^*(X_t^*, t) - \frac{\dot{\alpha}_t}{\alpha_t}X_t^*\big)\big)\,\mathrm{d}t + \sigma(t)\,\mathrm{d}B_t, \qquad X_0^* \sim N(0, I),
\tag{169}
$$

which matches equation (4) with the choice $v = v^*$.

# F METHODS TO SOLVE SOC PROBLEMS

## F.1 EXISTING METHODS

### F.1.1 THE ADJOINT METHOD

The most basic method of optimizing the simulation of an SDE is to directly differentiate through the simulation using gradients from the SOC objective function. The adjoint method simply uses the objective:

$$
\mathcal{L}(u; \boldsymbol{X}) := \int_0^1 \big(\tfrac{1}{2}\|u(X_t, t)\|^2 + f(X_t, t)\big)\,\mathrm{d}t + g(X_1), \qquad \boldsymbol{X} \sim p^u.
\tag{170}
$$

This is a stochastic estimate of the control objective in (7), and the goal is to take compute the gradient of $\mathcal{L}(u; \boldsymbol{X})$ with respect to the parameters $\theta$ of the control $u$. Due to the continuous-time nature of SDEs, there are two main approaches to implementing this numerically. Firstly, the *Discrete Adjoint* method uses a "discretize-then-differentiate" approach, where the numerical solver for simulating the SDE is simply stored in memory then differentiated through, and it has been studied extensively (*e.g.*, Bierkens & Kappen (2014); Gómez et al. (2014); Hartmann & Schütte (2012); Kappen et al. (2012); Rawlik et al. (2013); Haber & Ruthotto (2017)). This approach, however, uses an extremely large amount of memory as the full computational graph of the numerical solver must be stored in memory and implementations often must rely on gradient checkpointing (Chen et al., 2016) to reduce memory usage.

Secondly, the *Continuous Adjoint* method exploits the continuous-time nature of SDEs and uses an analytical expression for the gradient of the control objective with respect to the intermediate states $X_t$, expressed as an adjoint ODE, and then applies a numerical method to simulate this gradient itself, hence it is referred to as a "differentiate-then-discretize" approach (Pontryagin, 1962; Chen et al., 2018; Li et al., 2020). We first define the *adjoint state* as:

$$
\begin{aligned}
a(t; \boldsymbol{X}, u) &:= \nabla_{X_t}\big(\int_t^1 \big(\tfrac{1}{2}\|u(X_{t'}, t')\|^2 + f(X_{t'}, t')\big)\,\mathrm{d}t' + g(X_1)\big), \\
&\text{where } \boldsymbol{X} \text{ solves } \mathrm{d}X_t = \big(b(X_t, t) + \sigma(t)u(X_t, t)\big)\,\mathrm{d}t + \sigma(t)\mathrm{d}B_t.
\end{aligned}
\tag{171}
$$

This implies that $\mathbb{E}_{\boldsymbol{X}\sim p^u}\left[a(t;\boldsymbol{X},u)\mid X_t = x\right] = \nabla_x J(u;x,t)$, where $J$ denotes the cost functional defined in (9). It can then be shown that this adjoint state satisfies [7]:

$$\frac{\mathrm{d}}{\mathrm{d}t}a(t;\boldsymbol{X},u) = -\left[a(t;\boldsymbol{X},u)^{\mathsf{T}}\left(\nabla_{X_t}(b(X_t,t)+\sigma(t)u(X_t,t))\right)+\nabla_{X_t}\left(f(X_t,t)+\tfrac{1}{2}\|u(X_t,t)\|^2\right)\right],\tag{172}$$

$$a(1;\boldsymbol{X},u) = \nabla g(X_1).\tag{173}$$

The adjoint state is solved backwards in time, starting from the terminal condition (173). Compututation of (172) can be done with a vector-Jacobian product which can be efficiently done on automatic differentiation software (Paszke et al., 2019). Once the adjoint state has been solved for $t \in [0,1]$, then the gradient of $\mathcal{L}(u;\boldsymbol{X})$ with respect to the parameters $\theta$ can be obtained by integrating over the entire time interval:

$$\frac{\mathrm{d}\mathcal{L}}{\mathrm{d}\theta} = \tfrac{1}{2}\int_0^1 \frac{\partial}{\partial\theta}\|u(X_t,t)\|^2\mathrm{d}t + \int_0^1 \frac{\partial u(X_t,t)}{\partial\theta}^{\mathsf{T}}\sigma(t)^{\mathsf{T}}a(t;\boldsymbol{X},u)\mathrm{d}t,\tag{174}$$

where the first term is the partial derivative of $\mathcal{L}$ w.r.t. $\theta$ and the second term is the partial derivative through the sample trajectory $\boldsymbol{X}$. See Prop. 6 in App. F.2 for a statement and proof of this result. The discrete and continuous adjoint methods converge to the same gradient as the step size of the numerical solvers go to zero. Both are scalable to high dimensions and have seen their fair share of usage in optimizing neural ODE/SDEs (Chen et al., 2018; 2021; Li et al., 2020). As the adjoint methods are essentially gradient-based optimization algorithms applied on a highly non-convex problem, many have also reported they can be unstable empirically (Mohamed et al., 2020; Suh et al., 2022; Domingo-Enrich et al., 2023).

### F.1.2 IMPORTANCE-WEIGHTED MATCHING OBJECTIVES FOR REGRESSING ONTO THE OPTIMAL CONTROL

An alternative is to consider regressing onto the optimal control $u^*$, which is the approach of the cross-entropy method (Rubinstein & Kroese, 2013; Zhang et al., 2014) and stochastic optimal control matching (SOCM; Domingo-Enrich et al. (2023)). These methods make use of path integral theory (Kappen, 2005) to express the optimal control through importance sampling, resulting in an *importance-weighted* least-squares objective function

$$\mathcal{L}_{\text{SOCM}}(u;\boldsymbol{X}) := \int_0^1\|u(X_t,t)-\hat{u}^*(X_t,t)\|^2\mathrm{d}t \times \omega(u,\boldsymbol{X}),\qquad \boldsymbol{X}\sim p^u,\tag{175}$$

where $\omega$ is an importance weighting that approximates sampling from the optimal distribution $p^*$, and $\hat{u}^*$ is a stochastic estimator of the optimal control relying on having sampled from the optimal process. We defer to Domingo-Enrich et al. (2023) for the exact details. The functional landscape of this objective is convex, which is argued to help yield stable training. However, the need for importance sampling renders this impractical for high dimensional applications: the variance of the importance weighting $\omega$ grows exponentially with dimension of the stochastic process, leading to catastrophic failure. This unfortunately means that such importance-weighted matching objectives are impractical for fine-tuning dynamical generative models; however, a least-squares objective is greatly coveted as it can lead to stable training and simple interpretations.

### F.2 DERIVATION OF THE CONTINUOUS ADJOINT METHOD

**Proposition 6.** *The gradient $\frac{\mathrm{d}\mathcal{L}}{\mathrm{d}\theta}$ of the adjoint loss $\mathcal{L}(u;\boldsymbol{X})$ defined in (170) with respect to the parameters $\theta$ of the control can be expressed as in (174).*

*Proof.* First, note that we can write

$$\nabla_\theta \mathbb{E}\left[\int_0^T\left(\tfrac{1}{2}\|u_\theta(X_t^{u_\theta},t)\|^2+f(X_t^{u_\theta},t)\right)\mathrm{d}t+g(X_T^{u_\theta})\right]$$
$$=\mathbb{E}\left[\int_0^T\nabla_\theta u_\theta(X_t^{u_\theta},t)u_\theta(X_t^{u_\theta},t)\mathrm{d}t\right]+\nabla_\theta\mathbb{E}\left[\int_0^T\left(\tfrac{1}{2}\|v(X_t^{u_\theta},t)\|^2+f(X_t^{u_\theta},t)\right)\mathrm{d}t+g(X_T^{u_\theta})\right]\Big|_{v=\text{stopgrad}(u_\theta)}.\tag{176}$$

---

[7]Note we use the convention that a Jacobian matrix $J = \nabla_x v(x)$ is defined as $J_{ij} = \frac{\partial v_i(x)}{\partial x_j}$.

To develop the second term, we apply Lemma 5. Namely, by the Leibniz rule and equation (181), we have that

$$\nabla_\theta \mathbb{E}\big[\int_0^T \big(\tfrac{1}{2}\|v(X_t^{u_\theta},t)\|^2 + f(X_t^{u_\theta},t)\big)\,\mathrm{d}t + g(X_T^{u_\theta})\big]|_{v=\mathrm{stopgrad}(u_\theta)}$$
$$= \mathbb{E}\big[\nabla_\theta\big(\int_0^T \big(\tfrac{1}{2}\|v(X_t^{u_\theta},t)\|^2 + f(X_t^{u_\theta},t)\big)\,\mathrm{d}t + g(X_T^{u_\theta})\big)|_{v=\mathrm{stopgrad}(u_\theta)}\big] \qquad (177)$$
$$= \mathbb{E}\big[\int_0^T (\nabla_\theta u_\theta)(X_t^{u_\theta}(\omega),t)^\top \sigma(t)^\top a_t(\omega)\,\mathrm{d}t\big].$$

Plugging the right-hand side of this equation into (176) concludes the proof. $\qquad\square$

**Lemma 5.** *Let $v$ be an arbitrary fixed vector field. The unique solution of the ODE*

$$\frac{\mathrm{d}}{\mathrm{d}t}a(t;\boldsymbol{X}^u,u) = -\Big[\big(\nabla_{X_t^u}(b(X_t^u,t)+\sigma(t)u(X_t^u,t))\big)^\top a(t;\boldsymbol{X}^u,u) + \nabla_{X_t^u}\big(f(X_t^u,t)+\tfrac{1}{2}\|v(X_t^u,t)\|^2\big)\Big],$$
$$(178)$$

$$a(1;\boldsymbol{X}^u,u) = \nabla g(X_1^u), \qquad\qquad (179)$$

*satisfies:*

$$a(t;\boldsymbol{X}^u,u) := \nabla_{X_t^u}\big(\int_t^1 \big(\tfrac{1}{2}\|u(X_{t'}^u,t')\|^2 + f(X_{t'}^u,t')\big)\,\mathrm{d}t' + g(X_1^u)\big),$$
$$\text{where } \boldsymbol{X}^u \text{ solves } \mathrm{d}X_t^u = \big(b(X_t^u,t)+\sigma(t)u(X_t^u,t)\big)\,\mathrm{d}t + \sigma(t)\mathrm{d}B_t. \qquad (180)$$

*Moreover, when $u = u_\theta$ is parameterized by $\theta$ we have that*

$$\nabla_\theta\big(\int_0^T \big(\tfrac{1}{2}\|v(X_t^{u_\theta},t)\|^2 + f(X_t^{u_\theta},t)\big)\,\mathrm{d}t + g(X_T^{u_\theta})\big) = \int_0^T (\nabla_\theta u_\theta)(X_t^{u_\theta}(\omega),t)\sigma(t)^\top a_t(\omega)\,\mathrm{d}t. \qquad (181)$$

*Proof.* We use an approach based on Lagrange multipliers which mirrors and extends the derivation of the adjoint ODE (Domingo-Enrich et al., 2023, Lemma 8). For shortness, we use the notation $\tilde{b}_\theta(x,t) := b(x,t) + \sigma(t)u_\theta(x,t)$. Define a process $a : \Omega \times [0,T] \to \mathbb{R}^d$ such that for any $\omega \in \Omega$, $a(\omega,\cdot)$ is differentiable. For a given $\omega \in \Omega$, we can write

$$\int_0^T \big(\tfrac{1}{2}\|v(X_t^{u_\theta},t)\|^2 + f(X_t^{u_\theta},t)\big)\,\mathrm{d}t + g(X_T^{u_\theta})$$
$$= \int_0^T \big(\tfrac{1}{2}\|v(X_t^{u_\theta},t)\|^2 + f(X_t^{u_\theta},t)\big)\,\mathrm{d}t + g(X_T^{u_\theta}) \qquad (182)$$
$$\quad - \int_0^T \langle a_t(\omega), (dX_t^{u_\theta}(\omega) - \tilde{b}_\theta(X_t^{u_\theta}(\omega),t)\,\mathrm{d}t - \sigma(t)\,\mathrm{d}B_t)\rangle.$$

By stochastic integration by parts (Domingo-Enrich et al., 2023, Lemma 9), we have that

$$\int_0^T \langle a_t(\omega), dX_t^{u_\theta}(\omega)\rangle = \langle a_T(\omega), X_T^{u_\theta}(\omega)\rangle - \langle a_0(\omega), X_0^{u_\theta}(\omega)\rangle - \int_0^T \langle X_t^{u_\theta}(\omega), \tfrac{da_t}{dt}(\omega)\rangle\,\mathrm{d}t. \quad (183)$$

Hence, if $X_0^{u_\theta} = x_0$ is the initial condition, we have that[8]

$$\nabla_{x_0}\big(\int_0^T \big(\tfrac{1}{2}\|v(X_t^{u_\theta},t)\|^2 + f(X_t^{u_\theta},t)\big)\,\mathrm{d}t + g(X_T^{u_\theta})\big)$$
$$= \nabla_{x_0}\big(\int_0^T \big(\tfrac{1}{2}\|v(X_t^{u_\theta},t)\|^2 + f(X_t^{u_\theta},t)\big)\,\mathrm{d}t + g(X_T^{u_\theta})$$
$$\quad - \langle a_T(\omega), X_T^{u_\theta}(\omega)\rangle + \langle a_0(\omega), X_0^{u_\theta}(\omega)\rangle + \int_0^T \big(\langle a_t(\omega), \tilde{b}_\theta(X_t^{u_\theta}(\omega),t)\rangle + \langle \tfrac{da_t}{dt}(\omega), X_t^{u_\theta}(\omega)\rangle\big)\,\mathrm{d}t$$
$$\quad + \int_0^T \langle a_t(\omega), \sigma(t)\,\mathrm{d}B_t\rangle\big)$$
$$= \int_0^T \nabla_{x_0}X_t^{u_\theta}(\omega)^\top \nabla_x\big(\tfrac{1}{2}\|v(X_t^{u_\theta},t)\|^2 + f(X_t^{u_\theta}(\omega),t)\big)\,\mathrm{d}t + \nabla_{x_0}X_T^{u_\theta}(\omega)^\top \nabla_x g(X_T^{u_\theta}(\omega))$$
$$\quad - \nabla_{x_0}X_T^{u_\theta}(\omega)^\top a_T(\omega) + \nabla_{x_0}X_0^{u_\theta}(\omega)^\top a_0(\omega)$$
$$\quad + \int_0^T \big(\nabla_{x_0}X_t^{u_\theta}(\omega)^\top \nabla_x\tilde{b}_\theta(X_t^{u_\theta}(\omega),t)^\top a_t(\omega) + \nabla_{x_0}X_t^{u_\theta}(\omega)^\top \tfrac{da_t}{dt}(\omega)\big)\,\mathrm{d}t$$
$$= \int_0^T \nabla_{x_0}X_t^{u_\theta}(\omega)^\top \big(\nabla_x\big(\tfrac{1}{2}\|v(X_t^{u_\theta},t)\|^2 + f(X_t^{u_\theta}(\omega),t)\big) + \nabla_x\tilde{b}_\theta(X_t^{u_\theta}(\omega),t)^\top a_t(\omega) + \tfrac{da_t}{dt}(\omega)\big)\,\mathrm{d}t$$
$$\quad + \nabla_{x_0}X_T^{u_\theta}(\omega)^\top \big(\nabla_x g(X_T^{u_\theta}(\omega)) - a_T(\omega)\big) + a_0(\omega). \qquad (184)$$

---

[8]Unlike (Domingo-Enrich et al., 2023, Lemma 8), we use the convention that a Jacobian matrix $J = \nabla_x v(x)$ is defined as $J_{ij} = \frac{\partial v_i(x)}{\partial x_j}$. Their definition of $\nabla_x v$ is the transpose of ours.

In the last line we used that $\nabla_{x_0} X_0^{u_\theta}(\omega) = \nabla_{x_0} x_0 = I$. If choose $a$ such that

$$\begin{aligned} da_t(\omega) &= \big( - \nabla_x \tilde{b}_\theta(X_t^{u_\theta}(\omega), t)^\top a_t(\omega) - \nabla_x \big( \tfrac{1}{2}\|v(X_t^{u_\theta}, t)\|^2 + f(X_t^{u_\theta}(\omega), t) \big) \big) \, dt, \\ a_T(\omega) &= \nabla_x g(X_T^{u_\theta}(\omega)), \end{aligned} \tag{185}$$

which is the ODE (178)-(179), then we obtain that

$$\nabla_{x_0} \big( \textstyle\int_0^T \big( \tfrac{1}{2}\|v(X_t^{u_\theta}, t)\|^2 + f(X_t^{u_\theta}, t) \big) \, dt + g(X_T^{u_\theta}) \big) = a_0(\omega) \tag{186}$$

Without loss of generality, this argument can be extended from $t = 0$ to an arbitrary $t \in [0, 1]$, which proves the first statement of the lemma.

To prove (181), we similarly write

$$\begin{aligned} &\nabla_\theta \big( \textstyle\int_0^T \big( \tfrac{1}{2}\|v(X_t^{u_\theta}, t)\|^2 + f(X_t^{u_\theta}, t) \big) \, dt + g(X_T^{u_\theta}) \big) \\ &= \nabla_\theta \big( \textstyle\int_0^T \big( \tfrac{1}{2}\|v(X_t^{u_\theta}, t)\|^2 + f(X_t^{u_\theta}, t) \big) \, dt + g(X_T^{u_\theta}) \\ &\quad - \langle a_T(\omega), X_T^{u_\theta}(\omega) \rangle + \langle a_0(\omega), X_0^{u_\theta}(\omega) \rangle + \textstyle\int_0^T \big( \langle a_t(\omega), \tilde{b}_\theta(X_t^{u_\theta}(\omega), t) \rangle + \langle \tfrac{da_t}{dt}(\omega), X_t^{u_\theta}(\omega) \rangle \big) \, dt \\ &\quad + \textstyle\int_0^T \langle a_t(\omega), \sigma(t) \, dB_t \rangle \big) \\ &= \textstyle\int_0^T \nabla_\theta X_t^{u_\theta}(\omega)^\top \nabla_x \big( \tfrac{1}{2}\|v(X_t^{u_\theta}, t)\|^2 + f(X_t^{u_\theta}(\omega), t) \big) \, dt + \nabla_\theta X_T^{u_\theta}(\omega)^\top \nabla_x g(X_T^{u_\theta}(\omega)) \\ &\quad - \nabla_\theta X_T^{u_\theta}(\omega)^\top a_T(\omega) + \nabla_\theta X_0^{u_\theta}(\omega)^\top a_0(\omega) \\ &\quad + \textstyle\int_0^T \big( \nabla_\theta X_t^{u_\theta}(\omega)^\top \nabla_x \tilde{b}_\theta(X_t^{u_\theta}(\omega), t)^\top a_t(\omega) + \nabla_\theta \tilde{b}_\theta(X_t^{u_\theta}(\omega), t)^\top a_t(\omega) + \nabla_\theta X_t^{u_\theta}(\omega)^\top \tfrac{da_t}{dt}(\omega) \big) \, dt \\ &= \textstyle\int_0^T \nabla_\theta X_t^{u_\theta}(\omega)^\top \big( \nabla_x \big( \tfrac{1}{2}\|v(X_t^{u_\theta}, t)\|^2 + f(X_t^{u_\theta}(\omega), t) \big) + \nabla_x \tilde{b}_\theta(X_t^{u_\theta}(\omega), t)^\top a_t(\omega) + \tfrac{da_t}{dt}(\omega) \big) \, dt \\ &\quad + \nabla_\theta X_T^{u_\theta}(\omega)^\top \big( \nabla_x g(X_T^{u_\theta}(\omega)) - a_T(\omega) \big) + \textstyle\int_0^T (\nabla_\theta \tilde{b}_\theta)(X_t^{u_\theta}(\omega), t)^\top a_t(\omega) \, dt. \end{aligned} \tag{187}$$

In the last line we used that $\nabla_\theta X_0^{u_\theta}(\omega) = \nabla_\theta x = 0$. When $a$ satisfies (185), we obtain that

$$\begin{aligned} &\nabla_\theta \big( \textstyle\int_0^T \big( \tfrac{1}{2}\|v(X_t^{u_\theta}, t)\|^2 + f(X_t^{u_\theta}, t) \big) \, dt + g(X_T^{u_\theta}) \big) \\ &= \textstyle\int_0^T (\nabla_\theta \tilde{b}_\theta)(X_t^{u_\theta}(\omega), t) a_t(\omega) \, dt = \textstyle\int_0^T (\nabla_\theta u_\theta)(X_t^{u_\theta}(\omega), t)^\top \sigma(t)^\top a_t(\omega) \, dt. \end{aligned} \tag{188}$$

The last equality holds because $\tilde{b}_\theta(x, t) := b(x, t) + \sigma(t) u_\theta(x, t)$. $\qquad\qquad\square$

### F.3 PROOF OF PROP. 2: THEORETICAL GUARANTEES OF THE BASIC ADJOINT MATCHING LOSS

Let $\bar{u} = \texttt{stopgrad}(u_\theta)$. We can rewrite equation (174) as:

$$\nabla_\theta \mathcal{L}(u_\theta; \boldsymbol{X}^{\bar{u}}) = \tfrac{1}{2} \textstyle\int_0^1 \nabla_\theta \|u_\theta(X_t^{\bar{u}}, t)\|^2 dt + \textstyle\int_0^1 \nabla_\theta u(X_t^{\bar{u}}, t)^\top \sigma(t)^\top a(t; \boldsymbol{X}^{\bar{u}}, \bar{u}) dt \tag{189}$$

$$= \tfrac{1}{2} \textstyle\int_0^1 \nabla_\theta \|u_\theta(X_t^{\bar{u}}, t) + \sigma(t)^\top a(t; \boldsymbol{X}^{\bar{u}}, \bar{u})\|^2 dt = \nabla_\theta \mathcal{L}_{\text{Basic-Adj-Match}}(u_\theta; \boldsymbol{X}^{\bar{u}}) \tag{190}$$

This proves the first statement of the proposition. To prove that the only critical point of the expected basic Adjoint Matching loss is the optimal control, we first compute the first variation of $\mathbb{E}[\mathcal{L}_{\text{Basic-Adj-Match}}]$. Letting $v : \mathbb{R}^d \times [0, T] \to \mathbb{R}^d$ be arbitrary, we have that

$$\begin{aligned} &\tfrac{d}{d\epsilon} \mathbb{E}[\mathcal{L}_{\text{Basic-Adj-Match}}(u + \epsilon v; \boldsymbol{X}^{\bar{u}})] = \tfrac{d}{d\epsilon} \mathbb{E}\big[ \tfrac{1}{2} \textstyle\int_0^T \|(u + \epsilon v)(X_t^{\bar{u}}, t) + \sigma(t)^\top a(t, X^{\bar{u}}, \bar{u})\|^2 \, dt \big] \\ &= \mathbb{E}\big[ \textstyle\int_0^T \langle v(X_t^{\bar{u}}, t), u(X_t^{\bar{u}}, t) + \sigma(t)^\top a(t, X^{\bar{u}}, \bar{u}) \rangle \, dt \big] \\ &= \mathbb{E}\big[ \textstyle\int_0^T \langle v(X_t^{\bar{u}}, t), u(X_t^{\bar{u}}, t) + \sigma(t)^\top \mathbb{E}\big[ a(t, X^{\bar{u}}, \bar{u}) | X_t^{\bar{u}} \big] \rangle \, dt \big] \\ &\implies \tfrac{\delta}{\delta u} \mathbb{E}[\mathcal{L}_{\text{Basic-Adj-Match}}(u)(x, t) = u(x, t) + \mathbb{E}\big[ a(t, X^{\bar{u}}, \bar{u}) | X_t^{\bar{u}} = x \big] \end{aligned} \tag{191}$$

Hence, critical points satisfy that

$$\begin{aligned} u(x, t) &= -\sigma(t)^\top \mathbb{E}[a(t, X^u, u) | X_t^u = x] = -\sigma(t)^\top \mathbb{E}\big[ \nabla_{X_t^v} \textstyle\int_t^T \big( \tfrac{1}{2}\|v(X_t^v, t)\|^2 + f(X_t^v, t) \big) \, dt + g(X_T^v) | X_0^v = x \big] \\ &= -\sigma(t)^\top \nabla_x \mathbb{E}\big[ \textstyle\int_t^T \big( \tfrac{1}{2}\|v(X_t^v, t)\|^2 + f(X_t^v, t) \big) \, dt + g(X_T^v) | X_0^v = x \big] = -\sigma(t)^\top \nabla J(u; x, t), \end{aligned} \tag{192}$$

In this equation, the second equality holds by equation (180) from Lemma 5, and the third equality holds by the Leibniz rule.

Lemma 6 shows that any control $u$ that satisfies (192) is equal to the optimal control, which concludes the proof.

**Lemma 6.** *Suppose that for any* $x \in \mathbb{R}^d$, $t \in [0, T]$, $u(x, t) = -\sigma(t)^\top \nabla_x J(u; x, t)$. *Then,* $J(u; \cdot, \cdot)$ *satisfies the Hamilton-Jacobi-Bellman equation* (146). *By the uniqueness of the solution to the HJB equation, we have that* $J(u; x, t) = V(x, t)$ *for any* $x \in \mathbb{R}^d$, $t \in [0, T]$. *Hence,* $u(x, t) = -\sigma(t)^\top \nabla_x V(x, t)$ *is the optimal control.*

*Proof.* Since $J(u; x, t) = \mathbb{E}\big[\int_t^T \big(\frac{1}{2}\|u(X_t^u, t)\|^2 + f(X_t^u, t)\big)\, ds + g(X_T^u)|X_t^u = x\big]$, we have that

$$J(u; x, t) = \mathbb{E}\big[J(u; X_{t+\Delta t}^u, t + \Delta t)|X_t = x\big] + \mathbb{E}\big[\int_t^{t+\Delta t} \big(\frac{1}{2}\|u(X_s^u, s)\|^2 + f(X_s^u, s)\big)\, ds|X_t = x\big],$$
(193)

which means that

$$0 = \frac{\mathbb{E}[J(u; X_{t+\Delta t}^u, t + \Delta t)|X_t = x] - J(u; x, t)}{\Delta t} + \frac{\mathbb{E}\big[\int_t^{t+\Delta t} \big(\frac{1}{2}\|u(X_s^u, s)\|^2 + f(X_s^u, s)\big)\, ds|X_t = x\big]}{\Delta t}$$
(194)

Recall that the generator $\mathcal{T}^u$ of the controlled SDE (8) takes the form:

$$\mathcal{T}^u f(x, t) := \lim_{\Delta t \to 0} \frac{\mathbb{E}\big[f(X_{t+\Delta t}^u, t)|X_t=x\big] - f(x, t)}{\Delta t}$$
$$= \partial_t f(x, t) + \langle \nabla f(x, t), b(x, t) + \sigma(t)u(x, t)\rangle + \text{Tr}\big(\frac{\sigma(t)\sigma(t)^\top}{2}\nabla^2 f(x, t)\big)$$
(195)

Hence, if we take the limit $\Delta t \to 0$ on equation (194), we obtain that:

$$0 = \mathcal{T}^u J(u; x, t) + \frac{1}{2}\|u(x, t)\|^2 + f(x, t)$$
$$= \partial_t J(u; x, t) + \langle \nabla J(u; x, t), b(x, t) + \sigma(t)u(x, t)\rangle + \text{Tr}\big(\frac{\sigma(t)\sigma(t)^\top}{2}\nabla^2 J(u; x, t)\big) + \frac{1}{2}\|u(x, t)\|^2 + f(x, t).$$
(196)

Now using that $u(x, t) = -\sigma(t)^\top \nabla_x J(u; x, t)$, we have that

$$\langle \nabla J(u; x, t), \sigma(t)u(x, t)\rangle + \frac{1}{2}\|u(x, t)\|^2 = -\|\sigma(t)^\top \nabla_x J(u; x, t)\|^2 + \frac{1}{2}\|\sigma(t)^\top \nabla_x J(u; x, t)\|^2$$
$$= -\frac{1}{2}\|\sigma(t)^\top \nabla_x J(u; x, t)\|^2.$$
(197)

Plugging this back into (196), we obtain that

$$0 = \partial_t J(u; x, t) + \langle \nabla J(u; x, t), b(x, t)\rangle + \text{Tr}\big(\frac{\sigma(t)\sigma(t)^\top}{2}\nabla^2 J(u; x, t)\big) - \frac{1}{2}\|\sigma(t)^\top \nabla_x J(u; x, t)\|^2 + f(x, t).$$
(198)

And since $J(u; x, T) = g(x)$ by construction, we conclude that $J(u; x, t)$ satisfies the HJB equation (146). $\qquad\square$

### F.4 THEORETICAL GUARANTEES OF THE ADJOINT MATCHING LOSS

**Proposition 7** (Theoretical guarantee of the Adjoint Matching loss)**.** *The only critical point of the loss* $\mathbb{E}[\mathcal{L}_{\text{Adj-Match}}]$ *is the optimal control* $u^*$.

*Proof.* Let $v$ be an arbitrary control. If $\tilde{a}(t; \mathbf{X}^v)$ is the solution of the Lean Adjoint ODE (26)-(27), it satisfies the integral equation

$$\tilde{a}(t; \mathbf{X}^v) = \int_t^T \big(\nabla_x b(X_s^v, s)^\top \tilde{a}(s; \mathbf{X}^v) + \nabla_x f(X_s^v, s)\big)\, ds + \nabla g(X_T^v).$$
(199)

Hence,

$$\mathbb{E}\big[\tilde{a}(t; \mathbf{X}^v)\big|X_t^v\big] = \mathbb{E}\big[\int_t^T \big(\nabla_x b(X_s^v, s)^\top \tilde{a}(s; \mathbf{X}^v) + \nabla_x f(X_s^v, s)\big)\, ds + \nabla g(X_T^v)\big|X_t^v\big]$$
$$= \mathbb{E}\big[\int_t^T \big(\nabla_x b(X_s^v, s)^\top \mathbb{E}\big[\tilde{a}(s; \mathbf{X}^v)\big|X_s^v\big] + \nabla_x f(X_s^v, s)\big)\, ds + \nabla g(X_T^v)\big|X_t^v\big],$$
(200)

where we used the tower property of conditional expectation in the second equality.

Similarly, if $a(t; \mathbf{X}^v, v)$ is the solution of the Adjoint ODE (172)-(173), it satisfies the integral equation

$$a(t; \mathbf{X}^v, v) = \int_t^T \left( \nabla_x \big( b(X_s^v, s)^\top a(s; \mathbf{X}^v, v) + \sigma(s) v(X_s^v, s) \big) + \nabla_x \big( f(X_s^v, s) + \tfrac{1}{2} \| v(X_s^v, s) \|^2 \big) \right) \mathrm{d}s + \nabla g(X_T^v),$$
(201)

and its expected value satisfies

$$\mathbb{E}\big[ a(t; \mathbf{X}^v, v) \big| X_t^v \big]$$
$$= \mathbb{E}\big[ \int_t^T \left( \nabla_x \big( b(X_s^v, s) + \sigma(s) v(X_s^v, s) \big)^\top a(s; \mathbf{X}^v, v) + \nabla_x \big( f(X_s^v, s) + \tfrac{1}{2} \| v(X_s^v, s) \|^2 \big) \right) \mathrm{d}s + \nabla g(X_T^v) \big| X_t^v \big]$$
$$= \mathbb{E}\big[ \int_t^T \left( \nabla_x \big( b(X_s^v, s) + \sigma(s) v(X_s^v, s) \big)^\top \mathbb{E}\big[ a(s; \mathbf{X}^v, v) \big| X_s^v \big] + \nabla_x \big( f(X_s^v, s) + \tfrac{1}{2} \| v(X_s^v, s) \|^2 \big) \right) \mathrm{d}s + \nabla g(X_T^v) \big| X_t^v \big].$$
(202)

Let us rewrite $\mathbb{E}[\mathcal{L}_{\mathrm{Adj-Match}}]$ as follows:

$$\mathbb{E}[\mathcal{L}_{\mathrm{Adj-Match}}(u)] := \mathbb{E}\big[ \int_0^T \big\| u(X_t^v, t) + \sigma(t)^\top \mathbb{E}\big[ \tilde{a}(t, \mathbf{X}^v) | X_t^v \big] \big\|^2 \mathrm{d}t \big]|_{v=\mathrm{stopgrad}(u)}$$
$$+ \mathbb{E}\big[ \int_0^T \big\| \sigma(t)^\top \big( \mathbb{E}\big[ \tilde{a}(t, \mathbf{X}^v) | X_t^v \big] - \tilde{a}(t, \mathbf{X}^v) \big) \big\|^2 \mathrm{d}t \big]|_{v=\mathrm{stopgrad}(u)},$$
(203)

Now, suppose that $\hat{u}$ is a critical point of $\mathbb{E}[\mathcal{L}_{\mathrm{Adj-Match}}]$. By definition, this implies that the first variation of $\mathbb{E}[\mathcal{L}_{\mathrm{Adj-Match}}]$ is zero. Using (203), we can write this as follows:

$$0 = \frac{\delta}{\delta u} \mathbb{E}[\mathcal{L}_{\mathrm{Adj-Match}}(\hat{u})](x) = 2\big( \hat{u}(x, t) + \sigma(t)^\top \mathbb{E}[\tilde{a}(t, \mathbf{X}^{\hat{u}}) | X_t^{\hat{u}} = x] \big), \qquad (204)$$
$$\implies \hat{u}(x, t) = -\sigma(t)^\top \mathbb{E}[\tilde{a}(t, \mathbf{X}^{\hat{u}}) | X_t^{\hat{u}} = x]. \qquad (205)$$

Hence, we have

$$\nabla_x \hat{u}(X_t^{\hat{u}}, t)^\top \sigma(t)^\top \mathbb{E}[\tilde{a}(t, \mathbf{X}^{\hat{u}}) | X_t^{\hat{u}}] + \nabla_x \hat{u}(X_t^{\hat{u}}, t)^\top \hat{u}(X_t^{\hat{u}}, t) = 0, \qquad (206)$$
$$\implies \mathbb{E}\big[ \int_t^T \left( \nabla_x \big( \sigma(s) \hat{u}(X_s^{\hat{u}}, s) \big)^\top \mathbb{E}\big[ \tilde{a}(s; \mathbf{X}^{\hat{u}}) \big| X_s^{\hat{u}} \big] + \nabla_x \big( \tfrac{1}{2} \| \hat{u}(X_s^{\hat{u}}, s) \|^2 \big) \right) \mathrm{d}s \big| X_t^{\hat{u}} \big] = 0. \quad (207)$$

If we set $v = \hat{u}$ in equation (200), and add (207) to its right-hand side, we obtain that $\mathbb{E}[\tilde{a}(t, X^{\hat{u}}) | X_t^{\hat{u}}]$ also solves the integral equation

$$\mathbb{E}\big[ \tilde{a}(t; \mathbf{X}^{\hat{u}}) \big| X_t^{\hat{u}} \big]$$
$$= \mathbb{E}\big[ \int_t^T \left( \nabla_x \big( b(X_s^{\hat{u}}, s) + \sigma(s) \hat{u}(X_s^{\hat{u}}, s) \big)^\top \mathbb{E}\big[ \tilde{a}(s; \mathbf{X}^{\hat{u}}) \big| X_s^{\hat{u}} \big] + \nabla_x \big( f(X_s^{\hat{u}}, s) + \tfrac{1}{2} \| \hat{u}(X_s^{\hat{u}}, s) \|^2 \big) \right) \mathrm{d}s + \nabla g(X_T^{\hat{u}}) \big| X_t^{\hat{u}} \big].$$
(208)

Note that this integral equation is the same one as equation (202) when we set $v = \hat{u}$ in the latter. Prop. 8 states that the solution of the integral equation is unique, which means that $\mathbb{E}\big[ \tilde{a}(t; \mathbf{X}^{\hat{u}}) \big| X_t^{\hat{u}} \big] = \mathbb{E}\big[ a(t; \mathbf{X}^{\hat{u}}, \hat{u}) \big| X_t^{\hat{u}} \big]$ for all $t \in [0, T]$.

Since we can reexpress the basic Adjoint Matching loss as

$$\mathbb{E}[\mathcal{L}_{\mathrm{Basic-Adj-Match}}(u)] := \mathbb{E}\big[ \int_0^T \big\| u(X_t^v, t) + \sigma(t)^\top \mathbb{E}\big[ a(t; \mathbf{X}^v, v) | X_t^v \big] \big\|^2 \mathrm{d}t \big]|_{v=\mathrm{stopgrad}(u)}$$
$$+ \mathbb{E}\big[ \int_0^T \big\| \sigma(t)^\top \big( \mathbb{E}\big[ a(t; \mathbf{X}^v, v) | X_t^v \big] - a(t; \mathbf{X}^v, v) \big) \big\|^2 \mathrm{d}t \big]|_{v=\mathrm{stopgrad}(u)},$$
(209)

we obtain that when $\hat{u}$ is a critical point of $\mathbb{E}[\mathcal{L}_{\mathrm{Adj-Match}}]$,

$$\frac{\mathrm{d}}{\mathrm{d}u} \mathbb{E}[\mathcal{L}_{\mathrm{Basic-Adj-Match}}(\hat{u})](x) = 2\big( \hat{u}(x, t) + \sigma(t)^\top \mathbb{E}[a(t; \mathbf{X}^{\hat{u}}, \hat{u}) | X_t^{\hat{u}} = x] \big)$$
$$= 2\big( \hat{u}(x, t) + \sigma(t)^\top \mathbb{E}[\tilde{a}(t; \mathbf{X}^{\hat{u}}) | X_t^{\hat{u}} = x] \big) = 0,$$
(210)

where the second equality holds because $\mathbb{E}\big[ \tilde{a}(t; \mathbf{X}^{\hat{u}}) \big| X_t^{\hat{u}} \big] = \mathbb{E}\big[ a(t; \mathbf{X}^{\hat{u}}, \hat{u}) \big| X_t^{\hat{u}} \big]$, and the third equality holds by equation (205). Thus, we deduce that the critical points of $\mathbb{E}[\mathcal{L}_{\mathrm{Adj-Match}}]$ are critical points of $\mathbb{E}[\mathcal{L}_{\mathrm{Basic-Adj-Match}}]$. By Prop. 2, $\mathbb{E}[\mathcal{L}_{\mathrm{Basic-Adj-Match}}]$ has a single critical point, which is the optimal control $u^*$, which concludes the proof of the statement for $\mathbb{E}[\mathcal{L}_{\mathrm{Adj-Match}}]$. $\qquad \square$

**Proposition 8.** *Let $v$ be an arbitrary control. Consider the integral equation:*

$$Y_t = \mathbb{E}\big[\int_t^T \big(\nabla_x\big(b(X_s^v,s)+\sigma(s)v(X_s^v,s)\big)^\top Y_s + \nabla_x\big(f(X_s^v,s)+\tfrac{1}{2}\|v(X_s^v,s)\|^2\big)\big)\,\mathrm{d}s + \nabla g(X_T^v)\big|X_t^v\big],$$
(211)

*where $t \in [0,T]$. This equation has a unique solution, i.e. if $Y^1$, $Y^2$ are two solutions then $Y_1 = Y_2$.*

*Proof.* Let $Y^1$, $Y^2$ be two solutions of the integral equation. We have that

$$Y_t^1 - Y_t^2 = \mathbb{E}\big[\int_t^T \big((Y_s^1-Y_s^2)^\top \nabla_x b(X_s^*,s)\big)\mathrm{d}s\big|X_t^*\big].$$
(212)

Thus,

$$\|Y_t^1 - Y_t^2\|$$
$$\leq \mathbb{E}\big[\|\int_t^T \big((Y_s^1-Y_s^2)^\top\nabla_x b(X_s^*,s)\big)\mathrm{d}s\|\big|X_t^*\big] \leq \mathbb{E}\big[\int_t^T \|\big((Y_s^1-Y_s^2)^\top\nabla_x b(X_s^*,s)\big)\|\mathrm{d}s\big|X_t^*\big]$$
$$\leq \mathbb{E}\big[\int_t^T \|Y_s^1-Y_s^2\|\cdot\|\nabla_x b(X_s^*,s)\big)\|\mathrm{d}s\big|X_t^*\big] = \int_t^T \mathbb{E}\big[\|Y_s^1-Y_s^2\|\cdot\|\nabla_x b(X_s^*,s)\big)\|\big|X_t^*\big]\mathrm{d}s$$
$$\leq \int_t^T \big(\mathbb{E}\big[\|Y_s^1-Y_s^2\|^2\big|X_t^*\big]\big)^{1/2}\cdot\big(\mathbb{E}\big[\|\nabla_x b(X_s^*,s)\|^2\big|X_t^*\big]\big)^{1/2}\mathrm{d}s$$
(213)

And this implies that

$$\sup_{t'\in[0,t]}\big(\mathbb{E}[\|Y_t^1-Y_t^2\|^2|X_{t'}^*]\big)^{1/2}$$
$$\leq \int_t^T \big(\mathbb{E}\big[\|Y_s^1-Y_s^2\|^2\big|X_t^*\big]\big)^{1/2}\cdot\big(\mathbb{E}\big[\|\nabla_x b(X_s^*,s)\|^2\big|X_t^*\big]\big)^{1/2}\mathrm{d}s$$
$$\leq \int_t^T \sup_{t'\in[0,s]}\big(\mathbb{E}\big[\|Y_s^1-Y_s^2\|^2\big|X_{t'}^*\big]\big)^{1/2}\cdot\sup_{t'\in[0,s]}\big(\mathbb{E}\big[\|\nabla_x b(X_s^*,s)\|^2\big|X_{t'}^*\big]\big)^{1/2}\mathrm{d}s.$$
(214)

Applying Grönwall's inequality on the function $f(t) = \sup_{t'\in[0,t]}\big(\mathbb{E}[\|Y_t^1 - Y_t^2\|^2|X_{t'}^*]\big)^{1/2}$, we obtain that $\sup_{t'\in[0,t]}\big(\mathbb{E}[\|Y_t^1 - Y_t^2\|^2|X_{t'}^*]\big)^{1/2} = 0$ for all $t \in [0,T]$, which means that $Y_t^1 = Y_t^2$ almost surely. And since $\|Y_t^1 - Y_t^2\| \leq \int_t^T \big(\mathbb{E}\big[\|Y_s^1 - Y_s^2\|^2\big|X_t^*\big]\big)^{1/2}\cdot\big(\mathbb{E}\big[\|\nabla_x b(X_s^*,s)\|^2\big|X_t^*\big]\big)^{1/2}\mathrm{d}s = 0$, we obtain that $Y^1 = Y^2$. $\square$

### F.5 Pseudo-code of Adjoint Matching for Flow Matching and DDIM Fine-tuning

Note that for each pair of equations (218)-(219), (220)-(221), (222)-(223), the first equation corresponds to the updates in the DDPM paper, while the second equation is an Euler-Maruyama / Euler discretization of the continuous-time object. To check that both discretizations are equal up to first order, remark that

$$\sqrt{\frac{\bar{\alpha}_{k+1}}{\bar{\alpha}_k}} = \sqrt{1 + \frac{\bar{\alpha}_{k+1}-\bar{\alpha}_k}{\bar{\alpha}_k}} \approx 1 + \frac{\bar{\alpha}_{k+1}-\bar{\alpha}_k}{2\bar{\alpha}_k} + O((\bar{\alpha}_{k+1}-\bar{\alpha}_k)^2).$$
(224)

## G Adapting diffusion fine-tuning baselines to flow matching

### G.1 Adapting ReFL (Xu et al., 2023) to flow matching

Reward Feedback Learning (ReFL) is a diffusion fine-tuning algorithm introduced by Xu et al. (2023) which tries to increase the reward on denoised samples. Namely, if $\boldsymbol{X} = (X_t)_{t\in[0,1]}$ is the solution of the DDPM SDE (30), we can denoise $X_t$ as

$$\hat{X}_1(X_t) = \frac{X_t - \sqrt{1-\bar{\alpha}_t}\epsilon(X_t,t)}{\sqrt{\bar{\alpha}_t}}.$$
(225)

This equation follows from the stochastic interpolant equation (2) if we replace $\bar{X}_0$ with the noise predictor $\epsilon(X_t,t)$. And then, the ReFL optimization update is based on the gradient:

$$\nabla_\theta r(\hat{X}_1(X_t)) = \nabla_\theta r\big(\frac{X_t - \sqrt{1-\bar{\alpha}_t}\epsilon_\theta(X_t,t)}{\sqrt{\bar{\alpha}_t}}\big),$$
(226)

where the trajectories have been detached.

---

**Algorithm 1** Adjoint Matching for fine-tuning Flow Matching models

---

**Input:** Pre-trained FM velocity field $v^{\text{base}}$, step size $h$, number of fine-tuning iterations $N$.
Initialize fine-tuned vector fields: $v^{\text{finetune}} = v^{\text{base}}$ with parameters $\theta$.
**for** $n \in \{0, \ldots, N-1\}$ **do**

Sample $m$ trajectories $\boldsymbol{X} = (X_t)_{t \in \{0,\ldots,1\}}$ with memoryless noise schedule $\sigma(t) = \sqrt{2\beta_t(\frac{\dot{\alpha}_t}{\alpha_t}\beta_t - \dot{\beta}_t)}$,
*e.g.*:

$$X_{t+h} = X_t + h\left(2v_\theta^{\text{finetune}}(X_t, t) - \frac{\dot{\alpha}_t}{\alpha_t}X_t\right) + \sqrt{h}\sigma(t)\varepsilon_t, \qquad \varepsilon_t \sim \mathcal{N}(0, I), \qquad X_0 \sim \mathcal{N}(0, I).$$
(215)

For each trajectory, solve the *lean adjoint ODE* (26)-(27) backwards in time from $t = 1$ to $0$, *e.g.*:

$$\tilde{a}_{t-h} = \tilde{a}_t + h\tilde{a}_t^\mathsf{T}\nabla_{X_t}\left(2v^{\text{base}}(X_t, t) - \frac{\dot{\alpha}_t}{\alpha_t}X_t\right), \qquad \tilde{a}_1 = -\nabla_{X_1}r(X_1).$$
(216)

Note that $X_t$ and $\tilde{a}_t$ should be computed without gradients, *i.e.*, $X_t = \texttt{stopgrad}(X_t)$, $\tilde{a}_t = \texttt{stopgrad}(\tilde{a}_t)$.

For each trajectory, compute the Adjoint Matching objective (25):

$$\mathcal{L}_{\text{Adj-Match}}(\theta) = \sum_{t \in \{0,\ldots,1-h\}} \left\|\frac{2}{\sigma(t)}\left(v_\theta^{\text{finetune}}(X_t, t) - v^{\text{base}}(X_t, t)\right) + \sigma(t)\tilde{a}_t\right\|^2.$$
(217)

Compute the gradient $\nabla_\theta \mathcal{L}(\theta)$ and update $\theta$ using favorite gradient descent algorithm.
**end**
**Output:** Fine-tuned vector field $v^{\text{finetune}}$

---

To adapt ReFL to Flow Matching, we need to express the denoiser map in terms of the vector field $v$. We have that

$$\begin{aligned}
v(x, t) &= \mathbb{E}\left[\dot{\beta}_t\bar{X}_0 + \dot{\alpha}_t\bar{X}_1 \big| \beta_t\bar{X}_0 + \alpha_t\bar{X}_1 = x\right] \\
&= \mathbb{E}\left[\tfrac{\dot{\beta}_t}{\beta_t}\left(\beta_t\bar{X}_0 + \alpha_t\bar{X}_1\right) + \left(\dot{\alpha}_t - \tfrac{\dot{\beta}_t}{\beta_t}\alpha_t\right)\bar{X}_1 \big| \beta_t\bar{X}_0 + \alpha_t\bar{X}_1 = x\right] \\
&= \tfrac{\dot{\beta}_t}{\beta_t}x + \left(\dot{\alpha}_t - \tfrac{\dot{\beta}_t}{\beta_t}\alpha_t\right)\hat{X}_1(x, t).
\end{aligned}$$
(227)

where we defined the denoiser map $\hat{X}_1(x, t) := \mathbb{E}\left[\bar{X}_1 | \beta_t\bar{X}_0 + \alpha_t\bar{X}_1 = x\right]$. Hence,

$$\hat{X}_1(x, t) = \frac{v(x,t) - \frac{\dot{\beta}_t}{\beta_t}x}{\dot{\alpha}_t - \frac{\dot{\beta}_t}{\beta_t}\alpha_t}.$$
(228)

## G.2 ADAPTING DIFFUSION-DPO (WALLACE ET AL., 2023A) TO FLOW MATCHING

The Diffusion-DPO loss assumes access to ranked pairs of generated samples $x_1^w \succ x_1^l$, where $x^w$ and $x^l$ are the winning and losing samples. For DDPM, the loss implemented in practice reads (Wallace et al., 2023a, Eq. 46):

$$\begin{aligned}
L_{\text{DPO}}(\theta) = -\mathbb{E}_{(x_1^w, x_1^l)\sim\mathcal{D}, k\sim U[0,K], x_{kh}^w\sim q(x_{kh}^w|x_1^w), x_t^l\sim q(x_{kh}^l|x_1^l)}\Big[ \\
\log S\big(-\tfrac{\tilde{\beta}}{2}\big(\|\varepsilon^w - \epsilon_\theta(x_{kh}^w, kh)\|^2 - \|\varepsilon^w - \epsilon_{\text{ref}}(x_{kh}^w, kh)\|^2 \\
-\big(\|\varepsilon^l - \epsilon_\theta(x_{kh}^l, kh)\|^2 - \|\varepsilon^l - \epsilon_{\text{ref}}(x_{kh}^l, kh)\|^2\big)\big)\big)\Big],
\end{aligned}$$
(229)

where $S(x) = \frac{1}{1+e^{-x}}$ denotes the sigmoid function, and $q(x_{kh}^*|x_1^*)$ is the conditional distribution of the forward process, i.e. $x_{kh}^*$ is sampled as $x_{kh}^* = \sqrt{\gamma_{kh}}x_1^* + \sqrt{1 - \gamma_{kh}}\epsilon$, $\epsilon \sim N(0, I)$. Following the derivation of the Diffusion-DPO loss in (Wallace et al., 2023a, Sec. S4), we observe that the term $-\frac{\tilde{\beta}}{2}\|\varepsilon^w - \epsilon_\theta(x_{kh}^w, kh)\|^2$ arises from

$$-\frac{\tilde{\beta}}{2\frac{1-\gamma_{kh}}{\gamma_{kh}}}\|\hat{x}_1(x_{kh}^w) - x_1^w\|^2,$$
(230)

up to a constant term in $\theta$. If we switch to the more general flow matching scheme, the analog of this term is

$$-\frac{\tilde{\beta}}{2\frac{\beta_{kh}^2}{\alpha_{kh}^2}}\|\hat{x}_1(x_{kh}^w) - x_1^w\|^2.$$
(231)

---

**Algorithm 2** Adjoint Matching for fine-tuning DDIM

**Input:** Pre-trained denoiser $\epsilon^{\text{base}}$, number of fine-tuning iterations $N$.
Initialize fine-tuned denoiser: $\epsilon^{\text{finetune}} = \epsilon^{\text{base}}$ with parameters $\theta$.
**for** $n \in \{0, \ldots, N-1\}$ **do**
  Sample $m$ trajectories $\boldsymbol{X} = (X_t)_{t \in \{0,\ldots,1\}}$ according to DDPM, *e.g.*:

$$X_{k+1} = \sqrt{\tfrac{\bar{\alpha}_{k+1}}{\bar{\alpha}_k}}\big(X_k - \tfrac{1-\bar{\alpha}_k/\bar{\alpha}_{k+1}}{\sqrt{1-\bar{\alpha}_k}}\epsilon^{\text{finetune}}(X_k,k)\big) + \sqrt{\tfrac{1-\bar{\alpha}_{k+1}}{1-\bar{\alpha}_k}\big(1-\tfrac{\bar{\alpha}_k}{\bar{\alpha}_{k+1}}\big)}\varepsilon_k, \qquad (218)$$

$$\text{or } X_{k+1} = X_k + \tfrac{\bar{\alpha}_{k+1}-\bar{\alpha}_k}{2\bar{\alpha}_k}X_k - \tfrac{\bar{\alpha}_{k+1}-\bar{\alpha}_k}{\bar{\alpha}_k\sqrt{1-\bar{\alpha}_k}}\epsilon^{\text{finetune}}(X_k,k) + \sqrt{\tfrac{\bar{\alpha}_{k+1}-\bar{\alpha}_k}{\bar{\alpha}_k}}\varepsilon_k, \qquad (219)$$

  where $\varepsilon_k \sim \mathcal{N}(0,I)$, $X_0 \sim \mathcal{N}(0,I)$.
  For each trajectory, solve the *lean adjoint ODE* (26)-(27) backwards in time from $k=K$ to 0, *e.g.*:

$$\tilde{a}_k = \tilde{a}_{k+1} + \tilde{a}_{k+1}^{\mathsf{T}}\nabla_{X_k}\left(\sqrt{\tfrac{\bar{\alpha}_{k+1}}{\bar{\alpha}_k}}\big(X_k - \tfrac{1-\bar{\alpha}_k/\bar{\alpha}_{k+1}}{\sqrt{1-\bar{\alpha}_k}}\epsilon^{\text{base}}(X_k,k)\big) - X_k\right), \qquad \tilde{a}_K = \nabla_{X_K}r(X_K), \tag{220}$$

$$\text{or } \tilde{a}_k = \tilde{a}_{k+1} + \tilde{a}_{k+1}^{\mathsf{T}}\nabla_{X_t}\left(\tfrac{\bar{\alpha}_{k+1}-\bar{\alpha}_k}{2\bar{\alpha}_k}X_k - \tfrac{\bar{\alpha}_{k+1}-\bar{\alpha}_k}{\bar{\alpha}_k\sqrt{1-\bar{\alpha}_k}}\epsilon^{\text{base}}(X_k,k)\right), \qquad \tilde{a}_K = \nabla_{X_K}r(X_K). \tag{221}$$

  Note that $X_k$ and $\tilde{a}_k$ should be computed without gradients, *i.e.*, $X_k = \texttt{stopgrad}(X_k)$, $\tilde{a}_k = \texttt{stopgrad}(\tilde{a}_k)$.

  For each trajectory, compute the Adjoint Matching objective (25):

$$\mathcal{L}_{\text{Adj-Match}}(\theta) = \sum_{k \in \{0,\ldots,K-1\}} \left\| \sqrt{\tfrac{\bar{\alpha}_{k+1}}{\bar{\alpha}_k(1-\bar{\alpha}_{k+1})}}\big(1-\tfrac{\bar{\alpha}_k}{\bar{\alpha}_{k+1}}\big)(\epsilon^{\text{finetune}}(X_k,k) - \epsilon^{\text{base}}(X_k,k)) \right.$$
$$\left. - \sqrt{\tfrac{1-\bar{\alpha}_{k+1}}{1-\bar{\alpha}_k}}\big(1-\tfrac{\bar{\alpha}_k}{\bar{\alpha}_{k+1}}\big)\tilde{a}_k \right\|^2, \tag{222}$$

$$\text{or } \mathcal{L}_{\text{Adj-Match}}(\theta) = \sum_{k \in \{0,\ldots,K-1\}} \left\| \sqrt{\tfrac{\bar{\alpha}_{k+1}-\bar{\alpha}_k}{\bar{\alpha}_k(1-\bar{\alpha}_k)}}(\epsilon^{\text{finetune}}(X_k,k) - \epsilon^{\text{base}}(X_k,k)) - \sqrt{\tfrac{\bar{\alpha}_{k+1}-\bar{\alpha}_k}{\bar{\alpha}_k}}\tilde{a}_k \right\|^2. \tag{223}$$

  Compute the gradient $\nabla_\theta \mathcal{L}(\theta)$ and update $\theta$ using favorite gradient descent algorithm.
**end**
**Output:** Fine-tuned vector field $v^{\text{finetune}}$

---

Using the expression of the denoiser map in terms of the vector field $v$ in equation (228), we can rewrite (231) as:

$$-\frac{\tilde{\beta}}{2\frac{\beta_{kh}^2}{\alpha_{kh}^2}}\left\| \frac{v(x_{kh}^w,kh)-\frac{\dot{\beta}_{kh}}{\beta_{kh}}x_{kh}^w}{\dot{\alpha}_{kh}-\frac{\dot{\beta}_{kh}}{\beta_{kh}}\alpha_{kh}} - x_1^w \right\|^2 = -\frac{\tilde{\beta}}{2}\left\| \frac{v(x_{kh}^w,kh)-\frac{\dot{\beta}_{kh}}{\beta_{kh}}x_{kh}^w}{\frac{\alpha_{kh}}{\alpha_{kh}}\beta_{kh}-\dot{\beta}_{kh}} - \frac{\alpha_{kh}}{\beta_{kh}}x_1^w \right\|^2. \tag{232}$$

Thus, the Diffusion-DPO loss for Flow Matching reads

$$L_{\text{DPO}}(\theta) = -\mathbb{E}_{(x_1^w,x_1^l)\sim\mathcal{D},k\sim U[0,K],x_{kh}^w\sim q(x_{kh}^w|x_1^w),x_t^l\sim q(x_{kh}^l|x_1^l)}\Big[$$
$$\log S\Big(-\tfrac{\tilde{\beta}}{2}\big(\big\| \tfrac{v_\theta(x_{kh}^w,kh)-\frac{\dot{\beta}_{kh}}{\beta_{kh}}x_{kh}^w}{\frac{\alpha_{kh}}{\alpha_{kh}}\beta_{kh}-\dot{\beta}_{kh}} - \tfrac{\alpha_{kh}}{\beta_{kh}}x_1^w \big\|^2 - \big\| \tfrac{v_{\text{ref}}(x_{kh}^w,kh)-\frac{\dot{\beta}_{kh}}{\beta_{kh}}x_{kh}^w}{\frac{\alpha_{kh}}{\alpha_{kh}}\beta_{kh}-\dot{\beta}_{kh}} - \tfrac{\alpha_{kh}}{\beta_{kh}}x_1^w \big\|^2$$
$$- \big(\big\| \tfrac{v_\theta(x_{kh}^l,kh)-\frac{\dot{\beta}_{kh}}{\beta_{kh}}x_{kh}^l}{\frac{\alpha_{kh}}{\alpha_{kh}}\beta_{kh}-\dot{\beta}_{kh}} - \tfrac{\alpha_{kh}}{\beta_{kh}}x_1^l \big\|^2 - \big\| \tfrac{v_{\text{ref}}(x_{kh}^l,kh)-\frac{\dot{\beta}_{kh}}{\beta_{kh}}x_{kh}^l}{\frac{\alpha_{kh}}{\alpha_{kh}}\beta_{kh}-\dot{\beta}_{kh}} - \tfrac{\alpha_{kh}}{\beta_{kh}}x_1^l \big\|^2\big)\big)\Big)\Big], \tag{233}$$

(Wallace et al., 2023a, Sec. 5.1) claim that $\beta \in [2000, 5000]$ yields good performance on Stable Diffusion 1.5 and Stable Diffusion XL-1.0, which if we translate to our notation corresponds to $\tilde{\beta} \in [4000, 10000]$.

When we have access to the reward function $r$, instead of a winning sample $x_1^w$ and a losing sample $x_1^l$, we have a pair of samples $(x_1^a, x_1^b)$ with winning weights $S(r(x_1^a) - r(x_1^b)) = \frac{1}{1+\exp\left(r(x_1^b)-r(x_1^a)\right)}$, $S(-(r(x_1^a) - r(x_1^b))) = \frac{1}{1+\exp\left(-(r(x_1^b)-r(x_1^a))\right)}$. Hence, the loss (233) be-

comes:

$$L_{\text{DPO}}(\theta) = -\mathbb{E}_{(x_1^a, x_1^b) \sim \mathcal{D}, k \sim U[0,K], x_{kh}^a \sim q(x_{kh}^a | x_1^a), x_t^b \sim q(x_{kh}^b | x_1^b)} \left[ \sum_{s \in \{\pm 1\}} S(s(r(x_1^a) - r(x_1^b))) \times \right.$$

$$\log S\Big( -\frac{s\tilde{\beta}}{2} \Big( \Big\| \frac{v_\theta(x_{kh}^a, kh) - \frac{\dot{\beta}_{kh}}{\beta_{kh}} x_{kh}^a}{\frac{\dot{\alpha}_{kh}}{\alpha_{kh}} \beta_{kh} - \dot{\beta}_{kh}} - \frac{\alpha_{kh}}{\beta_{kh}} x_1^a \Big\|^2 - \Big\| \frac{v_{\text{ref}}(x_{kh}^a, kh) - \frac{\dot{\beta}_{kh}}{\beta_{kh}} x_{kh}^a}{\frac{\dot{\alpha}_{kh}}{\alpha_{kh}} \beta_{kh} - \dot{\beta}_{kh}} - \frac{\alpha_{kh}}{\beta_{kh}} x_1^a \Big\|^2$$

$$\left. - \Big( \Big\| \frac{v_\theta(x_{kh}^b, kh) - \frac{\dot{\beta}_{kh}}{\beta_{kh}} x_{kh}^b}{\frac{\dot{\alpha}_{kh}}{\alpha_{kh}} \beta_{kh} - \dot{\beta}_{kh}} - \frac{\alpha_{kh}}{\beta_{kh}} x_1^b \Big\|^2 - \Big\| \frac{v_{\text{ref}}(x_{kh}^b, kh) - \frac{\dot{\beta}_{kh}}{\beta_{kh}} x_{kh}^b}{\frac{\dot{\alpha}_{kh}}{\alpha_{kh}} \beta_{kh} - \dot{\beta}_{kh}} - \frac{\alpha_{kh}}{\beta_{kh}} x_1^b \Big\|^2 \Big) \Big) \Big) \right].$$

$$(234)$$

We want to emphasize that despite the similarities, even though the loss $L_{\text{DPO}}$ that we use (equation (234)) is very similar to the one implemented by Wallace et al. (2023a), the preference data pairs that we use are very different from theirs. We sample the preference data from the current model, which results in imperfect samples, while they consider off-policy, high-quality, curated preference samples. The reason for this discrepancy is that the starting point of our work is a reward model, not a set of preference data, and we only benchmark against approaches that leverage reward models for an apples-to-apples comparison. Our experimental results on DPO (Tab. 2, Fig. 5, Tab. 3) show that the resulting model performs like the base model, or a bit worse according to some metrics. Hence, we conclude that DPO is not a competitive alternative for on-policy fine-tune when the base model is not already good.

## H    EXPERIMENTAL DETAILS

Unless otherwise specified, we used the same hyperparameters across all fine-tuning methods. Namely, we used:

- $K = 40$ timesteps.
- Adam optimizer with learning rate $2 \times 10^{-5}$ and parameters $\beta_1 = 0.95$, $\beta_2 = 0.999$, $\epsilon = 1 \times 10^{-8}$, weight decay $1 \times 10^{-2}$, gradient norm clipping value 1. For Discrete Adjoint, these hyperparameters resulted in fine-tuning instability (see Tab. 6); the results that we report in all other tables for Discrete Adjoint were obtained with learning rate $1 \times 10^{-5}$.
- Bfloat16 precision.
- Effective batch size 40; for each run we used two 80GB A100 GPUs with batch size 20 each.
- A set of 40k fine-tuning prompts taken from a licensed dataset consisting of text and image pairs (note that we disregarded the images). Thus, each epoch lasts 1000 iterations; see the total amount of fine-tuning iterations for each algorithm in Tab. 3. For each of the three runs that we perform for each data point that we report, the set of 40k prompts is sampled independently among a total set of 100k prompts.

### H.1    NOISE SCHEDULE DETAILS

Since we use $K = 40$ discretization steps, the timesteps are $t \in \{0, 0.025, 0.05, 0.075, 0.1, \ldots, 0.95, 0.975\}$. To sample $X_{t+h}$ from $X_t$ we use equation (215). We use the choices $\alpha_t = t$, $\beta_t = 1 - t$, which means that $\sigma(t) = \sqrt{2\beta_t(\frac{\dot{\alpha}_t}{\alpha_t}\beta_t - \dot{\beta}_t)} = \sqrt{2(1-t)(\frac{1-t}{t} + 1)} = \sqrt{\frac{2(1-t)}{t}}$.

Note that if we plug $t = 0$ into this expression, we obtain infinity, and if we plug $t \lessapprox 1$, we obtain $\sigma(t) \approx 0$. For obvious reasons, the former issue requires a fix: we simply add a small offset to the denominator of $\sigma(t)$, replacing $\sqrt{1/t}$ by $\sqrt{1/(t+h)}$ (note that $h := 1/K = 0.025$). But the latter issue is also not completely satisfactory from a practical standpoint, because looking at the adjoint matching loss (25), we observe that $u(X_t^{\bar{u}}, t)$ is trained to approximate the conditional expectation of $\sigma(t)^\top \tilde{a}(t; X^{\bar{u}})$. Thus, if we set $\sigma(t)$ very close to zero for $t \lessapprox 1$, we are forcing the control $u$ to be close to zero as well, or equivalently preventing $v^{\text{finetune}}$ from deviating from $v^{\text{base}}$. While

this is the right thing to do from a theoretical perspective, we concluded experimentally that setting $\sigma(t)$ just slightly larger results in substantially faster fine-tuning, thanks to the additional leeway provided to $v^{\text{finetune}}$ to deviate from $v^{\text{base}}$. In particular, we added a small offset to the factor $1-t$ in the numerator $1-t$ of $\sigma(t)$: we replaced $1-t$ by $1-t+h$. Thus, the expression that we used to compute the diffusion coefficient in our experiments is

$$\sigma(t) = \sqrt{\tfrac{2(1-t+h)}{t+h}}. \tag{235}$$

When solving the lean adjoint ODE (26)-(27) backwards in time via the Euler scheme (216), the timesteps we use are $t \in \{1, 0.975, 0.95, 0.925, 0.9, \ldots, 0.05, 0.025\}$. We do not actually initialize the adjoint state as $\nabla_x g(X_1)$, but rather as $\nabla_x g(\hat{X}_1)$, where $\hat{X}_1 := X_{1-h} + h v^{\text{base}}(X_{1-h}, 1-h)$. That is, $\hat{X}_1$ is obtained by performing a final noiseless update, instead of using noise $\sigma(1-h) = \sqrt{4h}$ given by equation (235). The reason for this is that the regular final iterate $X_1$ contains some noise that was added in the final step, and that can distort the gradient $\nabla_x g(X_1)$. By setting $\tilde{a}(1; \boldsymbol{X}) = \nabla_x g(X_1)$, we get rid of this bias. Note that in the continuous time limit $h \to 0$, $\hat{X}_1 = X_1$, which means that this small trick is consistent.

## H.2 Selection of gradient evaluation timesteps

In Alg. 1, equation (217), we state that the term $\left\| \frac{2}{\sigma(t)} \left( v_\theta^{\text{finetune}}(X_t, t) - v^{\text{base}}(X_t, t) \right) + \sigma(t)\tilde{a}_t \right\|^2$ must be computed for all $K$ steps in $\{0, \ldots, 1-h\}$. However, the gradient signal provided by backpropagating through this expression for consecutive times $t$ and $t+h$ is quite similar. In the interest of computational efficiency, we sample a subset $\mathcal{K}$ of timesteps, and we only compute and backpropagate the terms $\left\| \frac{2}{\sigma(t)} \left( v_\theta^{\text{finetune}}(X_t, t) - v^{\text{base}}(X_t, t) \right) + \sigma(t)\tilde{a}_t \right\|^2$ for those timesteps. We construct $\mathcal{K}$ by sampling ten timesteps uniformly without repetition among $\{0, \ldots, 0.725\}$, and always sampling the last ten timesteps $\{0.75, \ldots, 0.975\}$. This is because fine-tuning the last ten steps (25% of the total) well is critical for good empirical performance, while the initial steps are not as important.

## H.3 Loss function clipping: the LCT hyperparameter

Note that the magnitude of $\sigma(t)^\mathsf{T} a(t; \boldsymbol{X}^{\bar{u}}, \bar{u})$ is much larger for times $t \gtrsim 0$ than for times $t \lesssim 1$. The reason is two-fold:

- As discussed in App. H.1, $\sigma(t)$ is much larger for $t \gtrsim 0$ than for $t \lesssim 1$.
- The magnitude of the lean adjoint state $\tilde{a}$ grows roughly exponentially as $t$ goes backward in time. In fact, if we assumed that $\nabla_x b(X_t, t)$ is constant in time, this statement would be exact.

Observe that when $\sigma(t)^\mathsf{T} a(t; \boldsymbol{X}^{\bar{u}}, \bar{u})$ is large, the gradient $\nabla_\theta \left\| \frac{2}{\sigma(t)} \left( v_\theta^{\text{finetune}}(X_t, t) - v^{\text{base}}(X_t, t) \right) + \sigma(t)\tilde{a}_t \right\|^2$ also has a high magnitude. Including such terms in our gradient computation decreases the signal to noise ratio of the gradient. Even more so, as discussed in App. H.2 for good practical performance it is critical to get a good gradient signal from the last 25% steps. Hence, including the high-magnitude terms for $t \lesssim 0$ in our gradients can muffle these other important, low-magnitude terms.

To fix this issue, we clip the terms such that $\left\| \frac{2}{\sigma(t)} \left( v_\theta^{\text{finetune}}(X_t, t) - v^{\text{base}}(X_t, t) \right) + \sigma(t)\tilde{a}_t \right\|^2 >$ LCT, where LCT stands for the loss clipping threshold. That is, the adjoint matching loss that we use in our experiments is of the form:

$$\hat{\mathcal{L}}_{\text{Adj-Match}}(\theta) = \sum_{t \in \mathcal{K}} \min \left\{ \text{LCT}, \left\| \tfrac{2}{\sigma(t)} \left( v_\theta^{\text{finetune}}(X_t, t) - v^{\text{base}}(X_t, t) \right) + \sigma(t)\tilde{a}_t \right\|^2 \right\}, \tag{236}$$

where $\mathcal{K}$ is the random timestep subset described in App. H.2.

For adjoint matching, we set $\text{LCT} = 1.6 \times \lambda^2$. Remark that LCT needs to grow quadratically with $\lambda$, because the magnitude of the lean adjoint $\tilde{a}$ grows quadratically with $\lambda$. We set the constant 1.6 through experimentation; all or almost all of the terms for the last ten timesteps fall below LCT, but only a fraction of the terms ($\approx 25\%$) for the first ten steps fall below LCT. The constant for LCT is a relevant hyperparameter that needs to be tuned to obtain a similar behavior.

We also used loss function clipping on the continuous adjoint loss. For that loss we set LCT $=$ $1600 \times \lambda^2$. The reason is that the magnitude of the regular adjoint states is significantly larger than the magnitude of the lean adjoint states (which is a big reason why adjoint matching outperforms the continuous adjoint).

## H.4    Computation of evaluation metrics

We used the `open_clip` library (Ilharco et al., 2021) to compute ClipScores. We computed Clip-Score diversity as the variance of Clip embeddings of 40 generations for a given prompt, averaged across 25 prompts. Namely,

$$\text{ClipScore\_Diversity} = \frac{1}{40} \sum_{k=1}^{40} \frac{2}{25 \cdot 24} \sum_{1 \leq i < j \leq 25} \|\text{Clip}(g_i^k) - \text{Clip}(g_j^k)\|^2, \qquad (237)$$

where $g_i^k$ denotes the $i$-th generation for the $k$-th prompt.

We used the `transformers` library to compute the PickScore processor and model (Kirstain et al., 2023). PickScore diversity is computed in analogy with ClipScore diversity.

We used the `hps` library to compute values of Human Preference Score v2 (Wu et al., 2023b).

To compute Dreamsim diversity we use the `dreamsim` library (Fu et al., 2023). Dreamsim diversity is computed in analogy with ClipScore diversity.

## H.5    Remarks on computational costs

Observe from the figures reported in Tab. 3 that the per iteration wall-clock time of Adjoint Matching (156 seconds) is very similar to that of the Discrete Adjoint loss (152 seconds). We report hardware and hyperparameter details at the beginning of App. H. The reason for which both algorithms take a similar time is that they perform a similar amount of forward and backward passes on the flow matching model and the reward model. Namely, for each sample in the batch, both algorithms perform $K$ forward passes on the flow model to obtain the trajectories. In order to compute the gradient of the training loss, the Discrete Adjoint loss does $K$ additional forward passes to evaluate the base flow model, one forward and backward pass on the reward model, and $K$ backward passes on the current flow model, which typically use gradient checkpointing to avoid memory overflow. In the case of Adjoint Matching, solving the lean adjoint ODE requires one forward and backward pass on the reward model, and $K$ backward passes on the base flow model. Finally, computing the gradient of the loss takes $K/2$ additional backward passes if we evaluate at only half of the timesteps as we do, although this computation is much quicker because it can be fully parallelized.

Meanwhile, computing the gradient of the Continuous Adjoint loss takes 204 seconds. With respect to Adjoint Matching, Continuous Adjoint performs additional backward passes to compute the gradients $\nabla_{X_t} \|u(X_t, t)\|^2$ when solving the adjoint ODE. Finally, we observe that models that directly fine-tune the reward are quicker, but that comes with its own set of issues that we discuss throughout the paper.

## H.6    Remarks on number of sampling timesteps

In our experiments and all baselines, we used 40 timesteps in the fine-tuning procedure ($h = 1/40$ in Alg. 1). The experiments reported in all tables and figures except for Tab. 8 were performed at 40 inference timesteps. In Tab. 8 (App. A), we show experimental results at 10, 20, 40, 100, and 200 inference timesteps, for the base model and the models fine-tuned with adjoint matching and DRaFT-1. We make the following observations about the results:

- The metrics for Adjoint Matching at 100 and 200 timesteps are statistically equal to the ones for 40 timesteps, with slight increases in Dreamsim diversity. This suggests that fine-tuning at large numbers of timesteps is a good idea if we want to perform inference at a large number of timesteps, as otherwise the capabilities of the model are limited by the number of fine-tuning timesteps instead of the inference compute. Also, at 100 and 200 timesteps the difference in performance of Adjoint Matching relative to DRaFT-1 increases.

- The metrics for Adjoint Matching at 10 and 20 timesteps are worse than at 40 timesteps, especially for 10. The difference in performance between Adjoint Matching and DRaFT-1

vanishes at 10 timesteps for all metrics except for diversity, for which Adjoint Matching is still clearly better.

