# OpenReview forum: "Adjoint Matching: Fine-tuning Flow and Diffusion Generative Models with Memoryless Stochastic Optimal Control"
_ICLR.cc/2025/Conference — ICLR 2025 Spotlight_

### Official Review · Reviewer_pgS7 · 2024-11-01

**Soundness:** 4
**Presentation:** 3
**Contribution:** 3
**Rating:** 8
**Confidence:** 4

**Summary:**

The authors propose a SOC formulation for fine-tuning flow-based generative models, demonstrating that naive approaches can introduce inherent bias. They introduce a memoryless noise schedule to ensure convergence to the tilted distribution and present the Adjoint Matching objective as a scalable training objective for SOC problems. Their comparisons show improved generalization, consistency, and diversity in the fine-tuned generative models.

**Strengths:**

The concept of using stochastic optimal control (SOC) to fine-tune diffusion-based generative models is not new, as [1] formulates fine-tuning for diffusion models (specifically forward-reverse models) through Doob's h-transform, which involves optimally controlled diffusion processes. The novelty lies in:

* Reformulating fine-tuning of flow-matching based generative models as an SOC problem, the authors introduce a suitable cost functional for the control policy parameterized by a training neural network. Additionally, they incorporate a memoryless noise schedule to ensure factorizability, thereby eliminating inherent bias.
* Proposing an adjoint-matching objective to solve the above SOC problem. It is well known that solving SOC problems is computationally challenging due to the need for continuous gradient graph caching. Extending the adjoint methods commonly used in dynamical learning as a dynamic solver for SOC objectives is, in my view, a significant contribution.


```
[1] Denker et al., Efficient Finetuning of Conditional Diffusion Models by Learning the Generalised h-transform.
```

**Weaknesses:**

* While I may have missed it in the appendix, it appears that the experiments in this paper primarily address the quality of fine-tuning, without providing quantitative results on aspects of the proposed adjoint objective, such as convergence plots or memory usage statistics comparing the adjoint matching loss to traditional SOC objectives, as shown in studies like [2, 3]. From a theoretical standpoint, although the author suggests that the approach could be sufficiently applied, it would strengthen the claim to include experiments validating the numerical effectiveness of the new SOC objective.


```
[2] Nüsken and Ritcher, Solving high-dimensional Hamilton-Jacobi-Bellman PDEs using neural networks: perspectives from the theory of controlled diffusions and measures on path space.
[3] Domingo-Enrich et al., Stochastic Optimal Control Matching.
```

**Questions:**

See weaknesses.

---

> ### Author Response · Authors · 2024-11-25
> **Reply to Reviewer pgS7**
>
> We thank the reviewer for their comments, and provide answers below.
>
> - **“it would strengthen the claim to include experiments validating the numerical effectiveness of the new SOC objective.”**
>
> We did run experiments using the toy settings and the models of Domingo-Enrich et al. [3]. Namely, these are experiments where the optimal control is either known analytical or can be computed, which allows for the evaluation of the control $L^2$ error between the learned control and the optimal one. The results we obtained showed that on the simple settings of Domingo-Enrich et al. [3], SOCM (their algorithm) outperforms all others, including Adjoint Matching, which attains a control $L^2$ error on par with Continuous Adjoint. These settings are relatively low-dimensional ($d=10,20$), and have low-magnitude state and terminal cost. The reason that Domingo-Enrich et al. [3], restrict their focus to such problems is that SOCM has an importance weight $\alpha$ that blows up when either the dimension or the magnitudes of the costs are higher; see Fig. 4 of Domingo-Enrich et al., [3] and the explanation below it, or the discussion in Appendix F.1.2 of our work.  Thus, we found that such experimental settings are not reflective of the realistic SOC problems one must tackle in high-dimensional machine learning, such as reward fine-tuning generative models. Given that our work is already extensive and multifaceted, we decided not to include such experiments in it. Adding such experiments would distract the reader from the single-minded goal of reward fine-tuning. A complete theoretical analysis between different SOC algorithms is worth a more focused look in a future work.

---

> > ### Comment · Reviewer_pgS7 · 2024-12-02
> >
> > Thank you for your response. I am somewhat convinced by the reasoning behind the decision to exclude the experiments in this paper.However, I still believe that experimental validation is essential to fully establish the validity of the proposed objective function. That said, this paper makes significant contributions, such as fine-tuning flow-based generative models and proposing an SOC objective well-suited for high-dimensional problems, which warrant to publication. Therefore, I have raised my score.

---

### Official Review · Reviewer_MA7f · 2024-11-03

**Soundness:** 4
**Presentation:** 3
**Contribution:** 3
**Rating:** 8
**Confidence:** 3

**Summary:**

This paper aims to fine-tune a pre-trained diffusion model. The contribution of this paper is two-fold:

First, the paper explains the problem of overoptimization when fine-tuning diffusion models based on a learned reward function. To avoid the initial value function bias problem, the paper proposes using a memoryless noise schedule, which is defined as a noise schedule that ensures that the joint distribution over the initial and final states is independent.

Second, the paper introduces (lean) adjoint matching for stochastic optimal control methods which resolves memory constraints of the discrete adjoint method and makes the continuous adjoint method more scalable by making the simulation of the adjoint ODE cheaper to compute.

**Strengths:**

- The paper is well-written and structured.
- Strengths of memory-less noise schedule:
    - The proposed approach is a less complex and provides an arguably more elegant solution to the initial value function bias compared to the work by Uehara et. al. [1].
    - The proposed approach is compatible with both DDPM and Flow/Bridge-Matching models.
- Strengths of (lean) adjoint matching:
    - Simple regression-based objective with circumvents memory problems associated with the discrete adjoint method
    - Simulating the adjoint ODE does not require control evaluations, making it more scalable than the continuous adjoint method.
    - Compatible with general SOC problems.
- Numerical evaluations show that the proposed approach outperforms competing methods on a variety of evaluation criteria.

**Weaknesses:**

- Weaknesses of the memory-less noise schedule:
    - The statement “however, there does not yet exist a simple approach which actually provably generates from the tilted distribution” is, to the best of my knowledge not true: Take for example the approach in [2, 3] and use the base measure as reference process. Then, we have the terminal cost

    $$
    g(X_1) =p^{\text{base}}(X_1)/p_{\text{target}}(X_1) = p^{\text{base}}(X_1)/(p^{\text{base}}(X_1)\exp(r(X_1)) = 1/\exp(r(X_1))
    $$

    which es ensures that

    $$
    p_{\text{target}}(X_1) = \exp(r(X_1)).
    $$

    The ‘memoryless’ property thus eludes to the fact, that we need a time-reversal of the base process. A more general discussion compared to [2, 3] can be found in [4, 5].

- Weaknesses of (lean) adjoint matching:
    - It is not clear (at least to me) which objective function lean adjoint matching is optimizing.
    - Lean/Continous Adjoint-matching requires simulating another differential equation which may result in computational overhead.
- Code is not public

**Questions:**

- Did the authors compare the proposed approach to the work by Uehara et. al. [1]?
- Did the authors test how adjoint matching works with only a few diffusion steps?
- How did the authors select the time points to discretize? Uniform in the interval [0,1]?
- Do the authors have an intuition as to why the lean adjoint matching objective outperforms the discrete/continuous adjoint matching objective? I read the explanation that the continuous adjoint needs clipping and is unstable, but are there other reasons?
- Are the authors planning to make the code public?

---

- [1] Uehara et al. (2024). Fine-Tuning of Continuous-Time Diffusion Models as Entropy-Regularized Control. *arXiv preprint arXiv:2402.15194*
- [2] Vargas, F., Grathwohl, W., & Doucet, A. (2023). Denoising diffusion samplers.  *ICLR 2023*. 2024.
- [3] Zhang, Q., & Chen, Y. (2021). Path integral sampler: a stochastic control approach for sampling. *arXiv preprint arXiv:2111.15141*
- [4] Nusken, Nikolas, et al. "Transport meets variational inference: Controlled Monte Carlo diffusions." *The Twelfth International Conference on Learning Representations: ICLR 2024*. 2024.
- [5] Richter, L., & Berner, J. (2023). Improved sampling via learned diffusions. *ICLR 2024*. 2024.

---

> ### Author Response · Authors · 2024-11-25
> **Reply to Reviewer MA7f (1)**
>
> We thank the reviewer for their comments, and provide answers below.
>
> - **“Weaknesses of the memory-less noise schedule: The statement “however, there does not yet exist a simple approach which actually provably generates from the tilted distribution” is, to the best of my knowledge not true [...]”**
>
> Our work is not the first one that tackles the problem of reward fine-tuning for diffusion processes, and it is not the first one that uses memoryless noise schedules (previous works did not use this terminology). However, the previous works that the reviewer refers to consider specific interpolation paths for the base model: [2] focuses on the denoising diffusion path ($\alpha_t = \sqrt{t}$, $\beta_t = \sqrt{1-t}$) and [3] focuses on the Föllmer drift path ($\alpha_t = t$, $\beta_t = \sqrt{t(1-t)}$). We are the first work to single out memorylessness as the unified condition to guarantee that the optimal process of the SOC problem will generate samples from the right distribution: the initial and final samples of the base process need to be independent. In our view, this is an important contribution, because it really simplifies the fine-tuning pipeline when compared to approaches that do not take it into account, such as the one of Uehara et al. [1]. While the fine-tuning approach of [1] is also generic in that it can work for multiple generative paths, it is arguably not simple, because it involves learning a value function, and then solving two SOC problems.
>
> Moreover, the insights of our Prop. 1 and Thm. 1 are clearly novel with respect to [2,3,4,5]: we show that the base process is memoryless provided that the noise schedule is “large” enough, but if we want to be able to perform inference on the fine-tuned model using a different noise schedule than the fine-tuning one (such as the commonly used zero noise schedule), we “must” use a unique memoryless noise schedule, which is the one that corresponds to the time reversal of the explicit Ornstein Uhlenbeck process (in [2]) or Brownian bridge process (in [3]). The proof techniques for our results are very different from techniques used in the literature, which makes them interesting from a theoretical perspective as well. Finally, we show samples using the zero noise schedule and the memoryless noise schedule in Figures 10 & 11, so we have shown that the memoryless noise schedule samples are both qualitatively and quantitatively worse. It is thus important to convert back to the zero noise schedule after fine-tuning, which is actually the main reason we need to use the memoryless noise schedule. This is a novel contribution of ours. We emphasize that the contributions we make in Subsec. 3.3 are highly novel and are not mere rewriting of previous works.
>
> For these reasons, we have reformulated this sentence to be more precise: “there does not yet exist a simple, generic approach [...]”.
>
> - **“Did the authors compare the proposed approach to the work by Uehara et. al. [1]?”**
>
> We do compare our approach to the work of Uehara et al. [1] (see App. B, lines 1318-1320) in words, but not experimentally. The work by Uehara et al. [1] tackles the same problem as ours, but instead of starting from a memoryless base process, it learns the initial value function $V(X_0,0)$, then learns a sampler for $\exp(-V(X_0,0))$, and then solves the fine-tuning SOC problem by setting $p_0$ to be the approximation of $\exp(-V(X_0,0))$ that has been learned. We did not compare our method with theirs experimentally, as their pipeline requires a lot more steps, and their memory usage is substantially higher because they need to store the value function model and the sampler for $\exp(-V(X_0,0))$ in memory as well. Hence, an apples-to-apples comparison with our method and all other baselines is very challenging.

---

> > ### Author Response · Authors · 2024-11-25
> > **Reply to Reviewer MA7f (2)**
> >
> > - **“Did the authors test how adjoint matching works with only a few diffusion steps?”**
> >
> > In our experiments and all baselines, we used 40 timesteps in the fine-tuning procedure ($h=1/40$ in Alg. 1). The experiments reported in all tables and figures except for Table 7 were performed at 40 inference timesteps. In Table 7 (App. A), we show experimental results at 10, 20, 40, 100, and 200 inference timesteps, for the base model and the models fine-tuned with adjoint matching and DRaFT-1. Some observations on the results:
> > - The metrics for Adjoint Matching at 100 and 200 timesteps are statistically equal to the ones for 40 timesteps, with slight increases in Dreamsim diversity metric. This suggests that fine-tuning at large numbers of timesteps may be a good idea if we want to perform inference at a large number of timesteps, as otherwise the capabilities of the model may be limited by the number of fine-tuning timesteps instead of the inference compute. Also, at 100 and 200 timesteps the difference in performance of Adjoint Matching relative to DRaFT-1 increases.
> > - The metrics for Adjoint Matching at 10 and 20 inference timesteps are worse than at 40 timesteps, especially for 10. The difference in performance between Adjoint Matching and DRaFT-1 vanishes at 10 timesteps for all metrics except for diversity, for which Adjoint Matching is still clearly better.
> >
> > We included this explanation in a newly created subsection of Appendix G (“Remarks on number of sampling timesteps”).
> >
> > - **“How did the authors select the time points to discretize? Uniform in the interval [0,1]?”**
> >
> > Yes, we choose the 40 timesteps to be [0,0.025,0.05,…,0.95,0.975]. We explain this in Appendix H.2.
> >
> > - **“Do the authors have an intuition as to why the lean adjoint matching objective outperforms the discrete/continuous adjoint matching objective? I read the explanation that the continuous adjoint needs clipping and is unstable, but are there other reasons?”**
> >
> > We included additional plots in the newly created Figure 5 in App. A. For different algorithms and throughout fine-tuning iterations, the three subfigures show (i) the average value of ImageReward (without reward multiplier), which is the reward model that we use, (ii) the average value of the control term (the integral of the norm squared of the gradient), and (iii) the average value of ClipScore, which we treat as an evaluation metric.
> >
> > Observe that the control terms for Adjoint Matching and Continuous Adjoint at reward multiplier $\lambda=12500$ stabilize at similar values, which is a sanity check for the fact that both algorithms are targeting the same solution. However, the ImageReward and ClipScore values for Continuous Adjoint are substantially lower than for Adjoint Matching, and their fluctuations across time are much larger. We attribute this to the fact that gradient variance is much higher for Continuous Adjoint than for Adjoint Matching, which is a consequence of the terms present in the adjoint ODE but not in the lean adjoint ODE. In fact, high gradient variance is one of the main failure points of classical RL, and the main solution in RL is to take larger batch sizes. In the case of SOC, we argue that another way to mitigate high gradient variance is to replace the Continuous Adjoint loss by the Adjoint Matching loss (even if their gradients are not equal in expectation).
> >
> > - **“Are the authors planning to make the code public?”**
> >
> > Yes, we will make the code public once the paper gets accepted.

---

> > > ### Author Response · Authors · 2024-11-25
> > > **Reply to Reviewer MA7f (3)**
> > >
> > > - **“Weaknesses of (lean) adjoint matching:
> > > It is not clear (at least to me) which objective function lean adjoint matching is optimizing."**
> > >
> > > Unlike the Continuous Adjoint / Basic Adjoint Matching loss, which is optimizing the control objective, Adjoint Matching is not optimizing any functional on the control. The critical subtlety is that even though we express the Adjoint Matching update as the gradient of the loss in equation 25, we are not formally optimizing this loss because of the presence of the stopgrad operator.  Instead, Adjoint Matching is a consistency method designed to solve a fixed point equation: $u(x,t) = - \sigma(t)^{\top} \nabla_x J(x,t)$. This fixed point equation has a unique solution, which is the optimal control. To arrive to this statement, which is formally stated as Proposition 7 in Appendix F.4, the approach that we follow is:
> > >
> > > - First, in Proposition 2 we show that all critical points of the Basic Adjoint Matching loss solve the fixed point equation $u(x,t) = - \sigma(t)^{\top} \nabla_x J(x,t)$, and that the solution of this fixed point equation is unique and equal to the optimal control.
> > > - Second, in the proof of Proposition 7 we show that critical points of the Adjoint Matching loss are critical points of the Basic Adjoint Matching loss, which concludes the proof.
> > >
> > > - **"Lean/Continous Adjoint-matching requires simulating another differential equation which may result in computational overhead.”**
> > >
> > > Note that in our first submission, we reported the total fine-tuning runtimes and number of iterations of Adjoint Matching, the Continuous and Discrete Adjoint and all the baselines in Table 3 (Appendix A). The comparisons are apples-to-apples, because the base generative model, the reward model, and the batch size are equal for all algorithms. We adjust the number of iterations such that total runtimes are similar for all algorithms. Already in our first submission, we had a reference pointing this out in the Evaluation results paragraph of the Experiments section, but we did not add an extensive comment on the computation cost of the algorithms. We have now added the following explanations in a newly created subsection of Appendix H (“Remarks on computational costs”).
> > >
> > > Observe from the figures reported that the per iteration wall-clock time of Adjoint Matching (156 seconds) is very similar to that of the Discrete Adjoint loss (152 seconds). The reason is that both algorithms perform a similar amount of forward and backward passes on the flow matching model and the reward model. Namely, for each sample in the batch, both algorithms perform $K$ forward passes on the flow model to obtain the trajectories. In order to compute the gradient of the training loss, the Discrete Adjoint loss does $K$ additional forward passes to evaluate the base flow model, one forward and backward pass on the reward model, and $K$ backward passes on the current flow model, which typically use gradient checkpointing to avoid memory overflow. In the case of Adjoint Matching, solving the lean adjoint ODE requires one forward and backward pass on the reward model, and $K$ backward passes on the base flow model. Finally, computing the gradient of the loss takes $K/2$ additional backward passes if we evaluate at only half of the timesteps as we do, although this computation is much quicker because it can be fully parallelized.
> > >
> > > Meanwhile, computing the gradient of the Continuous Adjoint loss takes 204 seconds. With respect to Adjoint Matching, Continuous Adjoint performs additional backward passes to compute the gradients $\nabla_{X_t} \|u(X_t,t)\|^2$ when solving the adjoint ODE. Finally, we observe that models that directly fine-tune the reward are quicker, but that comes with its own set of issues that we discuss throughout the paper.

---

> > > > ### Comment · Reviewer_MA7f · 2024-11-26
> > > > **Reply to authors**
> > > >
> > > > I thank the authors for addressing my questions. In response, I increase my score.

---

### Official Review · Reviewer_7e67 · 2024-11-04

**Soundness:** 3
**Presentation:** 3
**Contribution:** 3
**Rating:** 6
**Confidence:** 4

**Summary:**

This paper studies reward-based fine-tuning for diffusion models. The authors frame the reward fine-tuning problem as stochastic optimal control, and point out an “initial value function bias” problem that exists in previous RLHF fine-tuning approaches. The authors propose using a memoryless noise schedule for fine-tuning in order to turn the learned distribution into the desired reward-tilted distribution without bias. Furthermore, a novel algorithm named adjoint matching is proposed to solve the stochastic optimal control problem. Experimental results show that fine-tuning a flow matching base model with adjoint matching outperforms baselines such as DRaFT, ReFL, and DPO.

**Strengths:**

This paper provides a theoretically sound framework for the reward fine-tuning problem, viewing it as a stochastic optimal control problem. The observation of the value function bias problem in previous approaches and proposal of using “memoryless” noise schedules are based on this view.

The proposed Adjoint Matching algorithm for SOC, casting it as a least-squares regression problem, is novel and effective.

This paper is well-written, clearly structured and easy to follow.

**Weaknesses:**

The main paper presented experimental results on fine-tuning a Flow Matching model, and provided pseudo-code for fine-tuning denoising diffusion models; it would be more convincing if results on denoising diffusion models are provided.

The experiments with classifier-free guidance do not seem comprehensive. It would be better if there are similar quantitative comparisons with other baselines than selected DRaFT-1.

**Questions:**

see above.

---

> ### Author Response · Authors · 2024-11-25
> **Reply to Reviewer 7e67**
>
> We thank the reviewer for their comments, and provide answers below.
>
> - **“it would be more convincing if results on denoising diffusion models are provided.”**
>
> As we mention in Section 2, both Flow Matching and continuous-time DDIM are described by Equation 5 and 6.  While our particular models were trained using the particular scheduler usually associated with Flow Matching, our unified notation includes all pre-training schedulers. In our initial experiments, we saw good initial results on Stable Diffusion 2 model, and then switched to using our own pretrained model. A reason for this choice is that flow matching and its choice of noise scheduler is increasingly adopted as the standard approach, e.g., in Stable Diffusion 3 or Flux, since there is evidence that it is superior to DDIM. Another reason for this is to use only kosher data for training; our models are trained only on data with proper licensing. This is needed to protect individual privacy and to adhere to ethical standards in data usage. As the reviewer points out, we also provide an algorithm box (Alg. 2) translating our approach to denoising diffusion models, and we do not expect significant differences.
>
> - **“The experiments with classifier-free guidance do not seem comprehensive. It would be better if there are similar quantitative comparisons with other baselines than selected DRaFT-1.”**
>
> We would appreciate additional explanation from the reviewer to clarify their remarks. In our reply below, (i) we explain why we believe that our experiments on classifier-free guidance are appropriate within the scope of our work, and (ii) we explain why we only include classifier guidance comparisons against DRaFT-1. If our reply does not satisfy the reviewer, we will be happy to follow up with a response once more detail has been provided.
>
> (i) We would appreciate some more detail on how the experiments on classifier-free guidance do not seem comprehensive. For reference, we explain our experiments relating to classifier-free guidance in the next two paragraphs, and would like to note that in our first submission we already included pointers and a brief discussion of the guidance results in Section 5 of the paper.
>
> In Figure 4 (App. A), we evaluate three different guidance settings: $w=0$ (no guidance), $w=1$, $w=4$. The numerical values can be found in Table 4 (App. A), which also shows guidance results for the pre-trained model. Higher guidance values yield higher Clipscore (prompt alignment) and HPS v2 (human quality perception) values at the expense of sample diversity. In fact, models fine-tuned with adjoint matching perform much better than those fine-tuned with DRaFT-1 at high guidance values ($w=4$), relative to their respective performance at $w=0$ and $w=1$. This is not surprising because Adjoint Matching preserves the theoretical connection of the fine-tuned vector field to the score of the fine-tuned distribution, which means that classifier-free guidance on the fine-tuned model is as principled as on the base model. That does not hold for directly fine-tuned models.
>
> Moreover, in Figure 5 (App. A) we also present a qualitative comparison of images generated with the same prompt and the same random seed at different guidance values. The generated images confirm that classifier-free guidance can be used successfully on models that have been fine-tuned using adjoint matching; and that high guidance values yield less diverse images that fit the prompt more narrowly. Also, in Figures 7 and 8 (App. A) we compare images generated with the pre-trained model, Adjoint Matching, and DRaFT-1.
>
> (ii) The reviewer suggests that it would be good to have guidance comparisons to other baselines beyond DRaFT-1. We want to point out that according to our experiments, DRaFT-1 is the best performing baseline fine-tuning algorithm, and that all other baselines perform worse, both quantitatively and qualitatively. Moreover, note that the classifier-free guidance results are answering a simple “what-if”, and classifier-free guidance is completely orthogonal to the fine-tuning method; we see that any model with poorer fine-tuned results also has poorer results with classifier-free guidance. For these two reasons, we thought that it was not necessary to run guidance experiments on alternative baselines.

---

> > ### Author Response · Authors · 2024-12-02
> > **Reminder to Reviewer 7e67**
> >
> > Given that the discussion period will end very soon, we invite Reviewer 7e67 to provide feedback on our response and consider adjusting their score if deemed appropriate.

---

### Official Review · Reviewer_4cB8 · 2024-11-04

**Soundness:** 4
**Presentation:** 3
**Contribution:** 4
**Rating:** 8
**Confidence:** 4

**Summary:**

This paper provides theorectical insights on why optimizing a KL-regularized reward objective (which is popular and dominant in RLHF for LLM) could lead to a bias in the optimal solution for diffusion models, and how to address this issue by a proper choice of noise schedule. The paper also provides solutions on how to solve the stochastic optimal control problem by adjoint methods, and provide a more efficient alternative than the classical methods and prove its equivalence. Empirical examples are further provided to show the algorithm effectiveness.

**Strengths:**

This paper is theorectically well written, with detailed introduction to the literature, explanation of motivations, several interesting theorems and propositions, and provide theorectically-driven approaches for diffusion models fine-tuning. Diffusion Models fine-tuning or RLHF for diffusion models are an important direction which already contributes to improving the performance to SOTA diffusion models. This paper indeed provides a novel model-based SOC method for diffusion models alignment, which is theorectically sound and also yields good performance for tuning Flow Matching based models.

**Weaknesses:**

The main weakness is that the paper can be benefited from more comparisons with baseline methods empirically, specifically, there lacks a baseline in experiments in directly optimizing objective (17) (though theorectically there is value bias as shown in the paper), using stochastic control methods like adjoint matching proposed in this paper. There is comparison for this in the synthetic examples in Figure 2, but more practical downstream tasks evaluations are needed to show that the noise schedule proposed in this paper can indeed yield better performance.

**Minors**:

1）On line 229-230, the expression "Dividing (14) by (15)" is odd as (14) is not an equality, "Plug in normalization constant (15) to (14)" might be better.

2）The adjoint method in Section 4 needs more introduction or discussion, or an earlier pointer to the part in appendix, including more clarification on what is adjoint on line 350-352, what is this loss on Equation (21); The reference in Appendix should be referred earlier instead of being deferred to until line 422.

3）In proposition 2, it is better to first define earlier of the adjoint matching objective instead of combining the definition with the proposition.

**Questions:**

What is the complexity/computation cost of computing/sampling adjoints in Equation (26) and (27)?

In addition, for Diffusion-DPO, how the preference pairs are sampled for further fine-tuning? Can authors explain why Diffusion-DPO tuned models yield a decrease in performance?

---

> ### Author Response · Authors · 2024-11-25
> **Reply to Reviewer 4cB8 (1)**
>
> We thank the reviewer for their comments, and provide answers below.
>
> - **“there lacks a baseline in experiments in directly optimizing objective (17) (though theorectically there is value bias as shown in the paper). [...] There is comparison for this in the synthetic examples in Figure 2, but more practical downstream tasks evaluations are needed [...].”**
>
> We tried different noise schedules to fine-tune in Figure 2 to exemplify that such noise schedules do not yield the desired target distribution. Thus, our theoretical framework provides theoretical evidence that the noise schedule that we need to use is the memoryless one.
>
> Apart from the synthetic experiments in Figure 2, we would like to remark that the initial version of our paper already contained experiments at different noise schedules: Table 6 in App. A shows that when we fine-tune with $\sigma(t)=1$, and perform inference at either $\sigma(t)=1$ or $\sigma(t)=0$, the quality metrics are significantly worse across the board (except for diversity). Following the intuition from Figure 2, using non-memoryless schedules yields a generated distribution which is closer to the initial one than desired.
> In case the author referred to baselines from existing papers, we want to point out that there are no existing works that solve the SOC problem with wrong noise schedules without further corrections: existing works either use arbitrary schedules, but learn additional objects to eliminate the value function bias (Uehara et al., 2024), or just avoid KL regularization (Clark et al., 2024, Xu et al., 2023).
>
> - **"1) On line 229-230, the expression "Dividing (14) by (15)" is odd as (14) is not an equality, "Plug in normalization constant (15) to (14)" might be better."**
>
> We thank the reviewer for pointing this out. We changed this sentence to: “Dividing the RHS of (14) by (15)...”. We also added a previous sentence that mentions “the right-hand side (RHS) of (14)”. Hence, the terminology RHS should be clear to the reader.
>
> - **"2）The adjoint method in Section 4 needs more introduction or discussion, or an earlier pointer to the part in appendix, including more clarification on what is adjoint on line 350-352, what is this loss on Equation (21); The reference in Appendix should be referred earlier instead of being deferred to until line 422."**
>
> We agree with the reviewer. We added an earlier pointer to the appendix: “There are two approaches which yield the same gradient in the small step size limit (see \Cref{sec:adjoint_method} for more detail): [...]”. We added clarification on the loss on Equation (21) by changing the previous sentence to “The gradient of the continuous adjoint loss is given by [...]”.
>
> - **"3）In proposition 2, it is better to first define earlier of the adjoint matching objective instead of combining the definition with the proposition."**
>
> Observe that Proposition 2 introduces the Basic Adjoint Matching loss, a preliminary to the Adjoint Matching loss (equations 25-27), which is our main algorithmic contribution. The Basic Adjoint Matching loss shares the gradient with the Continuous Adjoint loss and with the Discrete Adjoint loss (up to numerical errors in the latter case). We introduce it because it offers a novel perspective on an existing algorithm, and it opens the door to proving the convergence guarantee in Prop. 2. Given the space constraints, we do not want to further emphasize the role of the Basic Adjoint Matching loss, and opted for a compact explanation which combines the definition with the proposition.

---

> > ### Author Response · Authors · 2024-11-25
> > **Reply to Reviewer 4cB8 (2)**
> >
> > - **"What is the complexity/computation cost of computing/sampling adjoints in Equation (26) and (27)?"**
> >
> > In our first submission, we already reported the total fine-tuning runtimes and number of iterations of Adjoint Matching, the Continuous and Discrete Adjoint and all the baselines in Table 3 (Appendix A). The comparisons are apples-to-apples, because the base generative model, the reward model, and the batch size are equal for all algorithms. We adjust the number of iterations such that total runtimes are similar for all algorithms. Already in our first submission, we added a reference pointing this out in the Evaluation results paragraph of the Experiments section, but we did not add an extensive comment on the computation cost of the algorithms. We added the following explanations in a newly created subsection of Appendix H (“Remarks on computational costs”).
> >
> > Observe from the figures reported in Tab. 3 that the per iteration wall-clock time of Adjoint Matching (156 seconds) is very similar to that of the Discrete Adjoint loss (152 seconds). We report hardware and hyperparameter details at the beginning of Section H. The reason for which both algorithms take a similar time is that they perform a similar amount of forward and backward passes on the flow matching model and the reward model. Namely, for each sample in the batch, both algorithms perform $K$ forward passes on the flow model to obtain the trajectories. In order to compute the gradient of the training loss, the Discrete Adjoint loss does $K$ additional forward passes to evaluate the base flow model, one forward and backward pass on the reward model, and $K$ backward passes on the current flow model, which typically use gradient checkpointing to avoid memory overflow. In the case of Adjoint Matching, solving the lean adjoint ODE requires one forward and backward pass on the reward model, and $K$ backward passes on the base flow model. Finally, computing the gradient of the loss takes $K/2$ additional backward passes if we evaluate at only half of the timesteps as we do, although this computation is much quicker because it can be fully parallelized.
> >
> > Meanwhile, computing the gradient of the Continuous Adjoint loss takes 204 seconds. With respect to Adjoint Matching, Continuous Adjoint performs additional backward passes to compute the gradients $\nabla_{X_t} \|u(X_t,t)\|^2$ when solving the adjoint ODE. Finally, we observe that models that directly fine-tune the reward are quicker, but that comes with its own set of issues that we discuss throughout the paper.
> >
> > - **"In addition, for Diffusion-DPO, how the preference pairs are sampled for further fine-tuning? Can authors explain why Diffusion-DPO tuned models yield a decrease in performance?"**
> >
> > We want to emphasize that despite the similarities, even though the loss $L_{\mathrm{DPO}}$ that we use (see equation 234) is very similar to the one implemented by Wallace et al., 2023, the preference data pairs that we use are very different from theirs. We sample the preference data from the current model, which results in imperfect samples, and rank the pairs according to the reward model, while they consider off-policy, high-quality, curated preference samples. The reason for this discrepancy is that the starting point of our work is a reward model, not a set of preference data, and we only benchmark against approaches that leverage reward models for an apples-to-apples comparison. Our experimental results on DPO (Tab. 2, Fig. 5, Tab. 3) show that the resulting model performs like the base model, or a bit worse according to some metrics. Hence, we conclude that our version of DPO is not a competitive alternative for on-policy fine-tuning when the base model is not already good.
> >
> > We added this comment at the end of Appendix G.2.

---

> > > ### Comment · Reviewer_4cB8 · 2024-12-01
> > >
> > > Thank the authors for the detailed responses, and I have no further questions. It's a nice paper bringing forth ideas of adjoints in stochastic control for diffusion models fine-tuning, so I would like to keep my current score.
> > >
> > > For the Diffusion-DPO, I do suggest the authors to do further experiments to formulate proper preference datasets, for example best of n sampling (like n =5,7) which is widely adopted already in RLHF for LLM literature, or the online DPO to yield a competitive baseline for Diffusion DPO and further validate the significance and advantage of the proposed methods.

---

### Author Response · Authors · 2024-11-25
**General response**

We thank the reviewers for their helpful suggestions.  We appreciate that the reviewers were supportive of our contributions and praised our theoretical framework (both memoryless schedules and adjoint matching), as well as our experimental results and our writing.  Based on the reviewers’ feedback, we have improved our manuscript further and uploaded a revised version.

Apart from editing some sentences for clarity as requested and fixing typos, the major changes that we have introduced are as follows:
- We introduced Figure 5 in Appendix A, which shows the reward model value, the control cost term and an evaluation metric (ClipScore) across fine-tuning iterations, for several fine-tuning algorithms. That provides further quantitative insights on the behavior of Adjoint Matching compared to its alternatives, which is a point that Reviewer MA7f raised.
- We added an additional subsection in Appendix H, titled “Remarks on computational costs”. We did this in response to questions by Reviewers 4cB8 and MA7f. While we reported the fine-tuning wall-clock times for each algorithm in Table 3 (Appendix A) already in our first submission, we had not included a detailed discussion.
- We added an additional subsection in Appendix H, titled “Remarks on number of sampling timesteps”. Reviewer MA7f inquired about the performance of Adjoint Matching at a small number of timesteps. While we only performed fine-tuning at 40 timesteps, in our initial submission we already showed experimental results at 10, 20, 100 and 200 timesteps in Table 7 (Appendix); an explanation of the results is now also included.

To summarize, our initial submission already contained a lot of experimental material in Appendix A, but many of it was not properly referenced or commented upon, and reviewers either missed it or requested more detailed comments about it. We have improved our work by commenting on relevant aspects that reviewers singled out. We encourage the reviewers to raise their scores if they are satisfied with our answers and changes.

---

> ### Author Response · Authors · 2024-12-01
> **Reminder of discussion period ending**
>
> We thank the reviewers for their work. We made some changes and additions to our manuscript following their advice, which are summarized in the "General response" above, and discussed in detail in our individual responses to the reviewers. We would like to remind the reviewers which have not yet replied to our responses (4cB8, 7e67, pgS7) that the discussion period extension will come to an end soon. We encourage them to increase their score if they find our responses and revisions satisfactory.

---

### Meta-Review · Area_Chair_guuY · 2024-12-21

**Metareview:**

This paper addresses the challenge of reward fine-tuning in dynamical generative models, such as Flow Matching and denoising diffusion models, by framing it as a stochastic optimal control (SOC) problem. It introduces a theoretically grounded approach requiring a specific memoryless noise schedule during fine-tuning to account for noise-sample dependencies. The proposed algorithm, Adjoint Matching, reformulates SOC as a regression problem, outperforming existing methods by improving consistency, realism, generalization to unseen human preferences, and sample diversity. Following the author response and author-reviewer discussions, this paper has received unanimous support from the reviewers. Therefore, I recommend acceptance. In the final version, the authors are encouraged to include a discussion on SPIN-Diffusion, a closely related work focused on fine-tuning diffusion models, which has demonstrated greater effectiveness compared to Diffusion-DPO.

**Additional Comments On Reviewer Discussion:**

The initial reviews highlighted concerns about insufficient experimental comparisons with existing works and a few instances of unclear writing. The authors' rebuttal addressed most of these issues, leading two reviewers (MA7f and pgS7) to raise their scores following the rebuttal and subsequent discussions.

---

### Decision · Program_Chairs · 2025-01-22

Accept (Spotlight)